# A Mendelian randomization study of the role of lipoprotein subfractions in coronary artery disease

Qingyuan Zhao[1]*, Jingshu Wang[2], Zhen Miao[3], Nancy R Zhang[4], Sean Hennessy[3], Dylan S Small[4], Daniel J Rader[3,5]

[1]Statistical Laboratory, University of Cambridge, Cambridge, United Kingdom; [2]Department of Statistics, University of Chicago, Chicago, United States; [3]Perelman School of Medicine, University of Pennsylvania, Philadelphia, United States; [4]Department of Statistics, University of Pennsylvania, Philadelphia, United States; [5]Department of Medicine, University of Pennsylvania, Philadelphia, United States

**Abstract** Recent genetic data can offer important insights into the roles of lipoprotein subfractions and particle sizes in preventing coronary artery disease (CAD), as previous observational studies have often reported conflicting results. We used the LD score regression to estimate the genetic correlation of 77 subfraction traits with traditional lipid profile and identified 27 traits that may represent distinct genetic mechanisms. We then used Mendelian randomization (MR) to estimate the causal effect of these traits on the risk of CAD. In univariable MR, the concentration and content of medium high-density lipoprotein (HDL) particles showed a protective effect against CAD. The effect was not attenuated in multivariable analyses. Multivariable MR analyses also found that small HDL particles and smaller mean HDL particle diameter may have a protective effect. We identified four genetic markers for HDL particle size and CAD. Further investigations are needed to fully understand the role of HDL particle size.

*For correspondence: qyzhao@statslab.cam.ac.uk

**Competing interests:** The authors declare that no competing interests exist.

## Introduction

Lipoprotein subfractions have been increasingly studied in epidemiological research and used in clinical practice to predict the risk of cardiovascular diseases (CVD) (*Rankin et al., 2014*; *Mora et al., 2009*; *China Kadoorie Biobank Collaborative Group et al., 2018*). Several studies have identified potentially novel subfraction predictors for CVD (*Mora et al., 2009*; *Hoogeveen et al., 2014*; *Williams et al., 2014*; *Ditah et al., 2016*; *Lawler et al., 2017*; *Fischer et al., 2014*) and demonstrated that the addition of subfraction measurements can significantly improve the risk prediction for CVD (*Würtz et al., 2012*; *van Schalkwijk et al., 2014*; *McGarrah et al., 2016*; *Rankin et al., 2014*). However, these observational studies often provide conflicting evidence on the precise roles of the lipoprotein subfractions. For example, while some studies suggested that small, dense low-density lipoprotein (LDL) particles may be more atherogenic (*Lamarche et al., 1997*; *Hoogeveen et al., 2014*), others found that larger LDL size is associated with higher CVD risk (*Campos et al., 2001*; *Mora, 2009*). Some recent observational studies found that the inverse association of CVD outcomes with smaller high-density lipoprotein (HDL) particles is stronger than the association with larger HDL particles (*Ditah et al., 2016*; *Kim et al., 2016*; *McGarrah et al., 2016*; *Silbernagel et al., 2017*), but other studies reached the opposite conclusion in different cohorts (*Li et al., 2016*; *Arsenault et al., 2009*). Currently, the utility of lipoprotein subfractions or particle sizes in routine clinical practice remains controversial (*Superko, 2009*; *Mora, 2009*; *Davidson et al., 2011*; *Bays et al., 2016*), as there is still a great uncertainty about their causal roles in CVD, largely due to a lack of intervention data (*Bays et al., 2016*).

Mendelian randomization (MR) is an useful causal inference method that avoids many common pitfalls of observational cohort studies (*Smith and Ebrahim, 2003*). By using genetic variation as instrumental variables, MR asks if the genetic predisposition to a higher level of the exposure (in this case, lipoprotein subfractions) is associated with higher occurrences of the disease outcome (*Didelez and Sheehan, 2007*). A positive association suggests a causally protective effect of the exposure if the genetic variants satisfy the instrumental variable assumptions (*Didelez and Sheehan, 2007*; *Davey Smith and Hemani, 2014*). Since MR can provide unbiased causal estimate even when there are unmeasured confounders, it is generally considered more credible than other non-randomized designs and is quickly gaining popularity in epidemiological research (*Gidding et al., 2012*; *Davies et al., 2018*). MR has been used to estimate the effect of several metabolites on CVD, but most prior studies are limited to just one or a few risk exposures at a time (*Emdin et al., 2016*; *Ference et al., 2017*).

In this study, we will use recent genetic data to investigate the roles of lipid and lipoprotein traits in the occurrence of coronary artery disease (CAD) and myocardial infarction (MI). In particular, we are interested in discovering lipoprotein subfractions that may be causal risk factors for CAD and MI in addition to the traditional lipid profile (LDL cholesterol, HDL cholesterol, and triglycerides levels). To this end, we will first estimate the genetic correlation of the lipoprotein subfractions and particle sizes with the tradition risk factors and remove the traits that have a high genetic correlation. We will then use MR to estimate the causal effects of the selected lipoprotein subfractions and particle sizes on CAD and MI. Finally, we will explore potential genetic markers for the identified lipoprotein and subfraction traits.

## Materials and methods

### GWAS summary datasets and lipoprotein particle measurements

*Table 1* describes all GWAS summary datasets used in this study, including two GWAS of the traditional lipid risk factors (*Willer et al., 2013*; *Hoffmann et al., 2018*), two recent GWAS of the human lipidome (*Kettunen et al., 2016*; *Davis et al., 2017*), and three GWAS of CAD or MI (*Nikpay et al., 2015*; *Nelson et al., 2017*; *Abbott et al., 2018*). In the two GWAS of the lipidome (*Kettunen et al., 2016*; *Davis et al., 2017*), high-throughput nuclear magnetic resonance (NMR) spectroscopy was used to measure the circulating lipid and lipoprotein traits (*Soininen et al., 2009*). We investigated

**Table 1.** Information about the GWAS summary datasets used in this article.

The columns are the phenotypes reported by the GWAS studies, the consortium or name of the first author of the publication, PubMed ID, population, sample size, other GWAS datasets with other lapping sample, and URLs we used to download the datasets.

| Phenotype | Dataset name | PubMed ID | Population | Sample size | Sample overlap with other datasets | URL to summary dataset |
|---|---|---|---|---|---|---|
| Traditional lipid traits | GERA | 29507422 *Hoffmann et al., 2018* | Multi-ethnic | 94,674 | | ftp://ftp.ebi.ac.uk/pub/databases/gwas/summary_statistics/ |
| | GLGC | 24097068 *Willer et al., 2013* | European | 188,578 | Kettunen, CARDIoGRAMplusC4D | http://csg.sph.umich.edu/abecasis/public/lipids2013/ |
| Lipoprotein subfraction traits | Davis | 29084231 *Davis et al., 2017* | Finnish | 8372 | | http://csg.sph.umich.edu/boehnke/public/metsim-2017-lipoproteins/ |
| | Kettunen | 27005778 *Kettunen et al., 2016* | European | 24,925 | GLGC, CARDIoGRAMplusC4D | http://www.computationalmedicine.fi/data#NMR_GWAS |
| Heart disease traits | CARDIoGRAMplusC4D (CAD) | 26343387 *Nikpay et al., 2015* | Mostly European | 185,000 | GLGC, Kettunen | http://www.cardiogramplusc4d.org/data-downloads/ |
| | CARDIoGRAMplusC4D + UK Biobank (CAD) | 28714975 *Nelson et al., 2017* | Mostly European | | | |
| | UK Biobank (MI) | Interim round two release *Abbott et al., 2018* | European | 360,420 | | http://www.nealelab.is/uk-biobank/ |

the 82 lipid and lipoprotein traits measured in these studies that are related to very-low-density lipo-protein (VLDL), LDL, intermediate-density lipoprotein (IDL), and HDL subfractions and particle sizes. All the subfraction traits are named with three components that are separated by hyphens: the first component indicates the size (XS, S, M, L, XL, XXL); the second component indicates the fraction according to the lipoprotein density (VLDL, LDL, IDL, HDL); the third component indicates the measurement (C for total cholesterol, CE for cholesterol esters, FC for free cholesterol, L for total lipids, P for particle concentration, PL for phospholipids, TG for triglycerides). For example, M-HDL-P refers to the concentration of medium HDL particles.

Aside from the concentration and content of lipoprotein subfractions, the two lipidome GWAS also measured the traditional lipid traits (TG, LDL-C, HDL-C), the average diameter of the fractions (VLDL-D, LDL-D, HDL-D) and the concentration of apolipoprotein A1 (ApoA1) and apolipoprotein B (ApoB). A full list of the lipoprotein measurements investigated in this article can be found in Appendix 1.

## Genetic correlation and phenotypic screening

Genetic correlation is a measure of association between the genetic determinants of two pheno-types. It is conceptually different from epidemiological correlation that can be directly estimated from cross-sectional data. In this study, we applied the LD-score regression (*Bulik-Sullivan et al., 2015*) to the lipidome GWAS (*Kettunen et al., 2016*; *Davis et al., 2017*) to estimate the genetic correlations between the lipoprotein subfractions, particle sizes, and traditional risk factors. We then removed lipoprotein subfractions and particle sizes that are strongly correlated with the traditional risk factors, defined as an estimated genetic correlation > 0.8 with TG, LDL-C, HDL-C, ApoB, or ApoA1 in the GWAS published by *Davis et al., 2017*. Because these traits are largely co-determined with the traditional risk factors, they do not represent independent biological mechanisms and may lead to multicollinearity issues in multivariate MR analyses. Finally, we obtained an independent estimate of the genetic correlations between the selected traits by applying the LD score regression to the GWAS published by *Kettunen et al., 2016*. We used Bonferroni's procedure to correct for multiple testing (familywise error rate at 0.05).

## Three-sample Mendelian randomization design

For MR, we employed a three-sample design (*Zhao et al., 2019b*) in which one GWAS was used to select independent genetic instruments that are associated with one or several lipoprotein measures. The other two GWAS were then used to obtain summary associations of the selected SNPs with the exposure and the outcome, as in a typical two-sample MR design (*Pierce and Burgess, 2013*; *Hemani et al., 2016*). More specifically, the selection GWAS was used to create a set of SNPs that are in linkage equilibrium with each other in a reference panel (distance >10 megabase pairs, $r^2$<0.001). This was done by ordering the SNPs by the p-values of their association with the trait(s) under investigation and then selecting them greedily using the linkage-disequilibrium (LD) clumping function in the PLINK software package (*Purcell et al., 2007*). To avoid winner's curse, we require the other two GWAS to have no overlapping sample with the selection GWAS.

As the GWAS published by *Davis et al., 2017* has a smaller sample size, we used it to select the genetic instruments so the larger dataset can be used for statistical estimation. In univariable MR, associations of the selected SNPs with the exposure trait (a lipoprotein subfraction or a particle size trait) were obtained from the GWAS published by *Kettunen et al., 2016* and the associations with MI were obtained using summary data from an interim release of UK BioBank (*Abbott et al., 2018*). To maximize the statistical power, we used the so-called 'genome-wide MR' design. Independent SNPs are selected by using LD clumping, but we do not truncate the list of SNPs by their p-values. More details about this design can be found in a previous methodological article (*Zhao et al., 2019b*).

To control for potential pleiotropic effects via the traditional risk factors, we performed two multi-variable MR analyses for each lipoprotein subfraction or particle size under investigation. The first multivariable MR analysis considers four exposures: TG, LDL-C, HDL-C, and the lipoprotein measure-ment under investigation. The second multivariable MR analysis replaces LDL-C and HDL-C with ApoB and ApoA1, in accordance with some recent studies (*Richardson et al., 2020*). SNPs were ranked by their minimum p-values with the four exposures and are selected as instruments only if

they were associated with at least one of the four exposures (p-value $\leq 10^{-4}$). Both multivariable MR analyses used the Davis (*Davis et al., 2017*) and GERA (*Hoffmann et al., 2018*) datasets for instrument selection, the Kettunen (*Kettunen et al., 2016*) and GLGC (*Willer et al., 2013*) datasets for the associations of the instruments with the exposures, and the CARDIoGRAMplusC4D + UK Biobank (*Nelson et al., 2017*) dataset for the associations with CAD.

### Statistical estimation

For univariable MR, we used the robust adjusted profile score (RAPS) because it is more efficient and robust than many conventional methods (*Zhao et al., 2020*; *Zhao et al., 2019b*). RAPS can consistently estimate the causal effect even when some of the genetic variants violate instrumental variables assumptions. For multivariable MR, we used an extension to RAPS called GRAPPLE to obtain the causal effect estimates of multiple exposures (*Wang et al., 2020*). GRAPPLE also allows the exposure GWAS to have overlapping sample with the outcome GWAS, while the original RAPS does not. We assessed the strength of the instruments using the modified Cochran's Q statistic (*Sanderson et al., 2019*). Because many lipoprotein subfraction traits were analyzed simultaneously, we used the Benjamini-Hochberg procedure to correct for multiple testing (*Benjamini and Hochberg, 1995*) and the false discovery rate was set to be 0.05. More detail about the statistical methods can be found in Appendix 3.

### Genetic markers for lipoprotein subfractions and CAD

To obtain genetic markers, we selected SNPs that are associated with the lipoprotein measurements identified in the MR (p-value $\leq 5 \times 10^{-8}$) and CAD (p-value $\leq 0.05$) but are not associated with LDL-C or ApoB (p-value $\geq 10^{-3}$). To maximize the power of this exploratory analysis, we meta-analyzed the results of the two lipidome GWAS (*Kettunen et al., 2016*; *Davis et al., 2017*) by inverse-variance weighting. For the associations with LDL-C and CAD, we used the GWAS summary data reported by the GLGC (*Willer et al., 2013*) and CARDIoGRAMplusC4D (*Nelson et al., 2017*) consortia. We used LD clumping to obtain independent markers (*Purcell et al., 2007*) and then validate the markers using tissue-specific gene expression data from the GTEx project.

### Sensitivity analysis and replicability

Because we had multiple GWAS summary datasets for the lipoprotein subfractions and CAD/MI (*Table 1*), we swapped the roles of the GWAS datasets in the three-sample MR design whenever permitted by the statistical methods to obtain multiple statistical estimates. These estimates are not completely independent of the primary results, but they can nonetheless be used to assess replicability. As a sensitivity analysis, We further analyzed univariable MR using inverse-variance weighting (IVW) (*Burgess et al., 2013*) and weighted median (*Bowden et al., 2016*) and compared with the primary results obtained by RAPS. We also assessed the assumptions made by RAPS using some diagnostic plots suggested in previous methodological articles (*Zhao et al., 2019b*).

## Results

### Genetic correlations and phenotypic screening

We obtained the genetic correlations of the lipoprotein subfractions and particle sizes with the traditional lipid risk factors: TG, LDL-C, HDL-C, ApoB, and ApoA1 (*Table 1*). We found that almost all VLDL subfractions traits (besides those related to very small VLDL subfraction) and the mean VLDL particle diameter have an estimated genetic correlation with TG very close to 1. Most traits related to the large and very large HDL subfractions also have a high genetic correlation with HDL-C and ApoA1.

After removing traits that are strongly correlated with the traditional risk factors, we obtained 27 traits that may involve independent genetic mechanisms. *Figure 1* shows the genetic correlation matrix for these traits and the traditional lipid factors. The selected traits can be divided into two groups based on whether they are related to VLDL/LDL/IDL particles or HDL particles. Within each group, most traits were strongly correlated with the others. In the first group, most traits had a positive genetic correlation with LDL-C and ApoB, while in the second group, most traits had a positive genetic correlation with HDL-C and ApoA1. Exceptions include LDL-D, which had a negative but

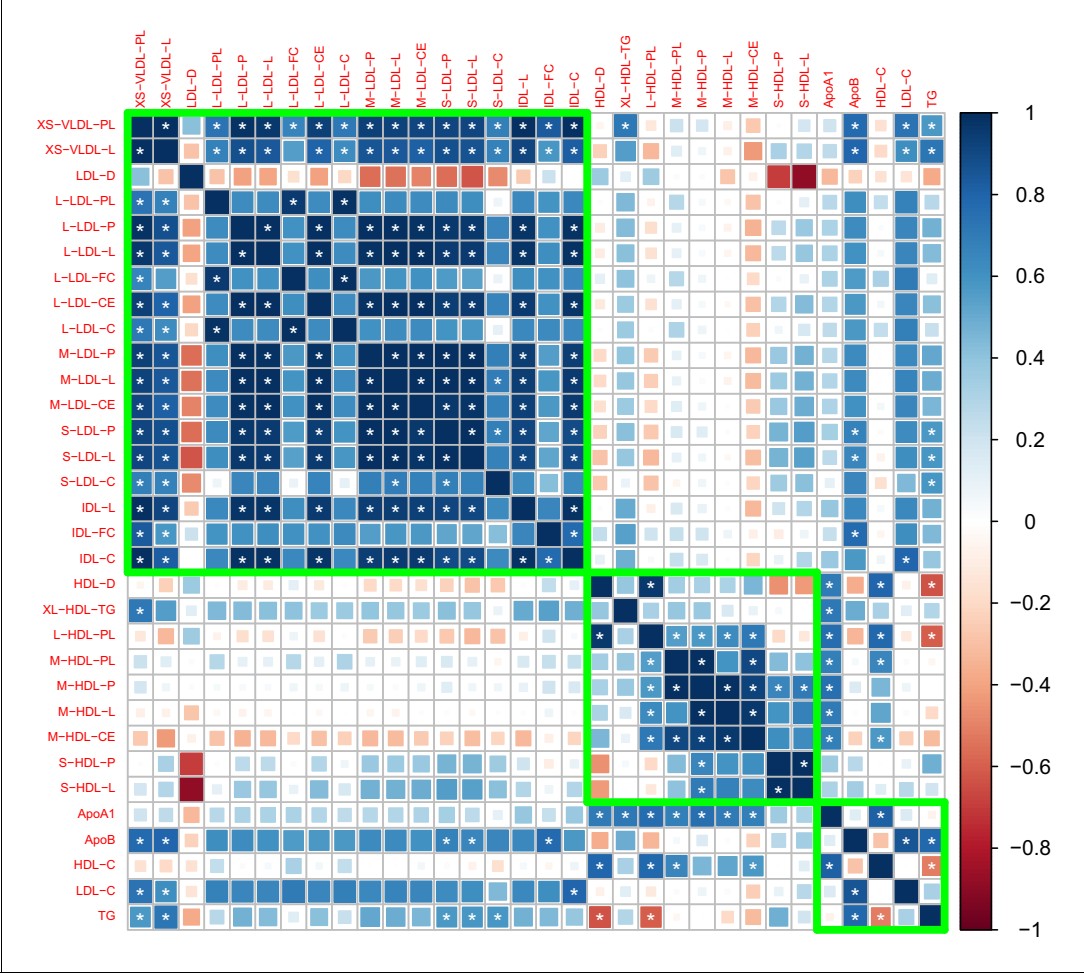

**Figure 1.** Genetic correlation matrix of the 27 lipoprotein subfraction traits selected in phenotypic screening and five traditional lipid traits. White asterisk indicates the correlation is statistically significant after Bonferroni correction for multiple comparisons at level 0.05.

statistically non-significant genetic correlation with LDL-C and ApoB, and S-HDL-P and S-HDL-L, which showed no or weak genetic correlation with HDL-C and ApoA1.

## Mendelian randomization

*Figure 2* shows the estimated causal effect of the selected lipoprotein measurements on MI or CAD that are statistically significant (false discovery rate = 0.05). The unfiltered results can be found in Appendix 3, which also contains results of the sensitivity and replicability analyses.

The concentration and lipid content of VLDL, LDL, and IDL subfractions showed harmful and nearly uniform effects on MI in univariable MR. However, after adjusting for the traditional lipid risk factors, the effects of these ApoB-related subfractions become close to zero (besides IDL-FC in one multivariable analysis). The mean diameter of LDL particles (LDL-D) showed a harmful effect on MI in univariable MR, though the effect was smaller than those of the LDL subfractions in univariable MR. The estimated effect of LDL-D was attenuated in the multivariable MR analyses.

The concentration and content of medium HDL particles showed protective effects in univariable and multivariable MR analyses. In particular, adjusting for the traditional lipid risk factors did not attenuate the effect of traits related to medium HDL. The concentration of and total lipid in small HDL particles showed protective effects in multivariable MR analyses, though the effect sizes were smaller than those of the medium HDL traits. The mean diameter of HDL particles (HDL-D) had almost no effect on MI in the univariable MR analysis, but after adjusting for the traditional lipid risk factors, it showed a harmful effect.

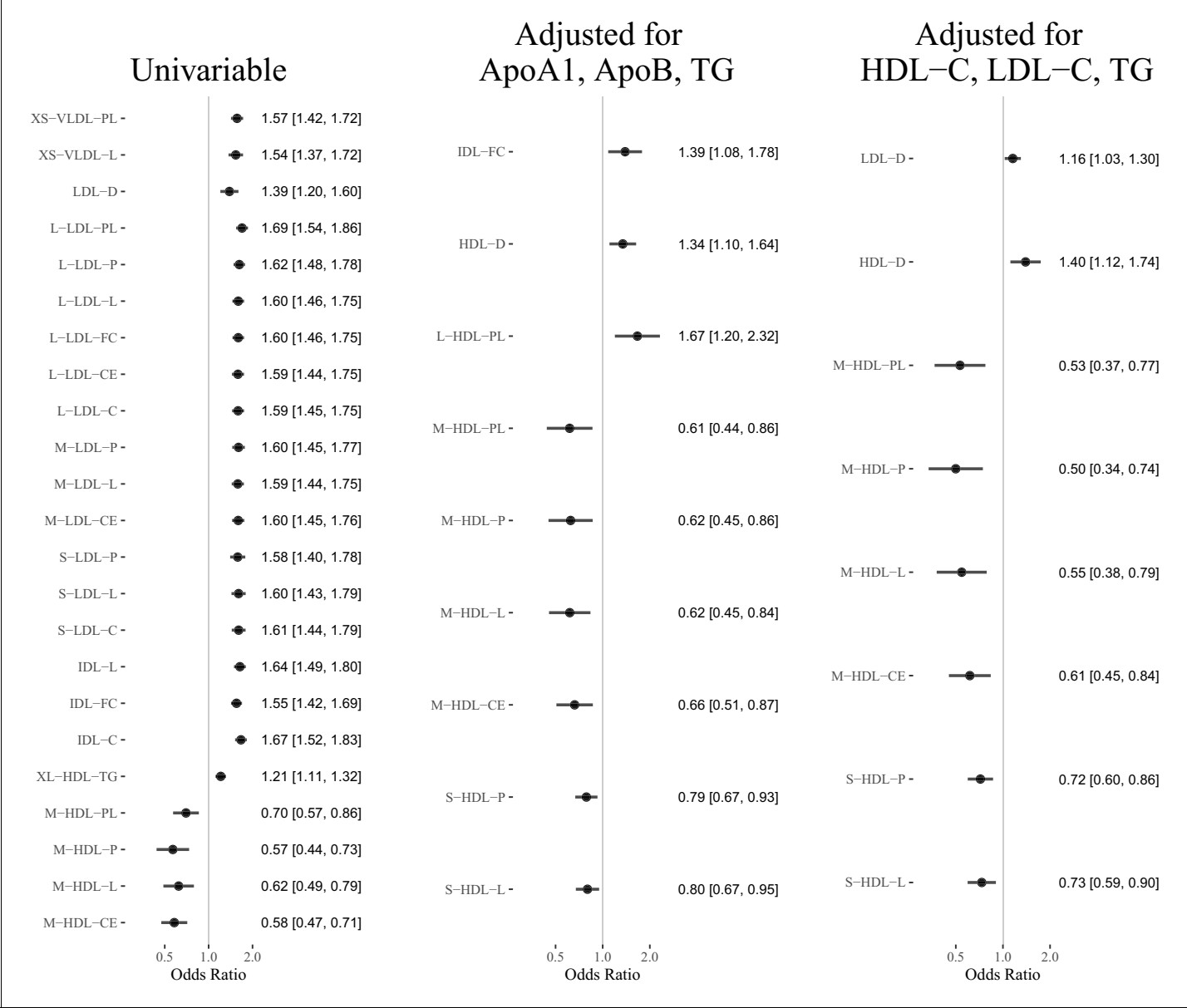

**Figure 2.** Results of the Mendelian randomization analyses (false discover rate = 0.05): Estimated odds ratio [95% confidence interval] per standard deviation increase of the selected lipoprotein measurements on MI or CAD.

*Table 2* reports the estimated effects of M-HDL-P, S-HDL-P, HDL-D, and traditional lipid traits (TG, LDL-C, HDL-C, ApoB, ApoA1) in the multivariable MR analyses. To better understand the role of HDL subfractions and particle sizes, we also included in the table the results of the multivariate MR analyses for the traditional lipid risk factors only. Those baseline analyses suggested that HDL-C/ApoA1 had a weak, non-significant protective effect on CAD, which is consistent with prior studies (*Holmes et al., 2015*; *Wang et al., 2020*). Adding S-HDL-P to the MR analysis did not substantially alter the estimated effects of the traditional lipid traits. However, when M-HDL-P or HDL-D was included in the model, the estimated effects of M-HDL-P and HDL-D changed substantially. In particular, when M-HDL-P was included in the multivariable MR analyses, HDL-C/ApoA1 showed a harmful effect on CAD. When HDL-D was included, HDL-C/ApoA1 showed a protective effect.

**Table 2.** Results of some multivariable Mendelian randomization analyses.

Each row in the table corresponds to a multivariable MR analysis with traditional lipid profile and the specified lipoprotein subfraction or particle size trait. Reported numbers are the point estimates and 95% confidence intervals of the exposure effect.

| Trait | Effect of TG | Effect of LDL-C | Effect of HDL-C | Effect of subfraction/particle size |
|---|---|---|---|---|
| None | 0.19 [0.09,0.29] | 0.38 [0.33,0.44] | −0.053 [−0.13,0.03] | |
| M-HDL-P | 0.37 [0.22,0.52] | 0.39 [0.32,0.45] | 0.30 [0.08,0.52] | −0.69 [−1.09,−0.3] |
| S-HDL-P | 0.23 [0.12,0.33] | 0.45 [0.38,0.52] | −0.11 [−0.2,−0.02] | −0.33 [−0.52,−0.15] |
| HDL-D | 0.11 [0.00,0.22] | 0.42 [0.36,0.49] | −0.44 [−0.69,−0.2] | 0.33 [0.11,0.56] |
| | Effect of TG | Effect of ApoB | Effect of ApoA1 | Effect of Subfraction/Particle size |
| None | 0.05 [−0.05,0.14] | 0.49 [0.38,0.60] | −0.095 [−0.21,0.02] | |
| M-HDL-P | −0.00 [−0.18,0.17] | 0.50 [0.31,0.69] | 0.13 [−0.06,0.32] | −0.47 [−0.80,−0.15] |
| S-HDL-P | 0.07 [−0.03,0.17] | 0.53 [0.41,0.65] | −0.13 [−0.25,−0.02] | −0.24 [−0.40,−0.08] |
| HDL-D | 0.06 [−0.04,0.15] | 0.61 [0.47,0.76] | −0.46 [−0.73,−0.19] | 0.30 [0.08,0.52] |

## Genetic markers associated with HDL subfractions and CAD

We identified four genetic variants that are associated with S-HDL-P, M-HDL-P, or HDL-D, not associated with LDL-C or ApoB, and associated with CAD: rs838880 (*SCARB1*), rs737337 (*DOCK6*),

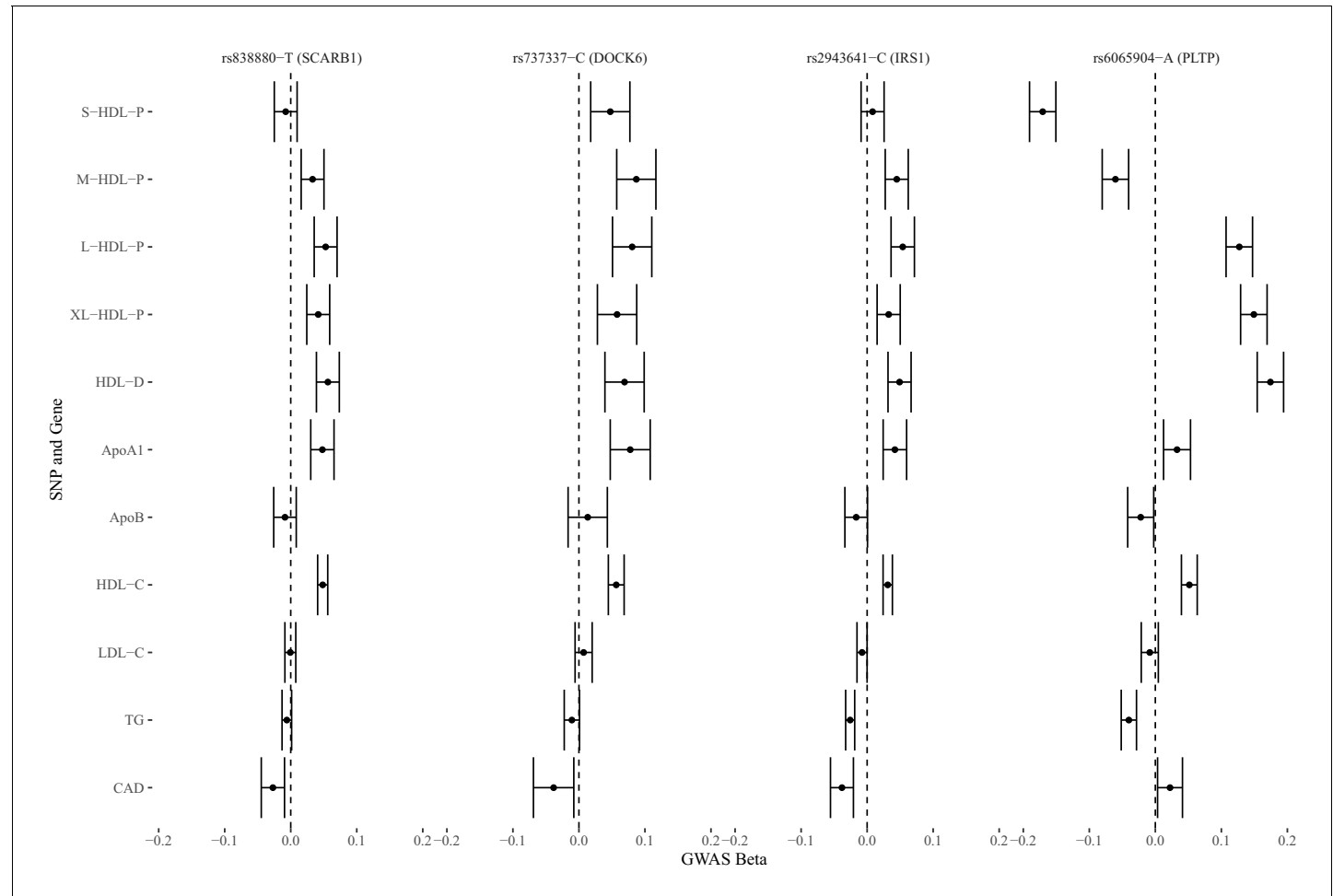

**Figure 3.** Genetic markers for HDL size (with risk alleles) and their associations with various lipid traits.

rs2943641 (*IRS1*), and rs6065904 (*PLTP*) (*Figure 3*). These SNP-cis gene pairs are also supported by examining expression quantitative trait loci (eQTL) in the tissue-specific GTEx data (Appendix 4). The first three variants were not associated with S-HDL-P. However, they had uniformly positive associations with M-HDL-P, L-HDL-P, XL-HDL-P, HDL-D, ApoA1, and HDL-C, and a negative association with CAD. The last variant rs6065904 had positive associations with S-HDL-P and M-HDL-P, negative associations with L-HDL-P, XL-HDL-P, HDL-D, negative but smaller associations with ApoA1 and HDL-C, and a negative association with CAD.

## Sensitivity and replicability analysis

We also investigated the effects of lipoprotein subfractions and particle sizes on MI/CAD using multiple GWAS datasets, MR designs and statistical methods. The results are provided in Appendix 3 and are generally in agreement with the primary results reported above. The diagnostic plots for S-HDL-P and M-HDL-P did not suggest evidence of violations of the instrument strength independent of direct effect (InSIDE) assumption (*Bowden et al., 2015*) made by RAPS and GRAPPLE (Appendix 4).

## Discussion

By using recent genetic data and MR, this study examines whether some lipoprotein subfractions and particle sizes, beyond the traditional lipid risk factors, may play a role in coronary artery disease. We find that VLDL subfractions have extremely high genetic correlations with blood triglyceride level and thus offer little extra value. We find some weak evidence that larger LDL particle size may have a small harmful effect on myocardial infarction and coronary artery disease.

Our main finding is that the size of HDL particles may play an important and previously undiscovered role. Although the concentration and lipid content of small and medium HDL particles appear to be positively correlated with HDL cholesterol and ApoA1, their genetic correlations are much smaller than 1, indicating possible independent biological pathway(s). Moreover, the MR analyses suggested that the small and medium HDL particles may have protective effects on CAD. We also find that larger HDL mean particle diameter may have a harmful effect on CAD. Finally, we identified four potential genetic markers for HDL particle size that are independent of LDL cholesterol and ApoB.

There has been a heated debate on the role of HDL particles in CAD in recent years following the failure of several trials for *CETP* inhibitors (*Barter et al., 2007*; *Schwartz et al., 2012*; *Lincoff et al., 2017*) and recombinant ApoA1 (*Nicholls et al., 2018*) targeting HDL cholesterol. Observational epidemiology studies have long demonstrated strong inverse association between HDL cholesterol and the risk of CAD or MI (*Miller and Miller, 1975*; *Lewington et al., 2007*; *Di Angelantonio et al., 2009*), but conflicting evidence has been found in MR studies. In an influential study, Voight and collaborators found that the genetic variants associated with HDL cholesterol had varied associations with CAD and that almost all variants suggesting a protective effect of HDL cholesterol were also associated with LDL cholesterol or triglycerides (*Voight et al., 2012*). Other MR studies also found that the effect of HDL cholesterol on CAD is heterogeneous (*Zhao et al., 2019b*) or attenuated after adjusting for LDL cholesterol and triglycerides (*Holmes et al., 2017*; *White et al., 2016*).

Notice that the harmful effect of larger HDL particle diameter found in this study relies on including HDL-C or ApoA1 in the multivariable MR analysis. Thus, the role of HDL particles in preventing CAD may be more complicated than, for example, that of LDL cholesterol or ApoB. It is possible that HDL cholesterol, HDL subfractions, and HDL particle size are all phenotypic markers for some underlying causal mechanism. A related theory is the HDL function hypothesis (*Rader and Hovingh, 2014*). Cholesterol efflux capacity, a measure of HDL function, has been documented as superior to HDL-C in predicting CVD risk (*Rohatgi et al., 2014*; *Saleheen et al., 2015*). Recent epidemiologic studies found that HDL particle size is positively associated with cholesterol efflux capacity in postmenopausal women (*El Khoudary et al., 2016*) and in an asymptomatic older cohort (*Mutharasan et al., 2017*). However, mechanistic efflux studies showed that small HDL particles actually mediate more cholesterol efflux (*Favari et al., 2009*; *Du et al., 2015*). A likely explanation of this seeming contradiction is that a high concentration of small HDL particles in the serum may mark a block in maturation of small HDL particles (*Mutharasan et al., 2017*). This can also partly

explain our finding that small HDL traits have a smaller effect than medium HDL traits, as increased medium HDL might indicate successful maturation of small HDL particles.

Among the reported genetic markers, *SCARB1* and *PLTP* have established relations to HDL metabolism and CAD. *SCARB1* encodes a plasma membrane receptor for HDL and is involved in hepatic uptake of cholesterol from peripheral tissues. Recently, a rare mutation (P376L) of *SCARB1* was reported to raise HDL-C level and increase CAD risk (*Zanoni et al., 2016*; *Samadi et al., 2019*). This is opposite direction to the conventional belief that HDL-C is protective and could be explained by HDL dysfunction. *PLTP* encodes the phospholipid transfer protein and mediates the transfer of phospholipid and cholesterol from LDL and VLDL to HDL. As a result, *PLTP* plays a complex but pivotal role in HDL particle size and composition. Several studies have suggested that high *PLTP* activity is a risk factor for CAD (*Schlitt et al., 2003*; *Schlitt et al., 2009*; *Zhao et al., 2019a*).

Our study should be viewed in the context of its limitations, in particular, the inherent limitations of the summary-data MR design. Any causal inference from non-experimental data makes unverifiable assumptions, so does our study. Conventional MR studies assume that the genetic variants are valid instrumental variables. The statistical methods used by us make less stringent assumptions about the instrumental variables, but those assumptions could still be violated even though our model diagnosis does not suggest evidence against the InSIDE assumption. Our study did not adjust for other risk factors for CAD such as body mass index, blood pressure, and smoking. All the GWAS datasets used in this study are from the European population, so the same conclusions might not generalize to other populations. Furthermore, our study used GWAS datasets from heterogeneous subpopulations, which may also introduce bias (*Zhao et al., 2019c*). We also did not use more than one subfraction traits as exposures in multivariable MR because of their high genetic correlations. Alternative statistical methods could be used to select the best causal risk factor from high-through-put experiments (*Zuber et al., 2019*). Finally, as pointed out by revieweres, triglycerides has a greater intra-individual biological variability than HDL particle size. It is likely that triglycerides and HDL size represent a gene/environment interaction with a very large environmental component. Further investigations are needed to fully understand this mechanism.

Recently, a NMR spectroscopy method has been developed to estimate HDL cholesterol efflux capacity from serum (*Kuusisto et al., 2019*). That method can form the basis of a genetic analysis of HDL cholesterol efflux capacity and may complement the results here. We believe more laboratorial and epidemiological research is needed to clarify the roles of HDL subfractions and particle size in cardiovascular diseases.

## Additional information

### Funding
No external funding was received for this work.

### Author contributions
Qingyuan Zhao, Conceptualization, Data curation, Software, Formal analysis, Investigation, Visualization, Methodology, Writing - original draft, Project administration; Jingshu Wang, Conceptualization, Data curation, Software, Validation, Investigation, Methodology, Writing - review and editing; Zhen Miao, Conceptualization, Investigation, Visualization, Writing - review and editing; Nancy R Zhang, Conceptualization, Supervision, Visualization, Methodology, Writing - review and editing; Sean Hennessy, Resources, Supervision, Methodology, Writing - review and editing; Dylan S Small, Conceptualization, Supervision, Investigation, Methodology, Writing - review and editing; Daniel J Rader, Conceptualization, Validation, Investigation, Writing - review and editing

### Author ORCIDs
Qingyuan Zhao (iD) https://orcid.org/0000-0001-9902-2768
Zhen Miao (iD) https://orcid.org/0000-0002-3255-9517
Dylan S Small (iD) http://orcid.org/0000-0003-4928-2646

### Decision letter and Author response
Decision letter https://doi.org/10.7554/eLife.58361.sa1

Author response https://doi.org/10.7554/eLife.58361.sa2

---

## Additional files

### Supplementary files
- Transparent reporting form

### Data availability
GWAS data used in the data are publicly available. Details can be found in Table 1.

---

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

# Appendix 1

## Lipid and lipoprotein traits

Two published GWAS of the human lipidome [Kettunen2016, Davis2017] measured lipoprotein subfractions and particle sizes using NMR spectroscopy. We investigated the 82 lipid and lipoprotein traits measured in these studies that are related to very-low-density lipoprotein (VLDL), LDL, and HDL subfractions and particle sizes. All the subfraction traits are named using three components separated by hyphen: the first indicates the size (XS, S, M, L, XL, XXL); the second indicates the category according to the lipoprotein density (VLDL, LDL, IDL, HDL); the third indicates the measurement (C for total cholesterol, CE for cholesterol esters, FC for free cholesterol, L for total lipids, P for particle concentration, PL for phospholipids, TG for triglycerides). A full list of lipid and lipoprotein traits used in our study can be found in *Appendix 1—table 1* below.

**Appendix 1—table 1.** All 82 traits included in this study and whether they are measured in the Kettunen and Davis GWAS (NA means not available).

| Trait | Description | Kettunen | Davis |
|---|---|---|---|
| VLDL traits and total triglycerides | | | |
| TG | Total triglycerides | | |
| VLDL-D | VLDL diameter | | |
| XS-VLDL-L | Total lipids in very small VLDL | | NA |
| XS-VLDL-P | Concentration of very small VLDL particles | | |
| XS-VLDL-PL | Phospholipids in very small VLDL | | |
| XS-VLDL-TG | Triglycerides in very small VLDL | | |
| S-VLDL-C | Total cholesterol in small VLDL | | |
| S-VLDL-FC | Free cholesterol in small VLDL | | |
| S-VLDL-L | Total lipids in small VLDL | | NA |
| S-VLDL-P | Concentration of small VLDL particles | | |
| S-VLDL-PL | Phospholipids in small VLDL | | |
| S-VLDL-TG | Triglycerides in small VLDL | | |
| M-VLDL-C | Total cholesterol in medium VLDL | | |
| M-VLDL-CE | Cholesterol esters in medium VLDL | | |
| M-VLDL-FC | Free cholesterol in medium VLDL | | |
| M-VLDL-L | Total lipids in medium VLDL | | NA |
| M-VLDL-P | Concentration of medium VLDL particles | | |
| M-VLDL-PL | Phospholipids in medium VLDL | | |
| M-VLDL-TG | Triglycerides in medium VLDL | | |
| L-VLDL-C | Total cholesterol in large VLDL | | |
| L-VLDL-CE | Cholesterol esters in large VLDL | | |
| L-VLDL-FC | Free cholesterol in large VLDL | | |
| L-VLDL-L | Total lipids in large VLDL | | NA |
| L-VLDL-P | Concentration of large VLDL particles | | |
| L-VLDL-PL | Phospholipids in large VLDL | | |
| L-VLDL-TG | Triglycerides in large VLDL | | |
| XL-VLDL-L | Total lipids in very large VLDL | | NA |
| XL-VLDL-P | Concentration of very large VLDL particles | | |
| XL-VLDL-PL | Phospholipids in very large VLDL | | |
| XL-VLDL-TG | Triglycerides in very large VLDL | | |

*Continued on next page*

*Appendix 1—table 1 continued*

| Trait | Description | Kettunen | Davis |
|---|---|---|---|
| XXL-VLDL-L | Total lipids in chylomicrons and extremely very large VLDL | | NA |
| XXL-VLDL-P | Concentration of chylomicrons and extremely very large VLDL particles | | |
| XXL-VLDL-PL | Phospholipids in chylomicrons and extremely very large | | |
| XXL-VLDL-TG | Triglycerides in chylomicrons and extremely very large | | |
| LDL and IDL traits | | | |
| LDL-C | Total cholesterol in LDL | | |
| ApoB | Apolipoprotein B | | |
| LDL-D | LDL diameter | | |
| S-LDL-C | Total cholesterol in small LDL | | |
| S-LDL-L | Total lipids in small LDL | | NA |
| S-LDL-P | Phospholipids in small LDL | | |
| M-LDL-C | Total cholesterol in medium LDL | | |
| M-LDL-CE | Cholesterol esters in medium LDL | | |
| M-LDL-L | Total lipids in medium LDL | | NA |
| M-LDL-P | Concentration of medium LDL particles | | |
| M-LDL-PL | Phospholipids in medium LDL | | |
| L-LDL-C | Total cholesterol in large LDL | | |
| L-LDL-CE | Cholesterol esters in large LDL | | |
| L-LDL-FC | Free cholesterol in large LDL | | |
| L-LDL-L | Total lipids in large LDL | | NA |
| L-LDL-P | Concentration of large LDL particles | | |
| L-LDL-PL | Phospholipids in large LDL | | |
| IDL-C | Total cholesterol in IDL | | |
| IDL-FC | Free cholesterol in IDL | | |
| IDL-L | Total lipids in IDL | | NA |
| IDL-P | Concentration of IDL particles | | |
| IDL-PL | Phospholipids in IDL | | |
| IDL-TG | Triglycerides in IDL | | |
| HDL traits | | | |
| HDL-C | Total cholesterol in HDL | | |
| ApoA1 | Apolipoprotein A1 | | |
| HDL-D | HDL diameter | | |
| S-HDL-L | Total lipids in small HDL | | NA |
| S-HDL-P | Concentration of small HDL particles | | |
| S-HDL-TG | Triglycerides in small HDL | | |
| M-HDL-C | Total cholesterol in medium HDL | | |
| M-HDL-CE | Cholesterol esters in medium HDL | | |
| M-HDL-FC | Free cholesterol in medium HDL | | |
| M-HDL-L | Total lipids in medium HDL | | NA |
| M-HDL-P | Concentration of medium HDL particles | | |
| M-HDL-PL | Phospholipids in medium HDL | | |
| L-HDL-C | Total cholesterol in large HDL | | |

*Continued on next page*

*Appendix 1—table 1 continued*

| Trait | Description | Kettunen | Davis |
|-------|-------------|----------|-------|
| L-HDL-CE | Cholesterol esters in large HDL | | |
| L-HDL-FC | Free cholesterol in large HDL | | |
| L-HDL-L | Total lipids in large HDL | | NA |
| L-HDL-P | Concentration of large HDL particles | | |
| L-HDL-PL | Phospholipids in large HDL | | |
| XL-HDL-C | Total cholesterol in very large HDL | | |
| XL-HDL-CE | Cholesterol esters in very large HDL | | |
| XL-HDL-FC | Free cholesterol in very large HDL | | |
| XL-HDL-L | Total lipids in very large HDL | | NA |
| XL-HDL-P | Concentration of very large HDL particles | | |
| XL-HDL-PL | Phospholipids in very large HDL | | |
| XL-HDL-TG | Triglycerides in very large HDL | | |

## Appendix 2

## Genetic correlations

We estimated the genetic correlation between lipoprotein subfractions, particle sizes, and traditional lipid risk factors using the LD score regression (*Li et al., 2016*). *Appendix 2—figure 1–3* show the estimated genetic correlation matrix between selected traits using different datasets. Below the figures, *Appendix 2—table 1* shows the estimated genetic correlations of the lipoprotein subbfractions with the traditional lipid risk factors using the Davis GWAS. The results in *Appendix 2—table 1* were then used to screen the traits as described in Materials and methods.

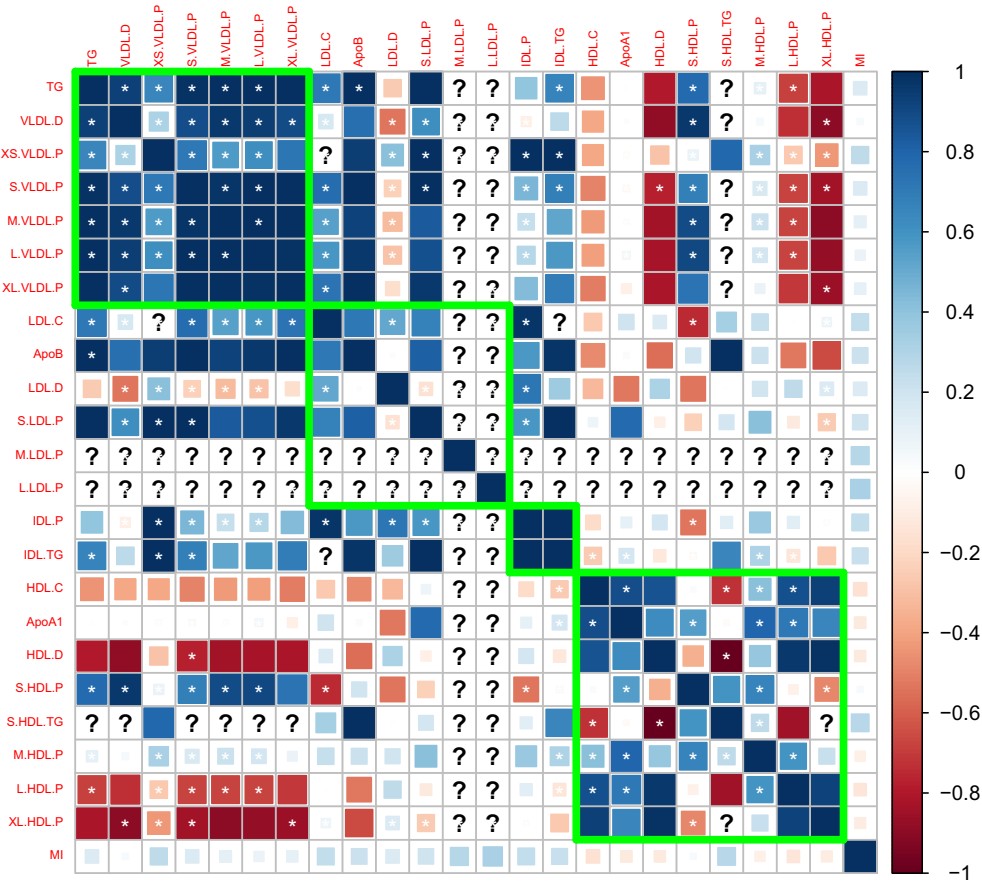

**Appendix 2—figure 1.** Genetic correlations computed using the *Davis et al., 2017* GWAS summary dataset.

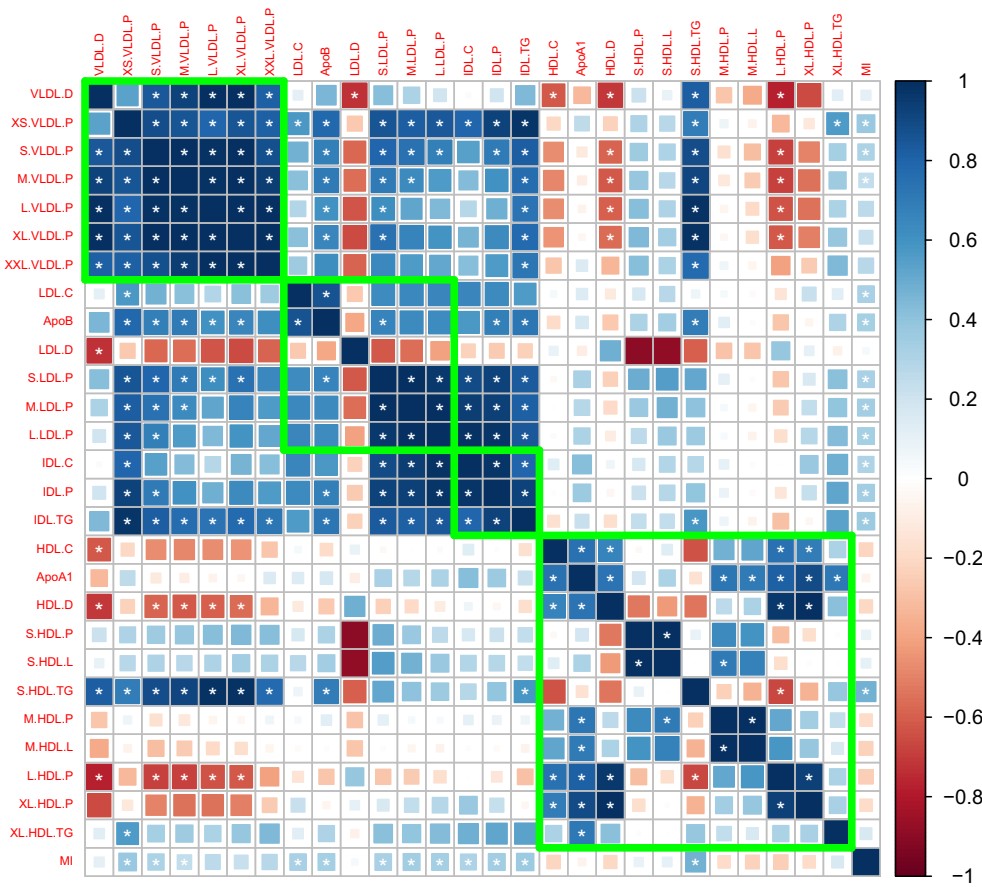

**Appendix 2—figure 2.** Genetic correlations computed using the *Kettunen et al., 2016* GWAS summary dataset.

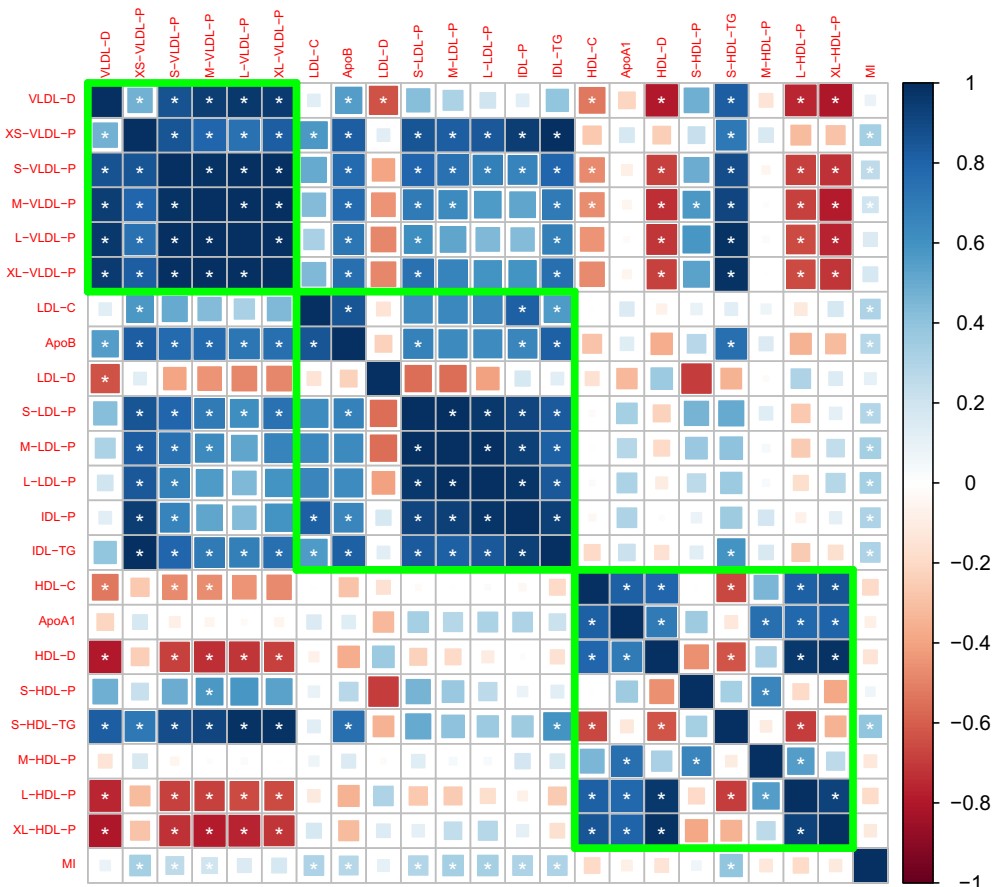

**Appendix 2—figure 3.** Genetic correlations computed by meta-analyzing the results in *Appendix 2—figures 1* and *2*.

**Appendix 2—table 1.** Estimated genetic correlation (standard error) of the lipoprotein subfractions with the traditional lipid risk factors using the Davis GWAS.

Bolded estimates are above 0.8 and the corresponding traits were removed in phenotypic screening.

| Trait | ApoA1 | ApoB | HDL-C | LDL-C | TG |
|---|---|---|---|---|---|
| S-HDL-L | 0.31 (0.28) | 0.34 (0.25) | 0.13 (0.26) | 0.27 (0.3) | 0.2 (0.22) |
| S-HDL-P | 0.36 (0.24) | 0.27 (0.22) | −0.01 (0.22) | 0.1 (0.31) | 0.48 (0.17) |
| S-HDL-TG | −0.13 (0.25) | 0.77 (0.13) | −0.66 (0.15) | 0.13 (0.28) | 1.03 (0.07) |
| M-HDL-C | 0.65 (0.14) | −0.18 (0.2) | 0.81 (0.09) | −0.09 (0.25) | −0.34 (0.17) |
| M-HDL-CE | 0.68 (0.14) | −0.23 (0.21) | 0.57 (0.12) | −0.24 (0.24) | −0.32 (0.18) |
| M-HDL-FC | 0.67 (0.12) | −0.08 (0.21) | 0.83 (0.08) | 0.04 (0.24) | −0.28 (0.18) |
| M-HDL-L | 0.71 (0.15) | 0.02 (0.27) | 0.52 (0.17) | −0.03 (0.29) | −0.19 (0.25) |
| M-HDL-P | 0.75 (0.12) | 0.15 (0.23) | 0.46 (0.14) | 0.08 (0.26) | 0 (0.19) |
| M-HDL-PL | 0.69 (0.13) | 0.04 (0.22) | 0.65 (0.11) | 0.02 (0.25) | −0.04 (0.19) |
| L-HDL-C | 0.76 (0.11) | −0.42 (0.13) | 0.95 (0.02) | −0.1 (0.18) | −0.62 (0.09) |
| L-HDL-CE | 0.82 (0.1) | −0.4 (0.12) | 0.93 (0.04) | −0.16 (0.17) | −0.62 (0.09) |
| L-HDL-FC | 0.66 (0.12) | −0.46 (0.13) | 0.92 (0.03) | −0.13 (0.18) | −0.7 (0.08) |
| L-HDL-L | 0.81 (0.11) | −0.29 (0.15) | 0.74 (0.07) | −0.15 (0.18) | −0.56 (0.12) |
| L-HDL-P | 0.79 (0.09) | −0.35 (0.13) | 0.82 (0.05) | −0.12 (0.16) | −0.61 (0.09) |

*Continued on next page*

*Appendix 2—table 1 continued*

| Trait | ApoA1 | ApoB | HDL-C | LDL-C | TG |
|---|---|---|---|---|---|
| L-HDL-PL | 0.77 (0.09) | −0.34 (0.13) | 0.79 (0.05) | −0.12 (0.17) | −0.61 (0.09) |
| XL-HDL-C | 0.75 (0.16) | −0.25 (0.19) | 0.9 (0.1) | 0.4 (0.27) | −0.63 (0.13) |
| XL-HDL-CE | 0.82 (0.16) | −0.17 (0.19) | 0.82 (0.09) | 0.41 (0.27) | −0.54 (0.12) |
| XL-HDL-FC | 0.72 (0.14) | −0.37 (0.18) | 0.94 (0.08) | 0.17 (0.23) | −0.71 (0.11) |
| XL-HDL-L | 0.93 (0.16) | −0.08 (0.25) | 0.68 (0.14) | 0.1 (0.27) | −0.35 (0.2) |
| XL-HDL-P | 0.81 (0.13) | −0.32 (0.16) | 0.86 (0.08) | 0.17 (0.21) | −0.69 (0.11) |
| XL-HDL-PL | 0.76 (0.12) | −0.41 (0.15) | 0.83 (0.07) | −0.09 (0.18) | −0.7 (0.09) |
| XL-HDL-TG | 0.72 (0.13) | 0.49 (0.17) | 0.33 (0.13) | 0.13 (0.26) | 0.3 (0.15) |
| HDL-D | 0.7 (0.11) | −0.36 (0.13) | 0.8 (0.06) | −0.08 (0.17) | −0.64 (0.09) |
| IDL-C | 0.38 (0.21) | 0.58 (0.19) | 0.07 (0.19) | 0.8 (0.14) | 0.39 (0.17) |
| IDL-FC | 0.23 (0.2) | 0.78 (0.12) | −0.05 (0.17) | 0.61 (0.19) | 0.44 (0.15) |
| IDL-L | 0.38 (0.23) | 0.65 (0.18) | 0.05 (0.2) | 0.64 (0.2) | 0.47 (0.17) |
| IDL-P | 0.31 (0.2) | 0.66 (0.14) | −0.04 (0.17) | 0.82 (0.13) | 0.49 (0.14) |
| IDL-PL | 0.25 (0.23) | 0.83 (0.1) | −0.12 (0.19) | 0.7 (0.19) | 0.64 (0.15) |
| IDL-TG | 0.22 (0.18) | 0.82 (0.08) | −0.2 (0.13) | 0.56 (0.15) | 0.67 (0.08) |
| S-LDL-C | 0.11 (0.28) | 0.66 (0.18) | −0.16 (0.22) | 0.44 (0.34) | 0.58 (0.14) |
| S-LDL-L | 0.26 (0.23) | 0.66 (0.17) | −0.06 (0.21) | 0.62 (0.21) | 0.58 (0.13) |
| S-LDL-P | 0.34 (0.2) | 0.68 (0.15) | −0.02 (0.19) | 0.63 (0.18) | 0.58 (0.13) |
| M-LDL-C | 0.15 (0.26) | 0.63 (0.18) | 0.22 (0.22) | 0.87 (0.08) | 0.13 (0.23) |
| M-LDL-CE | 0.3 (0.23) | 0.61 (0.2) | 0.05 (0.21) | 0.65 (0.2) | 0.45 (0.16) |
| M-LDL-L | 0.29 (0.22) | 0.63 (0.18) | 0.01 (0.21) | 0.66 (0.19) | 0.5 (0.15) |
| M-LDL-P | 0.29 (0.23) | 0.63 (0.18) | −0.01 (0.21) | 0.65 (0.21) | 0.51 (0.15) |
| M-LDL-PL | 0.2 (0.24) | 0.69 (0.16) | 0.11 (0.2) | 0.89 (0.06) | 0.18 (0.22) |
| L-LDL-C | 0.25 (0.24) | 0.58 (0.21) | 0.25 (0.22) | 0.68 (0.19) | 0.23 (0.21) |
| L-LDL-CE | 0.3 (0.23) | 0.58 (0.22) | 0.05 (0.21) | 0.65 (0.21) | 0.41 (0.17) |
| L-LDL-FC | 0.31 (0.24) | 0.57 (0.22) | 0.33 (0.23) | 0.7 (0.18) | 0.13 (0.23) |
| L-LDL-L | 0.31 (0.23) | 0.61 (0.2) | 0.04 (0.21) | 0.65 (0.21) | 0.44 (0.17) |
| L-LDL-P | 0.31 (0.23) | 0.63 (0.19) | 0.02 (0.21) | 0.65 (0.21) | 0.47 (0.16) |
| L-LDL-PL | 0.27 (0.25) | 0.61 (0.2) | 0.24 (0.22) | 0.67 (0.2) | 0.27 (0.2) |
| LDL-D | −0.33 (0.25) | −0.22 (0.23) | −0.15 (0.21) | −0.15 (0.29) | −0.37 (0.16) |
| XS-VLDL-L | 0.25 (0.23) | 0.8 (0.08) | −0.2 (0.17) | 0.61 (0.14) | 0.73 (0.09) |
| XS-VLDL-P | 0.17 (0.18) | 0.83 (0.07) | −0.26 (0.13) | 0.57 (0.13) | 0.71 (0.07) |
| XS-VLDL-PL | 0.21 (0.19) | 0.78 (0.09) | −0.15 (0.15) | 0.74 (0.14) | 0.57 (0.11) |
| XS-VLDL-TG | 0.06 (0.18) | 0.83 (0.08) | −0.37 (0.11) | 0.56 (0.13) | 0.85 (0.04) |
| S-VLDL-FC | −0.08 (0.2) | 0.94 (0.05) | −0.49 (0.12) | 0.59 (0.12) | 0.92 (0.03) |
| S-VLDL-L | −0.12 (0.24) | 0.7 (0.08) | −0.46 (0.15) | 0.5 (0.14) | 0.8 (0.05) |
| S-VLDL-P | −0.09 (0.19) | 0.78 (0.07) | −0.48 (0.11) | 0.5 (0.14) | 0.95 (0.02) |
| S-VLDL-PL | −0.03 (0.2) | 0.82 (0.08) | −0.43 (0.12) | 0.44 (0.17) | 0.92 (0.03) |
| S-VLDL-TG | −0.1 (0.2) | 0.9 (0.08) | −0.49 (0.11) | 0.49 (0.15) | 0.98 (0.01) |
| S-VLDL-C | 0.01 (0.2) | 0.9 (0.06) | −0.39 (0.13) | 0.61 (0.15) | 0.89 (0.05) |
| M-VLDL-C | −0.01 (0.2) | 0.8 (0.09) | −0.47 (0.12) | 0.41 (0.18) | 0.95 (0.02) |
| M-VLDL-CE | 0.01 (0.19) | 0.78 (0.08) | −0.43 (0.12) | 0.5 (0.15) | 0.9 (0.03) |

*Continued on next page*

*Appendix 2—table 1 continued*

| Trait | ApoA1 | ApoB | HDL-C | LDL-C | TG |
|---|---|---|---|---|---|
| M-VLDL-FC | 0 (0.21) | 0.83 (0.09) | −0.48 (0.12) | 0.4 (0.18) | 0.97 (0.01) |
| M-VLDL-L | −0.1 (0.24) | 0.66 (0.11) | −0.48 (0.15) | 0.4 (0.18) | 0.8 (0.05) |
| M-VLDL-P | −0.06 (0.19) | 0.78 (0.1) | −0.46 (0.12) | 0.43 (0.16) | 0.98 (0.02) |
| M-VLDL-PL | 0.03 (0.21) | 0.85 (0.09) | −0.48 (0.12) | 0.4 (0.18) | 0.98 (0.01) |
| M-VLDL-TG | −0.02 (0.21) | 0.82 (0.11) | −0.5 (0.13) | 0.33 (0.19) | 0.98 (0.02) |
| L-VLDL-C | −0.05 (0.2) | 0.83 (0.12) | −0.55 (0.12) | 0.36 (0.19) | 1 (0.02) |
| L-VLDL-CE | 0 (0.19) | 0.78 (0.12) | −0.44 (0.12) | 0.43 (0.19) | 0.93 (0.03) |
| L-VLDL-FC | −0.03 (0.2) | 0.84 (0.12) | −0.53 (0.13) | 0.36 (0.19) | 1 (0.02) |
| L-VLDL-L | −0.06 (0.24) | 0.66 (0.14) | −0.47 (0.16) | 0.36 (0.2) | 0.86 (0.05) |
| L-VLDL-P | −0.02 (0.21) | 0.72 (0.12) | −0.44 (0.13) | 0.33 (0.18) | 0.98 (0.02) |
| L-VLDL-PL | 0.01 (0.21) | 0.86 (0.12) | −0.53 (0.13) | 0.3 (0.2) | 1.04 (0.03) |
| L-VLDL-TG | −0.06 (0.21) | 0.78 (0.12) | −0.54 (0.13) | 0.26 (0.19) | 1 (0.02) |
| XL-VLDL-L | −0.08 (0.24) | 0.7 (0.15) | −0.52 (0.16) | 0.43 (0.2) | 0.85 (0.05) |
| XL-VLDL-P | −0.06 (0.2) | 0.76 (0.12) | −0.48 (0.13) | 0.44 (0.18) | 0.95 (0.03) |
| XL-VLDL-PL | −0.09 (0.23) | 0.82 (0.13) | −0.62 (0.15) | 0.32 (0.21) | 1.06 (0.04) |
| XL-VLDL-TG | −0.14 (0.21) | 0.86 (0.13) | −0.65 (0.13) | 0.34 (0.19) | 1.03 (0.04) |
| XXL-VLDL-L | −0.07 (0.25) | 0.65 (0.16) | −0.5 (0.17) | 0.38 (0.22) | 0.83 (0.06) |
| XXL-VLDL-P | 0.17 (0.2) | 0.72 (0.15) | −0.3 (0.15) | 0.39 (0.21) | 0.86 (0.07) |
| XXL-VLDL-PL | −0.3 (0.24) | 0.66 (0.17) | −0.8 (0.16) | 0.22 (0.21) | 1.06 (0.06) |
| XXL-VLDL-TG | −0.21 (0.25) | 0.64 (0.16) | −0.7 (0.15) | 0.22 (0.22) | 1.08 (0.05) |
| VLDL-D | −0.22 (0.2) | 0.55 (0.14) | −0.53 (0.12) | 0.12 (0.19) | 0.86 (0.04) |

## Appendix 3

## Mendelian randomization

We implemented several Mendelian randomization (MR) designs and statistical methods to estimate the causal effect of lipoprotein subfractions and particles sizes on coronary artery disease. In general, we adopted the three-sample summary data MR design described in *Zhao et al., 2019b*, *Wang et al., 2020* and we swapped the roles of the GWAS datasets whenever permitted by the statistical methods. More specifically, the statistical methods we used for univariable MR (RAPS, IVW, weighted median) require that the GWAS datasets for obtaining instruments, SNP effects on the exposure, and SNP effects on the outcome must have no overlapping sample. The multivariable MR method we used (GRAPPLE) allows the exposure and outcome GWAS to be dependent and estimates the proportion of overlapping sample. However, GRAPPLE still requires that the selection GWAS uses an non-overlapping sample.

The MR designs we implemented in this study are summarized in *Appendix 3—table 1*. We considered two ways of instrument selection for univariable MR. In 'traditional selection', the traditional lipid traits were used to select the instruments for the corresponding subfraction traits. That is, HDL-C was used to select SNPs for HDL subfractions and particle size, LDL-C for IDL and LDL subfractions and particle size, and TG for VLDL subfractions and particle size. This tends to select more instruments because the GWAS for traditional lipid traits had a larger sample size. In 'subfraction selection', the instrumental SNPs were selected for each lipoprotein subfraction and particle size using the same or closest trait in the selection GWAS. For example, if the exposure under investigation is S-HDL-L but it is not measured in the Davis GWAS (if it is used for selection), S-HDL-P is used instead for instrument selection.

For multivariable MR, we considered two models with different sets of exposures: TG, LDL-C, HDL-C, and the subfraction/particle size under investigation; TG, ApoB, ApoA1, and the subfraction/particle size under investigation. SNPs were selected as potential instruments if they were associated (p-value $\leq 10^{-4}$) with at least one of the four exposures. LD clumping was then used to obtain independent instruments, as described in Materials and Methods.

We briefly comment on the statistical methods used in univariable MR. All the three methods we used—RAPS, IVW, weighted median—require that the exposure GWAS and outcome GWAS have non-overlapping samples. RAPS and weighted median can provide consistent estimate of the causal effect even when some of the genetic variants are not valid instruments, provided that the direct effects of the genetic variants are independent of the strength of their associations with the exposure. The last condition is called the Instrument Strength Independent of Direct Effect (InSIDE) assumption in the MR literature [bowden2015mendelian]. RAPS is also robust to idiosyncratically large direct effect (*Bowden et al., 2015*). Because IVW and weighted median can be severely biased by weak instruments (*Zhao et al., 2020*), we only used them with the set of SNPs that have genome-wide significant association (p-value $\leq 5 \times 10^{-8}$) with the exposure. In comparison, RAPS does not suffer from weak instrument bias and we used it with all the SNPs obtained by LD clumping without any p-value threshold.

Below, *Appendix 3—figure 1* shows the MR results for the 27 lipoprotein measurements selected in phenotypic screening. Estimates that are statistically significant at a false discovery rate of 0.05 are shown in *Figure 2* of the main paper. *Appendix 3—table 2* shows the estimated effect of all the lipoprotein subfractions and particle sizes on myocardial infarction or coronary artery disease in various MR designs. Full results of the multivariable MR analyses, including the estimated effects of the traditional lipid risk factors, can be found in *Appendix 3—tables 5* and *6*. The results of the univariable MR analyses using IVW and weighted median estimators can be found in *Appendix 3—tables 3* and *4*.

**Appendix 3—table 1.** Three-sample Mendelian randomization designs.

| MR design | Selection | Exposure | Outcome | Reported in |
|-----------|-----------|----------|---------|-------------|

*Continued on next page*

*Appendix 3—table 1 continued*

| MR design | Selection | Exposure | Outcome | Reported in |
|---|---|---|---|---|
| Univariable (traditional selection) | GERA | Davis | CARDIoGRAMplusC4D | *Appendix 3—table 2–4* |
| | GERA | Davis | UK Biobank | *Appendix 3—table 2–4* |
| | GERA | Kettunen | UK Biobank | *Appendix 3—table 2–4* |
| | GLGC | Davis | UK Biobank | *Appendix 3—table 2–4* |
| Univariable (subfraction selection) | Davis | Kettunen | UK Biobank | *Figure 2*; *Appendix 3—figure 1* and *Appendix 3—table 2–4* |
| | Kettunen | Davis | UK Biobank | *Appendix 3—figure 1* and *Appendix 3—table 2–4* |
| Multivariable | Davis, GERA | Kettunen, GLGC | CARDIoGRAMplusC4D + UK Biobank | *Figure 2*, *Table 2*; *Appendix 3—figure 1* and *Appendix 3—table 2–4* |

## Pooled results

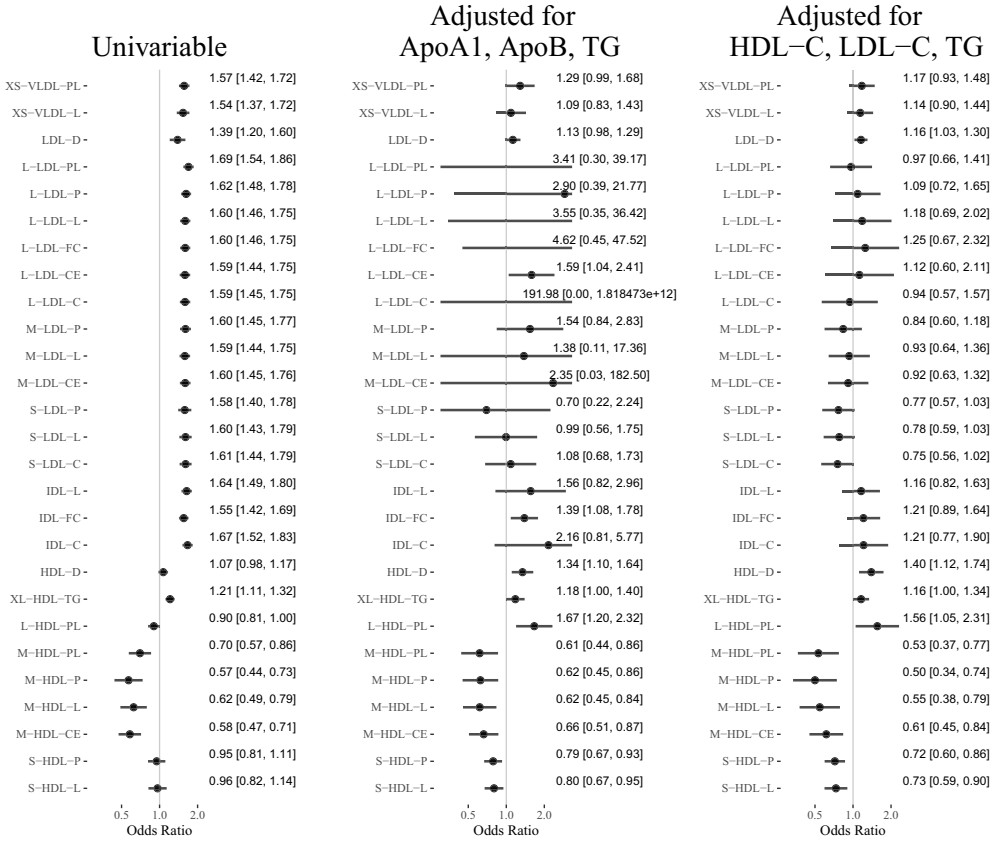

**Appendix 3—figure 1.** Mendelian randomization results for the 27 lipoprotein measurements selected in phenotypic screening.

In the tables below, Red indicates p-value is significant (at level 0.05) after Bonferroni correction for all the results in the corresponding table and blue indicates p-value ≤ 0.05.

**Appendix 3—table 2.** Mendelian randomization results using all selected SNPs (univariable MR using RAPS and multivariable MR using GRAPPLE).

| | Method: RAPS/GRAPPLE + All SNPs | | | | | | | |
|---|---|---|---|---|---|---|---|---|
| Screening | GERA | GERA | GERA | GLGC | Davis | Kettunen | GERA + Davis | GERA + Davis |
| Exposure | Davis | Davis | Kettunen | Davis | Kettunen | Davis | GLGC + Kettunen | GLGC + Kettunen |
| Outcome | CAD | UKB | UKB | UKB | UKB | UKB | CAD + UKB | CAD + UKB |
| Adjusted | | | | | | | HDL-C + LDL-C + TG | ApoA1 + ApoB + TG |
| VLDL traits | | | | | | | | |
| TG | 0.258 (0.053) | 0.296 (0.075) | NA | 0.262 (0.06) | NA | 0.289 (0.068) | NA | NA |
| VLDL-D | -0.099 (0.049) | 0.028 (0.074) | 0.072 (0.073) | 0.116 (0.065) | -0.163 (0.067) | -0.204 (0.071) | -0.588 (0.094) | -0.32 (0.112) |
| XS-VLDL-L | NA | NA | 0.368 (0.064) | NA | 0.429 (0.059) | NA | 0.132 (0.119) | 0.084 (0.141) |
| XS-VLDL-P | 0.17 (0.031) | 0.26 (0.048) | 0.367 (0.065) | 0.248 (0.047) | 0.429 (0.06) | 0.338 (0.056) | 0.118 (0.125) | 0.061 (0.158) |
| XS-VLDL-PL | 0.191 (0.034) | 0.284 (0.055) | 0.386 (0.069) | 0.278 (0.052) | 0.449 (0.049) | 0.435 (0.049) | 0.159 (0.12) | 0.253 (0.135) |
| XS-VLDL-TG | 0.201 (0.034) | 0.3 (0.053) | 0.388 (0.068) | 0.283 (0.046) | 0.372 (0.063) | 0.326 (0.055) | -0.157 (0.187) | -0.248 (0.15) |
| S-VLDL-C | 0.294 (0.06) | 0.343 (0.076) | NA | 0.322 (0.063) | NA | 0.424 (0.094) | -1.035 (0.323) | -1.265 (0.568) |
| S-VLDL-FC | 0.243 (0.051) | 0.303 (0.068) | 0.389 (0.079) | 0.286 (0.056) | 0.489 (0.071) | 0.416 (0.074) | -1.027 (0.337) | -0.489 (0.213) |
| S-VLDL-L | NA | NA | 0.356 (0.075) | NA | 0.376 (0.072) | NA | -0.898 (0.28) | -1.629 (0.586) |
| S-VLDL-P | 0.226 (0.047) | 0.288 (0.068) | 0.343 (0.074) | 0.261 (0.054) | 0.359 (0.069) | 0.271 (0.094) | -1.245 (0.463) | -1.644 (0.606) |
| S-VLDL-PL | 0.228 (0.047) | 0.294 (0.067) | 0.372 (0.074) | 0.273 (0.054) | 0.365 (0.066) | 0.336 (0.063) | -0.613 (0.182) | -1.213 (0.478) |
| S-VLDL-TG | 0.223 (0.049) | 0.283 (0.071) | 0.323 (0.073) | 0.25 (0.055) | 0.327 (0.071) | 0.275 (0.067) | NaN | -0.301 (0.108) |
| M-VLDL-C | 0.253 (0.053) | 0.304 (0.078) | 0.327 (0.074) | 0.276 (0.06) | 0.368 (0.07) | 0.312 (0.079) | -1.433 (0.451) | -0.373 (0.118) |
| M-VLDL-CE | 0.248 (0.051) | 0.309 (0.074) | 0.344 (0.077) | 0.285 (0.058) | 0.369 (0.073) | 0.295 (0.069) | -1.035 (0.293) | -0.995 (0.338) |
| M-VLDL-FC | 0.245 (0.058) | 0.283 (0.082) | 0.31 (0.076) | 0.259 (0.063) | 0.341 (0.069) | 0.341 (0.068) | -1.412 (0.444) | -0.799 (0.311) |
| M-VLDL-L | NA | NA | 0.311 (0.079) | NA | 0.358 (0.078) | NA | -1.878 (0.75) | -0.298 (0.098) |
| M-VLDL-P | 0.25 (0.062) | 0.282 (0.083) | 0.305 (0.081) | 0.247 (0.065) | 0.293 (0.089) | 0.269 (0.065) | -1.974 (0.745) | -0.312 (0.096) |
| M-VLDL-PL | 0.248 (0.056) | 0.295 (0.077) | 0.318 (0.075) | 0.259 (0.06) | 0.351 (0.071) | 0.31 (0.063) | -2.012 (0.943) | -0.297 (0.106) |
| M-VLDL-TG | 0.205 (0.064) | 0.248 (0.087) | 0.3 (0.082) | 0.224 (0.067) | 0.275 (0.092) | 0.246 (0.074) | -2.133 (0.879) | -0.806 (0.455) |
| L-VLDL-C | 0.299 (0.067) | 0.304 (0.1) | 0.297 (0.081) | 0.291 (0.077) | 0.289 (0.085) | 0.317 (0.077) | -1.254 (0.297) | -0.609 (0.337) |
| L-VLDL-CE | 0.247 (0.061) | 0.282 (0.088) | 0.282 (0.082) | 0.282 (0.072) | 0.285 (0.082) | 0.3 (0.112) | -1.081 (0.282) | -0.673 (0.217) |

*Continued on next page*

*Appendix 3—table 2 continued*

| | Method: RAPS/GRAPPLE + All SNPs | | | | | | | |
|---|---|---|---|---|---|---|---|---|
| Screening | GERA | GERA | GERA | GLGC | Davis | Kettunen | GERA + Davis GLGC + Kettunen | GERA + Davis GLGC + Kettunen |
| Exposure | Davis | Davis | Kettunen | Davis | Kettunen | Davis | | |
| Outcome | CAD | UKB | UKB | UKB | UKB | UKB | CAD + UKB | CAD + UKB |
| Adjusted | | | | | | | HDL-C + LDL-C + TG | ApoA1 + ApoB + TG |
| L-VLDL-FC | 0.316 (0.076) | 0.294 (0.108) | 0.311 (0.083) | 0.287 (0.081) | 0.351 (0.087) | 0.298 (0.078) | -1.274 (0.308) | -0.619 (0.291) |
| L-VLDL-L | NA | NA | 0.36 (0.096) | NA | 0.32 (0.102) | NA | -1.277 (0.313) | -0.532 (0.278) |
| L-VLDL-P | 0.268 (0.073) | 0.287 (0.103) | 0.281 (0.085) | 0.262 (0.075) | 0.219 (0.086) | 0.255 (0.082) | -1.357 (0.344) | -0.617 (0.229) |
| L-VLDL-PL | 0.322 (0.071) | 0.318 (0.102) | 0.346 (0.089) | 0.283 (0.077) | 0.397 (0.101) | 0.351 (0.076) | NaN | -0.287 (0.104) |
| L-VLDL-TG | 0.243 (0.077) | 0.238 (0.104) | 0.332 (0.094) | 0.246 (0.08) | 0.26 (0.103) | 0.324 (0.082) | -1.428 (0.372) | -0.252 (0.091) |
| XL-VLDL-L | NA | NA | 0.289 (0.098) | NA | 0.435 (0.14) | NA | -1.069 (0.203) | -0.577 (0.249) |
| XL-VLDL-P | 0.27 (0.074) | 0.262 (0.099) | 0.281 (0.093) | 0.279 (0.084) | 0.404 (0.122) | 0.251 (0.084) | -1.209 (0.238) | -0.373 (0.109) |
| XL-VLDL-PL | 0.446 (0.09) | 0.344 (0.13) | 0.31 (0.093) | 0.361 (0.118) | 0.375 (0.12) | 0.408 (0.102) | -1.214 (0.257) | -0.583 (0.268) |
| XL-VLDL-TG | 0.294 (0.092) | 0.229 (0.109) | 0.261 (0.094) | 0.284 (0.095) | 0.365 (0.111) | 0.319 (0.093) | -1.071 (0.205) | -0.603 (0.248) |
| XXL-VLDL-L | NA | NA | 0.397 (0.108) | NA | 0.312 (0.108) | NA | -1.355 (0.318) | -0.402 (0.144) |
| XXL-VLDL-P | 0.308 (0.08) | 0.327 (0.096) | 0.378 (0.097) | 0.297 (0.088) | 0.32 (0.101) | 0.227 (0.073) | -1.639 (0.502) | -1.089 (0.449) |
| XXL-VLDL-PL | 0.338 (0.091) | 0.346 (0.103) | 0.342 (0.103) | 0.351 (0.103) | 0.282 (0.114) | 0.317 (0.086) | -1.259 (0.262) | -0.814 (0.344) |
| XXL-VLDL-TG | 0.384 (0.108) | 0.374 (0.124) | 0.348 (0.1) | 0.433 (0.121) | 0.304 (0.138) | 0.359 (0.18) | -1.202 (0.262) | -1.075 (0.402) |
| IDL/LDL traits | | | | | | | | |
| LDL-C | 0.523 (0.043) | 0.512 (0.053) | 0.514 (0.042) | 0.473 (0.055) | 0.435 (0.048) | 0.464 (0.048) | NA | 0.319 (0.182) |
| ApoB | 0.605 (0.056) | 0.55 (0.062) | 0.551 (0.052) | 0.543 (0.069) | 0.61 (0.066) | 0.613 (0.06) | -0.532 (0.191) | NA |
| LDL-D | 0.271 (0.215) | 0.452 (0.299) | 2.064 (0.233) | 0.831 (0.684) | 0.328 (0.073) | 0.201 (0.055) | 0.145 (0.061) | 0.119 (0.071) |
| S-LDL-C | 0.624 (0.053) | 0.589 (0.061) | 0.539 (0.048) | 0.537 (0.067) | 0.474 (0.056) | 0.48 (0.05) | -0.282 (0.152) | 0.08 (0.238) |
| S-LDL-L | NA | NA | 0.561 (0.047) | NA | 0.473 (0.057) | NA | -0.251 (0.145) | -0.005 (0.29) |
| S-LDL-P | 0.621 (0.057) | 0.581 (0.065) | 0.56 (0.049) | 0.558 (0.073) | 0.459 (0.061) | 0.546 (0.063) | -0.266 (0.151) | -0.362 (0.596) |
| M-LDL-C | 0.648 (0.055) | 0.607 (0.062) | 0.545 (0.044) | 0.545 (0.068) | 0.455 (0.049) | 0.557 (0.054) | -0.271 (0.162) | -0.169 (0.909) |
| M-LDL-CE | 0.643 (0.056) | 0.601 (0.062) | 0.564 (0.042) | 0.545 (0.069) | 0.467 (0.05) | 0.55 (0.055) | -0.088 (0.188) | NaN |
| M-LDL-L | NA | NA | 0.559 (0.042) | NA | 0.461 (0.049) | NA | -0.069 (0.191) | NaN |

*Continued on next page*

Appendix 3—table 2 continued

| | Method: RAPS/GRAPPLE + All SNPs | | | | | | | |
|---|---|---|---|---|---|---|---|---|
| Screening | GERA | GERA | GERA | GLGC | Davis | Kettunen | GERA + Davis GLGC + Kettunen | GERA + Davis GLGC + Kettunen |
| Exposure | Davis | Davis | Kettunen | Davis | Kettunen | Davis | | |
| Outcome | CAD | UKB | UKB | UKB | UKB | UKB | CAD + UKB | CAD + UKB |
| Adjusted | | | | | | | HDL-C + LDL-C + TG | ApoA1 + ApoB + TG |
| M-LDL-P | 0.638 (0.056) | 0.597 (0.062) | 0.557 (0.043) | 0.54 (0.069) | 0.472 (0.051) | 0.46 (0.05) | -0.179 (0.174) | 0.432 (0.31) |
| M-LDL-PL | 0.658 (0.063) | 0.605 (0.067) | 0.556 (0.047) | 0.571 (0.077) | 0.506 (0.053) | 0.559 (0.057) | -0.407 (0.162) | -0.566 (0.839) |
| L-LDL-C | 0.627 (0.053) | 0.577 (0.059) | 0.515 (0.042) | 0.504 (0.063) | 0.465 (0.048) | 0.488 (0.052) | -0.059 (0.261) | NaN |
| L-LDL-CE | 0.638 (0.055) | 0.589 (0.06) | 0.555 (0.041) | 0.514 (0.065) | 0.463 (0.049) | 0.493 (0.054) | 0.116 (0.321) | 0.461 (0.213) |
| L-LDL-FC | 0.609 (0.051) | 0.557 (0.057) | 0.503 (0.041) | 0.491 (0.06) | 0.468 (0.047) | 0.457 (0.052) | 0.223 (0.315) | NaN |
| L-LDL-L | NA | NA | 0.543 (0.04) | NA | 0.468 (0.047) | NA | 0.167 (0.273) | NaN |
| L-LDL-P | 0.606 (0.052) | 0.559 (0.058) | 0.545 (0.041) | 0.49 (0.062) | 0.484 (0.046) | 0.494 (0.048) | 0.084 (0.213) | NaN |
| L-LDL-PL | 0.61 (0.053) | 0.558 (0.058) | 0.515 (0.043) | 0.492 (0.063) | 0.528 (0.048) | 0.502 (0.052) | -0.036 (0.195) | NaN |
| IDL-C | 0.596 (0.054) | 0.55 (0.059) | 0.562 (0.042) | 0.481 (0.064) | 0.511 (0.047) | 0.423 (0.051) | 0.192 (0.229) | 0.769 (0.501) |
| IDL-FC | 0.586 (0.054) | 0.539 (0.059) | 0.525 (0.044) | 0.494 (0.063) | 0.44 (0.044) | 0.402 (0.05) | 0.19 (0.156) | 0.33 (0.127) |
| IDL-L | NA | NA | 0.57 (0.043) | NA | 0.494 (0.048) | NA | 0.148 (0.175) | 0.444 (0.328) |
| IDL-P | 0.566 (0.052) | 0.536 (0.059) | 0.575 (0.044) | 0.488 (0.065) | 0.434 (0.049) | 0.412 (0.051) | 0.153 (0.148) | 0.292 (0.173) |
| IDL-PL | 0.583 (0.052) | 0.533 (0.058) | 0.532 (0.045) | 0.489 (0.064) | 0.471 (0.047) | 0.396 (0.05) | 0.153 (0.18) | 0.406 (0.184) |
| IDL-TG | 0.603 (0.066) | 0.595 (0.075) | 0.658 (0.063) | 0.567 (0.085) | 0.432 (0.056) | 0.315 (0.053) | 0.11 (0.103) | 0.047 (0.135) |
| HDL traits | | | | | | | | |
| HDL-C | -0.117 (0.031) | -0.199 (0.045) | -0.136 (0.055) | -0.317 (0.052) | -0.045 (0.059) | -0.108 (0.05) | NA | NaN |
| ApoA1 | -0.119 (0.042) | -0.193 (0.06) | 0.023 (0.058) | -0.264 (0.071) | 0.075 (0.064) | -0.13 (0.068) | -0.481 (0.271) | NA |
| HDL-D | -0.008 (0.027) | -0.124 (0.041) | 0.004 (0.046) | -0.092 (0.048) | 0.067 (0.045) | 0.007 (0.041) | 0.333 (0.114) | 0.296 (0.1) |
| S-HDL-L | NA | NA | -0.098 (0.095) | NA | -0.037 (0.085) | NA | -0.312 (0.106) | -0.224 (0.087) |
| S-HDL-P | -0.265 (0.084) | -0.362 (0.113) | -0.13 (0.092) | -0.317 (0.119) | -0.053 (0.081) | -0.08 (0.094) | -0.331 (0.095) | -0.24 (0.083) |
| S-HDL-TG | 0.354 (0.072) | 0.386 (0.088) | 0.65 (0.089) | 0.475 (0.097) | 0.351 (0.087) | 0.283 (0.073) | 0.253 (0.637) | -0.044 (0.466) |
| M-HDL-C | -0.323 (0.058) | -0.43 (0.079) | -0.364 (0.085) | -0.376 (0.091) | -0.46 (0.104) | -0.434 (0.075) | -0.508 (0.165) | -0.442 (0.143) |
| M-HDL-CE | -0.333 (0.058) | -0.458 (0.078) | -0.372 (0.09) | -0.385 (0.087) | -0.542 (0.105) | -0.443 (0.071) | -0.487 (0.157) | -0.413 (0.137) |

*Continued on next page*

Appendix 3—table 2 continued

| | Method: RAPS/GRAPPLE + All SNPs | | | | | | | |
|---|---|---|---|---|---|---|---|---|
| Screening | GERA | GERA | GERA | GLGC | Davis | Kettunen | GERA + Davis GLGC + Kettunen | GERA + Davis GLGC + Kettunen |
| Exposure | Davis | Davis | Kettunen | Davis | Kettunen | Davis | | |
| Outcome | CAD | UKB | UKB | UKB | UKB | UKB | CAD + UKB | CAD + UKB |
| Adjusted | | | | | | | HDL-C + LDL-C + TG | ApoA1 + ApoB + TG |
| M-HDL-FC | -0.275 (0.065) | -0.319 (0.08) | -0.262 (0.083) | -0.313 (0.092) | -0.313 (0.094) | -0.409 (0.082) | -0.649 (0.225) | -0.408 (0.166) |
| M-HDL-L | NA | NA | -0.311 (0.095) | NA | -0.474 (0.123) | NA | -0.606 (0.188) | -0.485 (0.155) |
| M-HDL-P | -0.298 (0.06) | -0.394 (0.086) | -0.273 (0.101) | -0.373 (0.1) | -0.565 (0.131) | -0.307 (0.079) | -0.694 (0.204) | -0.472 (0.166) |
| M-HDL-PL | -0.265 (0.058) | -0.346 (0.083) | -0.25 (0.09) | -0.335 (0.096) | -0.358 (0.104) | -0.3 (0.072) | -0.632 (0.191) | -0.486 (0.171) |
| L-HDL-C | -0.067 (0.03) | -0.144 (0.044) | -0.139 (0.051) | -0.144 (0.05) | -0.147 (0.052) | -0.049 (0.045) | 0.516 (0.213) | 0.575 (0.204) |
| L-HDL-CE | -0.063 (0.03) | -0.144 (0.044) | -0.116 (0.051) | -0.149 (0.051) | -0.134 (0.051) | -0.094 (0.047) | 0.519 (0.23) | 0.61 (0.206) |
| L-HDL-FC | -0.082 (0.03) | -0.144 (0.045) | -0.114 (0.053) | -0.128 (0.053) | -0.13 (0.051) | -0.03 (0.047) | 0.518 (0.181) | 0.59 (0.148) |
| L-HDL-L | NA | NA | -0.108 (0.05) | NA | -0.132 (0.052) | NA | 0.457 (0.189) | 0.541 (0.184) |
| L-HDL-P | -0.071 (0.028) | -0.146 (0.042) | -0.111 (0.05) | -0.13 (0.049) | -0.083 (0.05) | -0.1 (0.043) | 0.422 (0.191) | 0.476 (0.155) |
| L-HDL-PL | -0.087 (0.029) | -0.161 (0.043) | -0.141 (0.051) | -0.142 (0.051) | -0.105 (0.053) | -0.092 (0.044) | 0.443 (0.202) | 0.51 (0.169) |
| XL-HDL-C | 0.055 (0.046) | -0.013 (0.068) | 0.11 (0.066) | 0.064 (0.073) | 0.048 (0.069) | 0.112 (0.068) | 0.474 (0.223) | 0.565 (0.196) |
| XL-HDL-CE | 0.064 (0.044) | 0.006 (0.066) | 0.129 (0.066) | 0.08 (0.07) | 0.057 (0.068) | 0.046 (0.075) | 0.426 (0.177) | 0.511 (0.206) |
| XL-HDL-FC | 0.009 (0.039) | -0.05 (0.059) | 0.066 (0.058) | -0.026 (0.067) | 0.102 (0.06) | 0.049 (0.066) | 0.433 (0.16) | 0.609 (0.159) |
| XL-HDL-L | NA | NA | 0.073 (0.055) | NA | 0.038 (0.058) | NA | 0.358 (0.154) | 0.481 (0.141) |
| XL-HDL-P | 0.038 (0.033) | -0.022 (0.049) | 0.112 (0.057) | 0.017 (0.056) | 0.083 (0.055) | 0.023 (0.057) | 0.41 (0.139) | 0.39 (0.135) |
| XL-HDL-PL | 0.029 (0.031) | -0.031 (0.046) | 0.037 (0.05) | 0.005 (0.055) | 0.038 (0.052) | 0.013 (0.046) | 0.343 (0.118) | 0.466 (0.12) |
| XL-HDL-TG | 0.092 (0.027) | 0.112 (0.041) | 0.14 (0.047) | 0.135 (0.047) | 0.191 (0.042) | 0.136 (0.039) | 0.147 (0.074) | 0.165 (0.086) |

## Univariable MR results

**Appendix 3—table 3.** Mendelian randomization results using genome-wide significant SNPs and inverse variance weighted (IVW) estimator.

| | Method: IVW + Significant SNPs | | | | | |
|---|---|---|---|---|---|---|
| Selection | GERA | GERA | GERA | GLGC | Davis | Kettunen |
| Exposure | Davis | Davis | Kettunen | Davis | Kettunen | Davis |
| Outcome | CAD | UKB | UKB | UKB | UKB | UKB |

*Continued on next page*

*Appendix 3—table 3 continued*

| | Method: IVW + Significant SNPs | | | | | |
|---|---|---|---|---|---|---|
| Selection | GERA | GERA | GERA | GLGC | Davis | Kettunen |
| Exposure | Davis | Davis | Kettunen | Davis | Kettunen | Davis |
| Outcome | CAD | UKB | UKB | UKB | UKB | UKB |
| VLDL traits | | | | | | |
| TG | 0.184 (0.051) | 0.278 (0.076) | NA | 0.309 (0.074) | NA | 0.207 (0.064) |
| VLDL-D | 0.044 (0.06) | 0.052 (0.09) | 0.038 (0.102) | 0.118 (0.091) | -0.083 (0.16) | -0.083 (0.138) |
| XS-VLDL-L | NA | NA | 0.353 (0.08) | NA | 0.372 (0.083) | NA |
| XS-VLDL-P | 0.162 (0.04) | 0.256 (0.059) | 0.352 (0.081) | 0.273 (0.063) | 0.374 (0.084) | 0.373 (0.095) |
| XS-VLDL-PL | 0.165 (0.046) | 0.262 (0.069) | 0.37 (0.088) | 0.27 (0.075) | 0.443 (0.048) | 0.401 (0.07) |
| XS-VLDL-TG | 0.179 (0.041) | 0.277 (0.061) | 0.362 (0.082) | 0.288 (0.062) | 0.335 (0.076) | 0.314 (0.08) |
| S-VLDL-C | 0.237 (0.053) | 0.343 (0.08) | NA | 0.339 (0.083) | NA | 0.443 (0.116) |
| S-VLDL-FC | 0.21 (0.05) | 0.307 (0.076) | 0.344 (0.098) | 0.314 (0.076) | 0.262 (0.122) | 0.397 (0.116) |
| S-VLDL-L | NA | NA | 0.318 (0.095) | NA | 0.27 (0.106) | NA |
| S-VLDL-P | 0.188 (0.049) | 0.274 (0.074) | 0.311 (0.093) | 0.29 (0.072) | 0.266 (0.103) | 0.331 (0.142) |
| S-VLDL-PL | 0.198 (0.048) | 0.291 (0.072) | 0.342 (0.091) | 0.3 (0.072) | 0.281 (0.089) | 0.331 (0.125) |
| S-VLDL-TG | 0.174 (0.051) | 0.255 (0.076) | 0.296 (0.094) | 0.28 (0.073) | 0.261 (0.102) | 0.262 (0.093) |
| M-VLDL-C | 0.188 (0.053) | 0.265 (0.08) | 0.305 (0.096) | 0.287 (0.077) | 0.361 (0.078) | 0.32 (0.134) |
| M-VLDL-CE | 0.203 (0.051) | 0.285 (0.077) | 0.32 (0.098) | 0.295 (0.076) | 0.264 (0.094) | 0.291 (0.125) |
| M-VLDL-FC | 0.165 (0.056) | 0.233 (0.084) | 0.292 (0.098) | 0.27 (0.08) | 0.3 (0.084) | 0.303 (0.104) |
| M-VLDL-L | NA | NA | 0.265 (0.104) | NA | 0.357 (0.096) | NA |
| M-VLDL-P | 0.153 (0.056) | 0.214 (0.085) | 0.276 (0.104) | 0.258 (0.081) | 0.322 (0.092) | 0.268 (0.074) |
| M-VLDL-PL | 0.163 (0.054) | 0.23 (0.082) | 0.296 (0.097) | 0.266 (0.078) | 0.302 (0.084) | 0.289 (0.095) |
| M-VLDL-TG | 0.14 (0.058) | 0.196 (0.087) | 0.268 (0.107) | 0.247 (0.083) | 0.327 (0.093) | 0.245 (0.091) |
| L-VLDL-C | 0.177 (0.06) | 0.24 (0.091) | 0.288 (0.106) | 0.286 (0.089) | 0.108 (0.223) | 0.31 (0.084) |
| L-VLDL-CE | 0.178 (0.057) | 0.245 (0.087) | 0.262 (0.105) | 0.279 (0.086) | 0.182 (0.187) | 0.299 (0.077) |
| L-VLDL-FC | 0.176 (0.063) | 0.242 (0.094) | 0.295 (0.108) | 0.298 (0.091) | 0.321 (0.101) | 0.314 (0.082) |
| L-VLDL-L | NA | NA | 0.291 (0.119) | NA | 0.125 (0.232) | NA |
| L-VLDL-P | 0.164 (0.062) | 0.227 (0.093) | 0.269 (0.108) | 0.275 (0.09) | 0.332 (0.127) | 0.247 (0.076) |
| L-VLDL-PL | 0.173 (0.061) | 0.23 (0.092) | 0.308 (0.115) | 0.284 (0.088) | 0.32 (0.127) | 0.302 (0.079) |
| L-VLDL-TG | 0.149 (0.063) | 0.202 (0.095) | 0.268 (0.118) | 0.267 (0.092) | 0.33 (0.131) | 0.302 (0.08) |
| XL-VLDL-L | NA | NA | 0.263 (0.123) | NA | 0.365 (0.286) | NA |
| XL-VLDL-P | 0.149 (0.063) | 0.206 (0.095) | 0.247 (0.122) | 0.268 (0.096) | 0.346 (0.28) | 0.245 (0.077) |
| XL-VLDL-PL | 0.176 (0.067) | 0.243 (0.101) | 0.292 (0.119) | 0.323 (0.101) | 0.333 (0.265) | 0.344 (0.133) |
| XL-VLDL-TG | 0.151 (0.066) | 0.205 (0.1) | 0.241 (0.12) | 0.282 (0.1) | 0.323 (0.272) | 0.249 (0.081) |
| XXL-VLDL-L | NA | NA | 0.356 (0.127) | NA | -0.165 (0.425) | NA |
| XXL-VLDL-P | 0.228 (0.067) | 0.35 (0.099) | 0.372 (0.119) | 0.376 (0.098) | -0.12 (0.389) | 0.006 (0.153) |
| XXL-VLDL-PL | 0.211 (0.07) | 0.31 (0.105) | 0.275 (0.125) | 0.399 (0.107) | -0.145 (0.395) | 0.071 (0.191) |
| XXL-VLDL-TG | 0.221 (0.067) | 0.3 (0.102) | 0.292 (0.126) | 0.415 (0.104) | 0.09 (0.36) | 0.349 (0.303) |
| IDL/LDL traits | | | | | | |
| LDL-C | 0.427 (0.049) | 0.431 (0.054) | 0.409 (0.077) | 0.409 (0.054) | 0.416 (0.099) | 0.422 (0.063) |
| ApoB | 0.506 (0.058) | 0.525 (0.065) | 0.474 (0.093) | 0.473 (0.064) | 0.636 (0.092) | 0.569 (0.071) |
| LDL-D | 0.217 (0.151) | 0.423 (0.161) | 1.121 (0.178) | 0.271 (0.143) | 0.309 (0.126) | 0.211 (0.081) |
| S-LDL-C | 0.481 (0.056) | 0.467 (0.063) | 0.445 (0.087) | 0.438 (0.063) | 0.44 (0.128) | 0.436 (0.076) |

*Continued on next page*

*Appendix 3—table 3 continued*

| | Method: IVW + Significant SNPs | | | | | |
|---|---|---|---|---|---|---|
| **Selection** | **GERA** | **GERA** | **GERA** | **GLGC** | **Davis** | **Kettunen** |
| **Exposure** | **Davis** | **Davis** | **Kettunen** | **Davis** | **Kettunen** | **Davis** |
| **Outcome** | **CAD** | **UKB** | **UKB** | **UKB** | **UKB** | **UKB** |
| S-LDL-L | NA | NA | 0.44 (0.09) | NA | 0.456 (0.132) | NA |
| S-LDL-P | 0.501 (0.059) | 0.494 (0.068) | 0.449 (0.093) | 0.472 (0.067) | 0.49 (0.139) | 0.588 (0.097) |
| M-LDL-C | 0.475 (0.057) | 0.457 (0.064) | 0.426 (0.08) | 0.427 (0.064) | 0.418 (0.111) | 0.436 (0.087) |
| M-LDL-CE | 0.485 (0.058) | 0.47 (0.065) | 0.432 (0.078) | 0.436 (0.064) | 0.43 (0.107) | 0.444 (0.085) |
| M-LDL-L | NA | NA | 0.43 (0.08) | NA | 0.43 (0.11) | NA |
| M-LDL-P | 0.479 (0.057) | 0.465 (0.064) | 0.437 (0.081) | 0.44 (0.064) | 0.413 (0.122) | 0.439 (0.093) |
| M-LDL-PL | 0.5 (0.063) | 0.49 (0.071) | 0.437 (0.087) | 0.464 (0.07) | 0.443 (0.132) | 0.497 (0.099) |
| L-LDL-C | 0.449 (0.055) | 0.436 (0.061) | 0.432 (0.076) | 0.411 (0.061) | 0.409 (0.106) | 0.417 (0.076) |
| L-LDL-CE | 0.464 (0.056) | 0.451 (0.062) | 0.426 (0.075) | 0.422 (0.062) | 0.416 (0.102) | 0.433 (0.077) |
| L-LDL-FC | 0.425 (0.054) | 0.411 (0.059) | 0.424 (0.074) | 0.393 (0.059) | 0.387 (0.105) | 0.394 (0.078) |
| L-LDL-L | NA | NA | 0.427 (0.074) | NA | 0.407 (0.103) | NA |
| L-LDL-P | 0.448 (0.054) | 0.442 (0.06) | 0.435 (0.075) | 0.421 (0.059) | 0.413 (0.104) | 0.424 (0.075) |
| L-LDL-PL | 0.444 (0.056) | 0.438 (0.061) | 0.441 (0.078) | 0.423 (0.061) | 0.42 (0.109) | 0.429 (0.076) |
| IDL-C | 0.447 (0.055) | 0.455 (0.059) | 0.451 (0.075) | 0.433 (0.06) | 0.439 (0.085) | 0.422 (0.07) |
| IDL-FC | 0.429 (0.055) | 0.439 (0.059) | 0.468 (0.075) | 0.414 (0.059) | 0.431 (0.081) | 0.402 (0.074) |
| IDL-L | NA | NA | 0.467 (0.075) | NA | 0.445 (0.085) | NA |
| IDL-P | 0.443 (0.055) | 0.467 (0.06) | 0.48 (0.077) | 0.45 (0.059) | 0.446 (0.088) | 0.426 (0.071) |
| IDL-PL | 0.429 (0.055) | 0.443 (0.059) | 0.473 (0.078) | 0.427 (0.059) | 0.435 (0.092) | 0.407 (0.069) |
| IDL-TG | 0.461 (0.07) | 0.518 (0.076) | 0.625 (0.098) | 0.494 (0.073) | 0.342 (0.085) | 0.34 (0.123) |
| HDL traits | | | | | | |
| HDL-C | -0.085 (0.044) | -0.156 (0.057) | -0.146 (0.085) | -0.195 (0.06) | -0.082 (0.159) | -0.015 (0.109) |
| ApoA1 | -0.072 (0.054) | -0.155 (0.071) | -0.036 (0.09) | -0.194 (0.074) | 0.001 (0.192) | 0.066 (0.158) |
| HDL-D | -0.027 (0.042) | -0.071 (0.058) | -0.052 (0.073) | -0.092 (0.063) | 0.073 (0.098) | 0.074 (0.074) |
| S-HDL-L | NA | NA | -0.064 (0.148) | NA | -0.033 (0.092) | NA |
| S-HDL-P | -0.117 (0.087) | -0.172 (0.116) | -0.13 (0.146) | -0.298 (0.117) | -0.033 (0.09) | -0.115 (0.174) |
| S-HDL-TG | 0.224 (0.063) | 0.317 (0.082) | 0.496 (0.107) | 0.344 (0.085) | 0.334 (0.096) | 0.286 (0.17) |
| M-HDL-C | -0.214 (0.062) | -0.327 (0.078) | -0.48 (0.111) | -0.39 (0.079) | -0.423 (0.175) | -0.39 (0.159) |
| M-HDL-CE | -0.227 (0.062) | -0.338 (0.077) | -0.497 (0.111) | -0.4 (0.078) | -0.435 (0.194) | -0.341 (0.238) |
| M-HDL-FC | -0.158 (0.065) | -0.272 (0.084) | -0.341 (0.117) | -0.337 (0.085) | -0.288 (0.218) | -0.278 (0.144) |
| M-HDL-L | NA | NA | -0.436 (0.125) | NA | -0.514 (0.223) | NA |
| M-HDL-P | -0.172 (0.066) | -0.292 (0.087) | -0.414 (0.132) | -0.361 (0.089) | -0.386 (0.307) | -0.18 (0.118) |
| M-HDL-PL | -0.161 (0.064) | -0.275 (0.085) | -0.38 (0.126) | -0.345 (0.087) | -0.419 (0.301) | -0.2 (0.099) |
| L-HDL-C | -0.047 (0.044) | -0.097 (0.059) | -0.124 (0.08) | -0.133 (0.063) | 0.022 (0.106) | 0.021 (0.105) |
| L-HDL-CE | -0.049 (0.044) | -0.098 (0.059) | -0.12 (0.079) | -0.137 (0.063) | 0.023 (0.112) | 0.004 (0.106) |
| L-HDL-FC | -0.044 (0.046) | -0.094 (0.062) | -0.106 (0.082) | -0.127 (0.067) | 0.038 (0.103) | 0.017 (0.109) |
| L-HDL-L | NA | NA | -0.106 (0.077) | NA | 0.034 (0.102) | NA |
| L-HDL-P | -0.045 (0.043) | -0.097 (0.058) | -0.102 (0.077) | -0.125 (0.063) | 0.009 (0.111) | 0.025 (0.11) |
| L-HDL-PL | -0.054 (0.044) | -0.11 (0.06) | -0.115 (0.079) | -0.14 (0.064) | 0.006 (0.115) | 0.016 (0.115) |
| XL-HDL-C | 0.03 (0.06) | -0.012 (0.084) | 0.014 (0.099) | -0.05 (0.088) | -0.015 (0.165) | 0.161 (0.101) |
| XL-HDL-CE | 0.03 (0.059) | -0.009 (0.081) | 0.025 (0.098) | -0.042 (0.086) | -0.001 (0.166) | 0.221 (0.107) |

*Continued on next page*

*Appendix 3—table 3 continued*

| | Method: IVW + Significant SNPs | | | | | |
|---|---|---|---|---|---|---|
| Selection | GERA | GERA | GERA | GLGC | Davis | Kettunen |
| Exposure | Davis | Davis | Kettunen | Davis | Kettunen | Davis |
| Outcome | CAD | UKB | UKB | UKB | UKB | UKB |
| XL-HDL-FC | -0.003 (0.056) | -0.05 (0.076) | -0.001 (0.089) | -0.077 (0.081) | 0.072 (0.11) | 0.057 (0.092) |
| XL-HDL-L | NA | NA | 0.001 (0.085) | NA | -0.009 (0.138) | NA |
| XL-HDL-P | 0.015 (0.049) | -0.021 (0.067) | 0.013 (0.088) | -0.042 (0.071) | 0.103 (0.1) | 0.135 (0.093) |
| XL-HDL-PL | 0 (0.047) | -0.037 (0.065) | -0.026 (0.079) | -0.055 (0.069) | 0.081 (0.088) | 0.071 (0.069) |
| XL-HDL-TG | 0.086 (0.041) | 0.103 (0.059) | 0.14 (0.075) | 0.13 (0.063) | 0.165 (0.043) | 0.126 (0.051) |

**Appendix 3—table 4.** Mendelian randomization results using genome-wide significant SNPs and the weighted median estimator.

| | Method: Weighted median + Significant SNPs | | | | | |
|---|---|---|---|---|---|---|
| Selection | GERA | GERA | GERA | GLGC | Davis | Kettunen |
| Exposure | Davis | Davis | Kettunen | Davis | Kettunen | Davis |
| Outcome | CAD | UKB | UKB | UKB | UKB | UKB |
| VLDL traits | | | | | | |
| TG | 0.042 (0.055) | 0.191 (0.072) | NA | 0.228 (0.069) | NA | 0.195 (0.077) |
| VLDL-D | -0.098 (0.052) | 0.039 (0.095) | 0.057 (0.11) | 0.058 (0.093) | -0.107 (0.099) | -0.052 (0.115) |
| XS-VLDL-L | NA | NA | 0.312 (0.076) | NA | 0.393 (0.078) | NA |
| XS-VLDL-P | 0.101 (0.037) | 0.23 (0.052) | 0.303 (0.079) | 0.229 (0.052) | 0.409 (0.08) | 0.253 (0.059) |
| XS-VLDL-PL | 0.096 (0.039) | 0.242 (0.059) | 0.352 (0.087) | 0.228 (0.06) | 0.422 (0.065) | 0.319 (0.062) |
| XS-VLDL-TG | 0.125 (0.041) | 0.266 (0.057) | 0.287 (0.079) | 0.221 (0.056) | 0.361 (0.084) | 0.306 (0.069) |
| S-VLDL-C | 0.187 (0.059) | 0.232 (0.075) | NA | 0.256 (0.074) | NA | 0.303 (0.094) |
| S-VLDL-FC | 0.152 (0.057) | 0.207 (0.069) | 0.289 (0.093) | 0.227 (0.069) | 0.316 (0.109) | 0.279 (0.077) |
| S-VLDL-L | NA | NA | 0.282 (0.083) | NA | 0.306 (0.099) | NA |
| S-VLDL-P | 0.131 (0.057) | 0.202 (0.069) | 0.275 (0.085) | 0.221 (0.062) | 0.291 (0.093) | 0.226 (0.078) |
| S-VLDL-PL | 0.137 (0.053) | 0.205 (0.067) | 0.283 (0.083) | 0.218 (0.062) | 0.305 (0.092) | 0.263 (0.075) |
| S-VLDL-TG | 0.112 (0.057) | 0.204 (0.067) | 0.216 (0.088) | 0.229 (0.064) | 0.267 (0.099) | 0.244 (0.073) |
| M-VLDL-C | 0.12 (0.058) | 0.2 (0.07) | 0.255 (0.088) | 0.213 (0.066) | 0.303 (0.099) | 0.224 (0.081) |
| M-VLDL-CE | 0.144 (0.054) | 0.207 (0.071) | 0.262 (0.087) | 0.207 (0.068) | 0.301 (0.098) | 0.209 (0.072) |
| M-VLDL-FC | 0.081 (0.058) | 0.188 (0.074) | 0.221 (0.087) | 0.218 (0.068) | 0.272 (0.102) | 0.231 (0.08) |
| M-VLDL-L | NA | NA | 0.227 (0.095) | NA | 0.275 (0.109) | NA |
| M-VLDL-P | 0.047 (0.06) | 0.191 (0.072) | 0.221 (0.096) | 0.226 (0.069) | 0.31 (0.104) | 0.257 (0.079) |
| M-VLDL-PL | 0.103 (0.056) | 0.197 (0.071) | 0.228 (0.089) | 0.217 (0.064) | 0.29 (0.104) | 0.231 (0.078) |
| M-VLDL-TG | -0.005 (0.06) | 0.199 (0.075) | 0.224 (0.089) | 0.222 (0.068) | 0.318 (0.113) | 0.233 (0.085) |
| L-VLDL-C | 0.109 (0.068) | 0.2 (0.078) | 0.237 (0.093) | 0.231 (0.075) | 0.242 (0.122) | 0.262 (0.088) |
| L-VLDL-CE | 0.147 (0.063) | 0.211 (0.079) | 0.249 (0.09) | 0.253 (0.073) | 0.281 (0.11) | 0.286 (0.081) |
| L-VLDL-FC | 0.045 (0.065) | 0.199 (0.085) | 0.225 (0.093) | 0.224 (0.077) | 0.252 (0.125) | 0.228 (0.089) |
| L-VLDL-L | NA | NA | 0.243 (0.102) | NA | 0.261 (0.122) | NA |
| L-VLDL-P | 0.041 (0.064) | 0.209 (0.082) | 0.224 (0.092) | 0.21 (0.079) | 0.289 (0.122) | 0.223 (0.086) |
| L-VLDL-PL | 0.08 (0.063) | 0.201 (0.08) | 0.244 (0.101) | 0.224 (0.077) | 0.278 (0.123) | 0.247 (0.092) |
| L-VLDL-TG | -0.008 (0.061) | 0.215 (0.084) | 0.225 (0.103) | 0.161 (0.077) | 0.286 (0.13) | 0.277 (0.093) |
| XL-VLDL-L | NA | NA | 0.262 (0.111) | NA | NA | NA |

*Continued on next page*

*Appendix 3—table 4 continued*

| | Method: Weighted median + Significant SNPs | | | | | |
|---|---|---|---|---|---|---|
| Selection | GERA | GERA | GERA | GLGC | Davis | Kettunen |
| Exposure | Davis | Davis | Kettunen | Davis | Kettunen | Davis |
| Outcome | CAD | UKB | UKB | UKB | UKB | UKB |
| XL-VLDL-P | -0.026 (0.063) | 0.207 (0.091) | 0.289 (0.102) | 0.192 (0.088) | NA | 0.209 (0.101) |
| XL-VLDL-PL | -0.006 (0.067) | 0.197 (0.094) | 0.253 (0.094) | 0.213 (0.088) | NA | 0.24 (0.101) |
| XL-VLDL-TG | -0.026 (0.064) | 0.214 (0.092) | 0.229 (0.102) | 0.191 (0.088) | NA | 0.212 (0.099) |
| XXL-VLDL-L | NA | NA | 0.316 (0.114) | NA | -0.156 (0.22) | NA |
| XXL-VLDL-P | 0.091 (0.071) | 0.236 (0.089) | 0.267 (0.1) | 0.263 (0.088) | -0.104 (0.173) | 0.185 (0.098) |
| XXL-VLDL-PL | 0.153 (0.082) | 0.283 (0.096) | 0.267 (0.11) | 0.332 (0.095) | -0.139 (0.178) | 0.126 (0.124) |
| XXL-VLDL-TG | 0.126 (0.078) | 0.266 (0.096) | 0.244 (0.108) | 0.339 (0.097) | 0.227 (0.171) | 0.23 (0.123) |
| IDL/LDL traits | | | | | | |
| LDL-C | 0.263 (0.053) | 0.307 (0.066) | 0.274 (0.05) | 0.297 (0.063) | 0.435 (0.072) | 0.431 (0.067) |
| ApoB | 0.365 (0.073) | 0.472 (0.078) | 0.381 (0.063) | 0.375 (0.081) | 0.624 (0.08) | 0.565 (0.094) |
| LDL-D | 0.306 (0.09) | 0.413 (0.157) | 0.467 (0.163) | 0.271 (0.142) | 0.294 (0.075) | 0.193 (0.06) |
| S-LDL-C | 0.271 (0.058) | 0.342 (0.073) | 0.343 (0.056) | 0.273 (0.068) | 0.498 (0.08) | 0.274 (0.083) |
| S-LDL-L | NA | NA | 0.354 (0.061) | NA | 0.449 (0.081) | NA |
| S-LDL-P | 0.355 (0.063) | 0.366 (0.078) | 0.397 (0.069) | 0.329 (0.08) | 0.49 (0.089) | 0.581 (0.098) |
| M-LDL-C | 0.283 (0.055) | 0.313 (0.073) | 0.299 (0.05) | 0.244 (0.07) | 0.474 (0.074) | 0.297 (0.074) |
| M-LDL-CE | 0.27 (0.055) | 0.333 (0.077) | 0.299 (0.051) | 0.255 (0.071) | 0.437 (0.081) | 0.311 (0.077) |
| M-LDL-L | NA | NA | 0.303 (0.053) | NA | 0.432 (0.079) | NA |
| M-LDL-P | 0.251 (0.057) | 0.32 (0.071) | 0.309 (0.054) | 0.278 (0.07) | 0.409 (0.072) | 0.325 (0.078) |
| M-LDL-PL | 0.343 (0.063) | 0.337 (0.081) | 0.316 (0.055) | 0.318 (0.078) | 0.457 (0.074) | 0.353 (0.085) |
| L-LDL-C | 0.251 (0.052) | 0.29 (0.067) | 0.303 (0.048) | 0.231 (0.063) | 0.45 (0.075) | 0.309 (0.071) |
| L-LDL-CE | 0.251 (0.054) | 0.32 (0.068) | 0.293 (0.052) | 0.241 (0.066) | 0.481 (0.074) | 0.322 (0.077) |
| L-LDL-FC | 0.251 (0.048) | 0.214 (0.061) | 0.301 (0.049) | 0.214 (0.062) | 0.427 (0.068) | 0.289 (0.065) |
| L-LDL-L | NA | NA | 0.289 (0.051) | NA | 0.412 (0.07) | NA |
| L-LDL-P | 0.281 (0.053) | 0.321 (0.067) | 0.29 (0.053) | 0.244 (0.066) | 0.42 (0.072) | 0.351 (0.072) |
| L-LDL-PL | 0.286 (0.05) | 0.32 (0.067) | 0.313 (0.052) | 0.298 (0.065) | 0.413 (0.074) | 0.35 (0.076) |
| IDL-C | 0.283 (0.056) | 0.349 (0.068) | 0.315 (0.053) | 0.313 (0.07) | 0.51 (0.072) | 0.383 (0.068) |
| IDL-FC | 0.283 (0.053) | 0.334 (0.066) | 0.337 (0.053) | 0.314 (0.065) | 0.422 (0.067) | 0.367 (0.064) |
| IDL-L | NA | NA | 0.329 (0.056) | NA | 0.494 (0.069) | NA |
| IDL-P | 0.331 (0.06) | 0.44 (0.067) | 0.343 (0.056) | 0.371 (0.069) | 0.463 (0.074) | 0.328 (0.068) |
| IDL-PL | 0.265 (0.055) | 0.332 (0.066) | 0.344 (0.056) | 0.316 (0.066) | 0.451 (0.072) | 0.359 (0.066) |
| IDL-TG | 0.233 (0.067) | 0.371 (0.086) | 0.605 (0.078) | 0.337 (0.085) | 0.315 (0.082) | 0.215 (0.057) |
| HDL traits | | | | | | |
| HDL-C | -0.017 (0.04) | -0.167 (0.058) | -0.17 (0.072) | -0.167 (0.058) | -0.096 (0.077) | -0.085 (0.07) |
| ApoA1 | 0.094 (0.049) | -0.06 (0.076) | -0.069 (0.087) | -0.167 (0.07) | 0.005 (0.083) | -0.051 (0.121) |
| HDL-D | 0.079 (0.034) | 0.062 (0.061) | 0.102 (0.064) | 0.088 (0.061) | 0.099 (0.061) | 0.096 (0.058) |
| S-HDL-L | NA | NA | -0.174 (0.113) | NA | NA | NA |
| S-HDL-P | -0.173 (0.069) | 0.018 (0.106) | -0.171 (0.109) | -0.235 (0.113) | NA | -0.049 (0.108) |
| S-HDL-TG | 0.157 (0.061) | 0.238 (0.085) | 0.312 (0.105) | 0.228 (0.086) | 0.327 (0.105) | 0.229 (0.076) |
| M-HDL-C | -0.169 (0.054) | -0.236 (0.082) | -0.264 (0.097) | -0.241 (0.077) | -0.392 (0.098) | -0.266 (0.084) |
| M-HDL-CE | -0.166 (0.053) | -0.23 (0.08) | -0.271 (0.099) | -0.238 (0.075) | -0.394 (0.103) | -0.23 (0.085) |

*Continued on next page*

*Appendix 3—table 4 continued*

| | Method: Weighted median + Significant SNPs | | | | | |
|---|---|---|---|---|---|---|
| **Selection** | **GERA** | **GERA** | **GERA** | **GLGC** | **Davis** | **Kettunen** |
| **Exposure** | **Davis** | **Davis** | **Kettunen** | **Davis** | **Kettunen** | **Davis** |
| **Outcome** | **CAD** | **UKB** | **UKB** | **UKB** | **UKB** | **UKB** |
| M-HDL-FC | -0.166 (0.055) | -0.254 (0.086) | -0.281 (0.098) | -0.282 (0.087) | -0.28 (0.102) | -0.22 (0.1) |
| M-HDL-L | NA | NA | -0.296 (0.113) | NA | -0.448 (0.122) | NA |
| M-HDL-P | -0.157 (0.056) | -0.199 (0.09) | -0.298 (0.112) | -0.231 (0.086) | -0.291 (0.136) | -0.165 (0.131) |
| M-HDL-PL | -0.143 (0.058) | -0.183 (0.088) | -0.285 (0.108) | -0.183 (0.085) | -0.321 (0.114) | -0.203 (0.12) |
| L-HDL-C | 0.086 (0.037) | -0.009 (0.066) | 0.031 (0.083) | -0.032 (0.08) | 0.003 (0.09) | 0.006 (0.068) |
| L-HDL-CE | 0.086 (0.038) | -0.011 (0.067) | 0.075 (0.077) | -0.037 (0.076) | 0.015 (0.091) | -0.006 (0.068) |
| L-HDL-FC | 0.09 (0.039) | -0.005 (0.067) | 0.079 (0.081) | -0.019 (0.076) | 0.041 (0.078) | 0.027 (0.074) |
| L-HDL-L | NA | NA | 0.074 (0.077) | NA | 0.068 (0.084) | NA |
| L-HDL-P | 0.081 (0.036) | 0.046 (0.062) | 0.075 (0.074) | -0.01 (0.066) | 0.066 (0.07) | 0.078 (0.064) |
| L-HDL-PL | 0.084 (0.039) | 0 (0.067) | 0.051 (0.082) | -0.021 (0.071) | 0.054 (0.075) | 0.074 (0.071) |
| XL-HDL-C | 0.163 (0.047) | 0.122 (0.091) | 0.136 (0.087) | 0.132 (0.09) | 0.02 (0.098) | 0.161 (0.096) |
| XL-HDL-CE | 0.139 (0.044) | 0.106 (0.088) | 0.122 (0.09) | 0.148 (0.085) | 0.038 (0.091) | 0.336 (0.092) |
| XL-HDL-FC | 0.135 (0.048) | 0.065 (0.079) | 0.133 (0.081) | 0.027 (0.077) | 0.159 (0.079) | 0.052 (0.086) |
| XL-HDL-L | NA | NA | 0.119 (0.075) | NA | 0.023 (0.078) | NA |
| XL-HDL-P | 0.115 (0.035) | 0.087 (0.07) | 0.12 (0.073) | 0.129 (0.067) | 0.16 (0.071) | 0.15 (0.073) |
| XL-HDL-PL | 0.101 (0.037) | 0.064 (0.07) | 0.11 (0.072) | 0.121 (0.069) | 0.141 (0.069) | 0.088 (0.065) |
| XL-HDL-TG | 0.074 (0.027) | 0.107 (0.047) | 0.126 (0.051) | 0.118 (0.042) | 0.156 (0.05) | 0.114 (0.045) |

## Multivariable MR results

**Appendix 3—table 5.** Multivariable Mendelian randomization results (adjusted for HDL-C, LDL-C, and TG).

| Trait | HDL-C | LDL-C | TG | Subfraction |
|---|---|---|---|---|
| VLDL traits | | | | |
| VLDL-D | -0.251 (0.052) | 0.29 (0.037) | 0.6 (0.087) | -0.588 (0.094) |
| XS-VLDL-L | -0.086 (0.046) | 0.286 (0.077) | 0.089 (0.099) | 0.132 (0.119) |
| XS-VLDL-P | -0.083 (0.045) | 0.299 (0.078) | 0.093 (0.106) | 0.118 (0.125) |
| XS-VLDL-PL | -0.083 (0.046) | 0.249 (0.098) | 0.112 (0.076) | 0.159 (0.12) |
| XS-VLDL-TG | -0.114 (0.046) | 0.463 (0.079) | 0.286 (0.173) | -0.157 (0.187) |
| S-VLDL-C | -0.267 (0.084) | 0.754 (0.112) | 1.033 (0.28) | -1.035 (0.323) |
| S-VLDL-FC | -0.195 (0.068) | 0.898 (0.163) | 0.935 (0.26) | -1.027 (0.337) |
| S-VLDL-L | -0.25 (0.072) | 0.755 (0.112) | 0.876 (0.233) | -0.898 (0.28) |
| S-VLDL-P | -0.31 (0.101) | 0.819 (0.157) | 1.209 (0.4) | -1.245 (0.463) |
| S-VLDL-PL | -0.168 (0.051) | 0.673 (0.074) | 0.626 (0.159) | -0.613 (0.182) |
| S-VLDL-TG | -0.499 (0.305) | 0.906 (0.34) | 2.532 (1.57) | -2.628 (1.741) |
| M-VLDL-C | -0.201 (0.068) | 0.808 (0.127) | 1.472 (0.424) | -1.433 (0.451) |
| M-VLDL-CE | -0.168 (0.061) | 0.799 (0.111) | 0.996 (0.249) | -1.035 (0.293) |
| M-VLDL-FC | -0.2 (0.072) | 0.658 (0.089) | 1.469 (0.417) | -1.412 (0.444) |
| M-VLDL-L | -0.355 (0.139) | 0.602 (0.096) | 1.787 (0.654) | -1.878 (0.75) |
| M-VLDL-P | -0.362 (0.124) | 0.569 (0.08) | 1.889 (0.676) | -1.974 (0.745) |

*Continued on next page*

*Appendix 3—table 5 continued*

| Trait | HDL-C | LDL-C | TG | Subfraction |
|---|---|---|---|---|
| M-VLDL-PL | -0.332 (0.141) | 0.722 (0.159) | 1.996 (0.869) | -2.012 (0.943) |
| M-VLDL-TG | -0.408 (0.153) | 0.432 (0.061) | 1.974 (0.772) | -2.133 (0.879) |
| L-VLDL-C | -0.216 (0.063) | 0.509 (0.046) | 1.163 (0.254) | -1.254 (0.297) |
| L-VLDL-CE | -0.272 (0.072) | 0.465 (0.04) | 1.038 (0.242) | -1.081 (0.282) |
| L-VLDL-FC | -0.144 (0.059) | 0.493 (0.044) | 1.233 (0.27) | -1.274 (0.308) |
| L-VLDL-L | -0.228 (0.066) | 0.414 (0.045) | 1.17 (0.263) | -1.277 (0.313) |
| L-VLDL-P | -0.115 (0.056) | 0.442 (0.046) | 1.351 (0.317) | -1.357 (0.344) |
| L-VLDL-PL | -0.221 (0.111) | 0.473 (0.07) | 2.135 (0.948) | -2.316 (1.112) |
| L-VLDL-TG | -0.196 (0.066) | 0.355 (0.05) | 1.357 (0.322) | -1.428 (0.372) |
| XL-VLDL-L | -0.126 (0.049) | 0.451 (0.04) | 0.896 (0.159) | -1.069 (0.203) |
| XL-VLDL-P | -0.127 (0.053) | 0.474 (0.043) | 1.038 (0.183) | -1.209 (0.238) |
| XL-VLDL-PL | -0.138 (0.055) | 0.5 (0.044) | 1.052 (0.204) | -1.214 (0.257) |
| XL-VLDL-TG | -0.129 (0.049) | 0.424 (0.04) | 0.944 (0.167) | -1.071 (0.205) |
| XXL-VLDL-L | -0.228 (0.067) | 0.444 (0.043) | 0.978 (0.207) | -1.355 (0.318) |
| XXL-VLDL-P | 0.063 (0.076) | 0.452 (0.05) | 1.371 (0.384) | -1.639 (0.502) |
| XXL-VLDL-PL | -0.185 (0.056) | 0.371 (0.042) | 0.997 (0.185) | -1.259 (0.262) |
| XXL-VLDL-TG | -0.152 (0.059) | 0.41 (0.04) | 0.966 (0.19) | -1.202 (0.262) |
| LDL/IDL traits | | | | |
| ApoB | -0.084 (0.046) | 0.8 (0.146) | 0.427 (0.101) | -0.532 (0.191) |
| LDL-D | -0.057 (0.042) | 0.367 (0.03) | 0.21 (0.053) | 0.145 (0.061) |
| S-LDL-C | -0.062 (0.043) | 0.614 (0.126) | 0.261 (0.062) | -0.282 (0.152) |
| S-LDL-L | -0.06 (0.044) | 0.584 (0.118) | 0.266 (0.068) | -0.251 (0.145) |
| S-LDL-P | -0.033 (0.047) | 0.589 (0.119) | 0.29 (0.078) | -0.266 (0.151) |
| M-LDL-C | -0.082 (0.044) | 0.623 (0.146) | 0.203 (0.054) | -0.271 (0.162) |
| M-LDL-CE | -0.074 (0.043) | 0.485 (0.167) | 0.169 (0.059) | -0.088 (0.188) |
| M-LDL-L | -0.071 (0.044) | 0.444 (0.171) | 0.19 (0.063) | -0.069 (0.191) |
| M-LDL-P | -0.054 (0.044) | 0.539 (0.153) | 0.213 (0.063) | -0.179 (0.174) |
| M-LDL-PL | -0.081 (0.045) | 0.747 (0.134) | 0.232 (0.062) | -0.407 (0.162) |
| L-LDL-C | -0.071 (0.049) | 0.437 (0.242) | 0.167 (0.054) | -0.059 (0.261) |
| L-LDL-CE | -0.07 (0.048) | 0.277 (0.301) | 0.149 (0.065) | 0.116 (0.321) |
| L-LDL-FC | -0.112 (0.057) | 0.184 (0.304) | 0.163 (0.053) | 0.223 (0.315) |
| L-LDL-L | -0.075 (0.049) | 0.229 (0.26) | 0.146 (0.068) | 0.167 (0.273) |
| L-LDL-P | -0.083 (0.046) | 0.33 (0.2) | 0.128 (0.064) | 0.084 (0.213) |
| L-LDL-PL | -0.101 (0.046) | 0.446 (0.177) | 0.155 (0.057) | -0.036 (0.195) |
| IDL-C | -0.108 (0.057) | 0.231 (0.215) | 0.128 (0.064) | 0.192 (0.229) |
| IDL-FC | -0.107 (0.05) | 0.23 (0.147) | 0.123 (0.056) | 0.19 (0.156) |
| IDL-L | -0.1 (0.05) | 0.274 (0.161) | 0.123 (0.069) | 0.148 (0.175) |
| IDL-P | -0.101 (0.047) | 0.269 (0.134) | 0.109 (0.071) | 0.153 (0.148) |
| IDL-PL | -0.076 (0.048) | 0.25 (0.162) | 0.134 (0.071) | 0.153 (0.18) |
| IDL-TG | -0.083 (0.046) | 0.314 (0.069) | 0.103 (0.089) | 0.11 (0.103) |
| HDL traits | | | | |
| ApoA1 | 0.345 (0.25) | 0.544 (0.081) | 0.334 (0.109) | -0.481 (0.271) |
| HDL-D | -0.442 (0.124) | 0.421 (0.033) | 0.111 (0.055) | 0.333 (0.114) |

*Continued on next page*

*Appendix 3—table 5 continued*

| Trait | HDL-C | LDL-C | TG | Subfraction |
|---|---|---|---|---|
| S-HDL-L | -0.117 (0.046) | 0.488 (0.044) | 0.189 (0.054) | -0.312 (0.106) |
| S-HDL-P | -0.112 (0.046) | 0.453 (0.035) | 0.225 (0.056) | -0.331 (0.095) |
| S-HDL-TG | 0.002 (0.145) | 0.314 (0.156) | -0.007 (0.469) | 0.253 (0.637) |
| M-HDL-C | 0.179 (0.097) | 0.36 (0.038) | 0.147 (0.054) | -0.508 (0.165) |
| M-HDL-CE | 0.167 (0.087) | 0.319 (0.036) | 0.166 (0.055) | -0.487 (0.157) |
| M-HDL-FC | 0.339 (0.141) | 0.436 (0.04) | 0.247 (0.059) | -0.649 (0.225) |
| M-HDL-L | 0.27 (0.108) | 0.362 (0.032) | 0.299 (0.063) | -0.606 (0.188) |
| M-HDL-P | 0.302 (0.112) | 0.386 (0.033) | 0.371 (0.075) | -0.694 (0.204) |
| M-HDL-PL | 0.311 (0.117) | 0.402 (0.033) | 0.333 (0.07) | -0.632 (0.191) |
| L-HDL-C | -0.589 (0.211) | 0.469 (0.039) | 0.146 (0.055) | 0.516 (0.213) |
| L-HDL-CE | -0.602 (0.239) | 0.477 (0.042) | 0.137 (0.056) | 0.519 (0.23) |
| L-HDL-FC | -0.573 (0.177) | 0.437 (0.034) | 0.171 (0.054) | 0.518 (0.181) |
| L-HDL-L | -0.556 (0.193) | 0.437 (0.034) | 0.142 (0.055) | 0.457 (0.189) |
| L-HDL-P | -0.515 (0.198) | 0.417 (0.03) | 0.133 (0.056) | 0.422 (0.191) |
| L-HDL-PL | -0.53 (0.201) | 0.415 (0.034) | 0.152 (0.055) | 0.443 (0.202) |
| XL-HDL-C | -0.447 (0.182) | 0.342 (0.036) | 0.071 (0.079) | 0.474 (0.223) |
| XL-HDL-CE | -0.425 (0.146) | 0.366 (0.038) | 0.051 (0.069) | 0.426 (0.177) |
| XL-HDL-FC | -0.459 (0.147) | 0.377 (0.031) | 0.097 (0.062) | 0.433 (0.16) |
| XL-HDL-L | -0.405 (0.146) | 0.364 (0.031) | 0.077 (0.068) | 0.358 (0.154) |
| XL-HDL-P | -0.451 (0.134) | 0.374 (0.03) | 0.078 (0.064) | 0.41 (0.139) |
| XL-HDL-PL | -0.422 (0.119) | 0.412 (0.033) | 0.115 (0.055) | 0.343 (0.118) |
| XL-HDL-TG | -0.186 (0.073) | 0.336 (0.035) | 0.045 (0.086) | 0.147 (0.074) |

**Appendix 3—table 6.** Multivariable Mendelian randomization results (adjusted for ApoA1, ApoB, and TG).

| Trait | ApoA1 | ApoB | TG | Subfraction |
|---|---|---|---|---|
| VLDL traits | | | | |
| VLDL-D | -0.227 (0.067) | 0.545 (0.092) | 0.208 (0.139) | -0.32 (0.112) |
| XS-VLDL-L | -0.123 (0.063) | 0.53 (0.163) | -0.121 (0.085) | 0.084 (0.141) |
| XS-VLDL-P | -0.121 (0.064) | 0.553 (0.17) | -0.123 (0.088) | 0.061 (0.158) |
| XS-VLDL-PL | -0.147 (0.066) | 0.273 (0.138) | 0.028 (0.05) | 0.253 (0.135) |
| XS-VLDL-TG | -0.102 (0.06) | 0.762 (0.168) | 0.069 (0.055) | -0.248 (0.15) |
| S-VLDL-C | -0.384 (0.141) | 1.426 (0.354) | 0.606 (0.351) | -1.265 (0.568) |
| S-VLDL-FC | -0.188 (0.077) | 1.001 (0.235) | 0.081 (0.053) | -0.489 (0.213) |
| S-VLDL-L | -0.46 (0.146) | 1.776 (0.417) | 0.7 (0.316) | -1.629 (0.586) |
| S-VLDL-P | -0.494 (0.159) | 1.677 (0.386) | 0.825 (0.372) | -1.644 (0.606) |
| S-VLDL-PL | -0.262 (0.097) | 1.41 (0.343) | 0.532 (0.261) | -1.213 (0.478) |
| S-VLDL-TG | -0.18 (0.069) | 0.792 (0.121) | 0.078 (0.051) | -0.301 (0.108) |
| M-VLDL-C | -0.157 (0.062) | 0.867 (0.132) | 0.085 (0.051) | -0.373 (0.118) |
| M-VLDL-CE | -0.221 (0.069) | 1.224 (0.223) | 0.47 (0.21) | -0.995 (0.338) |
| M-VLDL-FC | -0.222 (0.074) | 0.902 (0.133) | 0.482 (0.251) | -0.799 (0.311) |
| M-VLDL-L | -0.174 (0.065) | 0.76 (0.104) | 0.073 (0.05) | -0.298 (0.098) |
| M-VLDL-P | -0.181 (0.065) | 0.764 (0.1) | 0.077 (0.051) | -0.312 (0.096) |

*Continued on next page*

*Appendix 3—table 6 continued*

| Trait | ApoA1 | ApoB | TG | Subfraction |
|---|---|---|---|---|
| M-VLDL-PL | -0.159 (0.065) | 0.776 (0.116) | 0.08 (0.051) | -0.297 (0.106) |
| M-VLDL-TG | -0.263 (0.106) | 0.724 (0.094) | 0.547 (0.406) | -0.806 (0.455) |
| L-VLDL-C | -0.218 (0.084) | 0.732 (0.101) | 0.352 (0.278) | -0.609 (0.337) |
| L-VLDL-CE | -0.293 (0.079) | 0.781 (0.096) | 0.405 (0.189) | -0.673 (0.217) |
| L-VLDL-FC | -0.197 (0.069) | 0.737 (0.094) | 0.365 (0.25) | -0.619 (0.291) |
| L-VLDL-L | -0.194 (0.071) | 0.666 (0.087) | 0.289 (0.234) | -0.532 (0.278) |
| L-VLDL-P | -0.184 (0.061) | 0.677 (0.086) | 0.415 (0.217) | -0.617 (0.229) |
| L-VLDL-PL | -0.155 (0.063) | 0.715 (0.095) | 0.075 (0.051) | -0.287 (0.104) |
| L-VLDL-TG | -0.154 (0.062) | 0.67 (0.083) | 0.073 (0.05) | -0.252 (0.091) |
| XL-VLDL-L | -0.186 (0.066) | 0.694 (0.088) | 0.263 (0.19) | -0.577 (0.249) |
| XL-VLDL-P | -0.167 (0.061) | 0.742 (0.088) | 0.075 (0.05) | -0.373 (0.109) |
| XL-VLDL-PL | -0.191 (0.068) | 0.712 (0.092) | 0.271 (0.197) | -0.583 (0.268) |
| XL-VLDL-TG | -0.195 (0.068) | 0.666 (0.087) | 0.334 (0.21) | -0.603 (0.248) |
| XXL-VLDL-L | -0.173 (0.066) | 0.732 (0.098) | 0.088 (0.052) | -0.402 (0.144) |
| XXL-VLDL-P | -0.071 (0.065) | 0.705 (0.097) | 0.607 (0.321) | -1.089 (0.449) |
| XXL-VLDL-PL | -0.244 (0.082) | 0.666 (0.091) | 0.414 (0.257) | -0.814 (0.344) |
| XXL-VLDL-TG | -0.3 (0.091) | 0.694 (0.095) | 0.627 (0.306) | -1.075 (0.402) |
| IDL/LDL traits | | | | |
| LDL-C | -0.119 (0.062) | 0.247 (0.167) | 0.066 (0.054) | 0.319 (0.182) |
| LDL-D | -0.123 (0.06) | 0.544 (0.091) | -0.036 (0.087) | 0.119 (0.071) |
| S-LDL-C | -0.097 (0.06) | 0.438 (0.216) | 0.044 (0.051) | 0.08 (0.238) |
| S-LDL-L | -0.097 (0.063) | 0.503 (0.268) | 0.043 (0.051) | -0.005 (0.29) |
| S-LDL-P | -0.059 (0.103) | 0.932 (0.597) | -0.122 (0.112) | -0.362 (0.596) |
| M-LDL-C | -0.099 (0.065) | 0.78 (1.034) | -0.172 (0.425) | -0.169 (0.909) |
| M-LDL-CE | -0.157 (0.128) | -0.346 (2.587) | 0.195 (0.855) | 0.854 (2.221) |
| M-LDL-L | -0.123 (0.095) | 0.247 (1.479) | -0.001 (0.445) | 0.32 (1.293) |
| M-LDL-P | -0.134 (0.07) | 0.13 (0.286) | 0.053 (0.052) | 0.432 (0.31) |
| M-LDL-PL | -0.075 (0.077) | 1.165 (0.868) | -0.248 (0.253) | -0.566 (0.839) |
| L-LDL-C | -0.855 (1.68) | -5.337 (13.402) | 2.405 (5.735) | 5.257 (11.72) |
| L-LDL-CE | -0.151 (0.065) | 0.129 (0.193) | 0.061 (0.052) | 0.461 (0.213) |
| L-LDL-FC | -0.397 (0.219) | -1.139 (1.395) | 0.786 (0.711) | 1.531 (1.189) |
| L-LDL-L | -0.265 (0.148) | -0.854 (1.42) | 0.41 (0.51) | 1.266 (1.188) |
| L-LDL-P | -0.258 (0.153) | -0.607 (1.225) | 0.276 (0.402) | 1.064 (1.029) |
| L-LDL-PL | -0.312 (0.187) | -0.741 (1.411) | 0.39 (0.518) | 1.227 (1.245) |
| IDL-C | -0.3 (0.123) | -0.334 (0.616) | 0.276 (0.254) | 0.769 (0.501) |
| IDL-FC | -0.199 (0.069) | 0.247 (0.118) | 0.044 (0.049) | 0.33 (0.127) |
| IDL-L | -0.215 (0.089) | 0.021 (0.409) | 0.101 (0.15) | 0.444 (0.328) |
| IDL-P | -0.175 (0.075) | 0.214 (0.172) | 0.04 (0.051) | 0.292 (0.173) |
| IDL-PL | -0.183 (0.07) | 0.159 (0.172) | 0.031 (0.049) | 0.406 (0.184) |
| IDL-TG | -0.143 (0.075) | 0.565 (0.146) | -0.119 (0.087) | 0.047 (0.135) |
| HDL traits | | | | |
| HDL-C | -1.513 (1.109) | 0.982 (0.314) | 0.27 (0.291) | 1.446 (1.112) |
| HDL-D | -0.457 (0.138) | 0.613 (0.073) | 0.056 (0.049) | 0.296 (0.1) |

*Continued on next page*

*Appendix 3—table 6 continued*

| Trait | ApoA1 | ApoB | TG | Subfraction |
|-------|-------|------|-----|-------------|
| S-HDL-L | -0.128 (0.059) | 0.524 (0.062) | 0.067 (0.05) | -0.224 (0.087) |
| S-HDL-P | -0.132 (0.059) | 0.531 (0.059) | 0.071 (0.05) | -0.24 (0.083) |
| S-HDL-TG | -0.11 (0.113) | 0.595 (0.221) | -0.057 (0.297) | -0.044 (0.466) |
| M-HDL-C | 0.091 (0.084) | 0.459 (0.101) | -0.1 (0.083) | -0.442 (0.143) |
| M-HDL-CE | 0.09 (0.078) | 0.291 (0.083) | 0.082 (0.05) | -0.413 (0.137) |
| M-HDL-FC | 0.148 (0.11) | 0.378 (0.063) | 0.066 (0.049) | -0.408 (0.166) |
| M-HDL-L | 0.133 (0.091) | 0.491 (0.097) | -0.029 (0.086) | -0.485 (0.155) |
| M-HDL-P | 0.129 (0.097) | 0.501 (0.097) | -0.004 (0.09) | -0.472 (0.166) |
| M-HDL-PL | 0.162 (0.107) | 0.519 (0.096) | -0.037 (0.087) | -0.486 (0.171) |
| L-HDL-C | -0.724 (0.232) | 0.856 (0.132) | 0.032 (0.093) | 0.575 (0.204) |
| L-HDL-CE | -0.761 (0.236) | 0.899 (0.145) | 0.004 (0.084) | 0.61 (0.206) |
| L-HDL-FC | -0.749 (0.174) | 0.842 (0.102) | 0.094 (0.05) | 0.59 (0.148) |
| L-HDL-L | -0.717 (0.217) | 0.815 (0.12) | 0.023 (0.089) | 0.541 (0.184) |
| L-HDL-P | -0.653 (0.191) | 0.749 (0.104) | 0.057 (0.049) | 0.476 (0.155) |
| L-HDL-PL | -0.679 (0.201) | 0.774 (0.109) | 0.05 (0.049) | 0.51 (0.169) |
| XL-HDL-C | -0.639 (0.194) | 0.692 (0.095) | -0.058 (0.086) | 0.565 (0.196) |
| XL-HDL-CE | -0.576 (0.2) | 0.667 (0.096) | -0.077 (0.086) | 0.511 (0.206) |
| XL-HDL-FC | -0.734 (0.174) | 0.674 (0.073) | 0.094 (0.052) | 0.609 (0.159) |
| XL-HDL-L | -0.652 (0.168) | 0.733 (0.097) | -0.06 (0.084) | 0.481 (0.141) |
| XL-HDL-P | -0.52 (0.147) | 0.691 (0.094) | -0.075 (0.084) | 0.39 (0.135) |
| XL-HDL-PL | -0.652 (0.151) | 0.687 (0.076) | 0.079 (0.051) | 0.466 (0.12) |
| XL-HDL-TG | -0.281 (0.111) | 0.539 (0.09) | -0.152 (0.092) | 0.165 (0.086) |

## Q-statistics for multivariable Mendelian randomization

Here we provide the list of modified Cochran's Q-statistics for the multivariable MR analyses (*Appendix 3—tables 7* and *8*).

**Appendix 3—table 7.** Modified Cochran's Q-statistics (p-values) for the multivariable Mendelian randomization analyses (adjusted for HDL-C, LDL-C, and TG).
DF is short for degrees of freedom.

| Trait | DF | HDL-C | LDL-C | TG | Subfraction |
|-------|-----|-------|-------|-----|-------------|
| VLDL traits | | | | | |
| VLDL-D | 432 | 7640.8 (0) | 1918.9 (7.9e-186) | 877.6 (1.4e-32) | 840.2 (1.6e-28) |
| XS-VLDL-L | 436 | 7983.9 (0) | 1104.9 (1.1e-59) | 1935.8 (2.2e-187) | 926 (1.9e-37) |
| XS-VLDL-P | 436 | 7927.8 (0) | 1066.6 (1.1e-54) | 1814 (4.8e-167) | 893.6 (9.6e-34) |
| XS-VLDL-PL | 435 | 8291.5 (0) | 968.1 (1.4e-42) | 2771.5 (0) | 849.8 (4.3e-29) |
| XS-VLDL-TG | 431 | 7549.8 (0) | 894.4 (1.3e-34) | 739.5 (1.3e-18) | 682.5 (1.2e-13) |
| S-VLDL-C | 429 | 8598.1 (0) | 652.6 (1.7e-11) | 1220.7 (4.6e-77) | 541.3 (0.00018) |
| S-VLDL-FC | 434 | 7861.2 (0) | 576 (5.4e-06) | 519.4 (0.003) | 507.9 (0.0082) |
| S-VLDL-L | 438 | 7105.3 (0) | 626 (8.5e-09) | 525.2 (0.0026) | 514.3 (0.0069) |
| S-VLDL-P | 438 | 6686.5 (0) | 616.5 (3.6e-08) | 515.6 (0.0061) | 507.3 (0.012) |
| S-VLDL-PL | 437 | 7589.1 (0) | 702.8 (1e-14) | 591.5 (1.1e-06) | 555.1 (0.00011) |
| S-VLDL-TG | 437 | 7658.7 (0) | 612.7 (5.3e-08) | 498.9 (0.021) | 494.5 (0.03) |

*Continued on next page*

*Appendix 3—table 7 continued*

| Trait | DF | HDL-C | LDL-C | TG | Subfraction |
|---|---|---|---|---|---|
| M-VLDL-C | 432 | 9167.8 (0) | 740.8 (1.3e-18) | 558.9 (3.5e-05) | 551.5 (8.3e-05) |
| M-VLDL-CE | 432 | 8055.2 (0) | 705.9 (1.6e-15) | 556.6 (4.6e-05) | 539.7 (0.00031) |
| M-VLDL-FC | 436 | 8272.8 (0) | 814.8 (2.7e-25) | 528.3 (0.0016) | 519.1 (0.0037) |
| M-VLDL-L | 429 | 7109.2 (0) | 1269.2 (5.5e-84) | 532.6 (0.00047) | 515.9 (0.0025) |
| M-VLDL-P | 436 | 8260.7 (0) | 2059.5 (2.1e-208) | 527.5 (0.0017) | 516.8 (0.0046) |
| M-VLDL-PL | 435 | 6849.2 (0) | 599.6 (2.6e-07) | 496.8 (0.021) | 493.5 (0.027) |
| M-VLDL-TG | 436 | 6123.7 (0) | 9854.8 (0) | 532.3 (0.0011) | 521 (0.0031) |
| L-VLDL-C | 435 | 8617.2 (0) | 8966 (0) | 654.7 (4.3e-11) | 561.5 (3.9e-05) |
| L-VLDL-CE | 434 | 6636.6 (0) | 11134 (0) | 581.6 (2.6e-06) | 539.5 (0.00041) |
| L-VLDL-FC | 431 | 7779.6 (0) | 6691 (0) | 595.1 (2.5e-07) | 562.7 (1.9e-05) |
| L-VLDL-L | 434 | 8104.9 (0) | 5191.4 (0) | 560.3 (3.9e-05) | 548.6 (0.00015) |
| L-VLDL-P | 435 | 2308 (5.1e-252) | 10360.3 (0) | 545.4 (0.00024) | 537.9 (0.00054) |
| L-VLDL-PL | 430 | 8155.4 (0) | 1310.8 (8.6e-90) | 491.8 (0.021) | 489.7 (0.024) |
| L-VLDL-TG | 438 | 8581.8 (0) | 4800.1 (0) | 569.1 (2.3e-05) | 559.2 (7.5e-05) |
| XL-VLDL-L | 437 | 8686.8 (0) | 8322.2 (0) | 674.7 (1.9e-12) | 620.2 (1.7e-08) |
| XL-VLDL-P | 431 | 8550.2 (0) | 2459.4 (2e-280) | 608.3 (3.6e-08) | 588.6 (6.3e-07) |
| XL-VLDL-PL | 431 | 7478.2 (0) | 5042.5 (0) | 613.3 (1.7e-08) | 591.6 (4.1e-07) |
| XL-VLDL-TG | 433 | 8237.3 (0) | 9628.9 (0) | 651.8 (4.6e-11) | 618.3 (1.1e-08) |
| XXL-VLDL-L | 439 | 8476.2 (0) | 10436.4 (0) | 652.9 (1.3e-10) | 570.7 (2.2e-05) |
| XXL-VLDL-P | 437 | 1291.3 (2.8e-85) | 9987.4 (0) | 540.3 (0.00053) | 529.5 (0.0016) |
| XXL-VLDL-PL | 436 | 9631.8 (0) | 11287.1 (0) | 641.6 (4.8e-10) | 595.5 (5.3e-07) |
| XXL-VLDL-TG | 429 | 7809.4 (0) | 9476.4 (0) | 595.6 (1.7e-07) | 564 (1.2e-05) |
| LDL/IDL traits | | | | | |
| ApoB | 435 | 9220.8 (0) | 550.1 (0.00014) | 1809.7 (1.2e-166) | 535.1 (0.00072) |
| LDL-D | 429 | 2909.2 (0) | 3918.8 (0) | 2706 (0) | 1426.1 (2.9e-107) |
| S-LDL-C | 431 | 8189.7 (0) | 569.8 (7.8e-06) | 4880.9 (0) | 564.1 (1.6e-05) |
| S-LDL-L | 435 | 8403.8 (0) | 574.4 (7.8e-06) | 3931.2 (0) | 564.3 (2.7e-05) |
| S-LDL-P | 431 | 7371.4 (0) | 547.1 (0.00012) | 3144.7 (0) | 537.9 (0.00034) |
| M-LDL-C | 430 | 9723.7 (0) | 570.9 (5.8e-06) | 6568.6 (0) | 562.9 (1.6e-05) |
| M-LDL-CE | 432 | 8442.1 (0) | 558.3 (3.8e-05) | 5773.6 (0) | 549.1 (0.00011) |
| M-LDL-L | 430 | 8801.7 (0) | 555.4 (4e-05) | 5176.1 (0) | 548.2 (9.5e-05) |
| M-LDL-P | 429 | 8798.9 (0) | 541.6 (0.00018) | 5049.7 (0) | 535.2 (0.00035) |
| M-LDL-PL | 436 | 7981.7 (0) | 573.9 (9.6e-06) | 4304.8 (0) | 558.9 (6e-05) |
| L-LDL-C | 432 | 8865.2 (0) | 567.7 (1.2e-05) | 6179.8 (0) | 567 (1.3e-05) |
| L-LDL-CE | 433 | 8464.3 (0) | 558.7 (4.1e-05) | 5731.3 (0) | 555.6 (5.9e-05) |
| L-LDL-FC | 431 | 7481.1 (0) | 580.6 (1.9e-06) | 6760.8 (0) | 580.2 (2e-06) |
| L-LDL-L | 433 | 8486.8 (0) | 604.5 (8.9e-08) | 5755.8 (0) | 601.8 (1.3e-07) |
| L-LDL-P | 434 | 8310.7 (0) | 592.1 (6.3e-07) | 5553.3 (0) | 584.9 (1.7e-06) |
| L-LDL-PL | 435 | 8341.4 (0) | 588.5 (1.2e-06) | 5327.8 (0) | 577.4 (5.3e-06) |
| IDL-C | 434 | 7873.9 (0) | 645.5 (1.7e-10) | 6336 (0) | 642.1 (2.9e-10) |
| IDL-FC | 432 | 8036 (0) | 729.5 (1.4e-17) | 6630.5 (0) | 725.6 (3e-17) |
| IDL-L | 434 | 7869.8 (0) | 694.5 (2.4e-14) | 5198.3 (0) | 689 (7e-14) |
| IDL-P | 436 | 9660.5 (0) | 736.7 (9e-18) | 5002 (0) | 726.6 (7.1e-17) |

*Continued on next page*

*Appendix 3—table 7 continued*

| Trait | DF | HDL-C | LDL-C | TG | Subfraction |
|---|---|---|---|---|---|
| IDL-PL | 431 | 8432.6 (0) | 680.6 (1.7e-13) | 5023 (0) | 677.4 (3e-13) |
| IDL-TG | 436 | 7741.2 (0) | 1077.5 (4.2e-56) | 1992.9 (4.9e-197) | 931.6 (4.4e-38) |
| HDL traits | | | | | |
| ApoA1 | 434 | 494.1 (0.024) | 511.5 (0.006) | 932.1 (1.8e-38) | 492 (0.028) |
| HDL-D | 438 | 783.5 (6.6e-22) | 8500 (0) | 5713.2 (0) | 860.1 (9.4e-30) |
| S-HDL-L | 438 | 3067.3 (0) | 4414.6 (0) | 3763.2 (0) | 882.2 (3.7e-32) |
| S-HDL-P | 438 | 2592.4 (1.1e-301) | 7652.1 (0) | 3097.3 (0) | 951.1 (4.9e-40) |
| S-HDL-TG | 425 | 896.9 (6.9e-36) | 641.3 (5.2e-11) | 540.1 (0.00013) | 523 (8e-04) |
| M-HDL-C | 437 | 957.6 (5.5e-41) | 10172.4 (0) | 4875.5 (0) | 628.3 (4.9e-09) |
| M-HDL-CE | 434 | 955.3 (3.2e-41) | 1383.1 (1.7e-99) | 4355.4 (0) | 648.3 (1e-10) |
| M-HDL-FC | 432 | 759.4 (2.4e-20) | 2989.1 (0) | 3512.2 (0) | 538.2 (0.00037) |
| M-HDL-L | 435 | 914.2 (3e-36) | 11535.3 (0) | 2327.7 (1.7e-255) | 570.3 (1.3e-05) |
| M-HDL-P | 434 | 997.6 (2.3e-46) | 10709.6 (0) | 1942.9 (3.2e-189) | 561.3 (3.4e-05) |
| M-HDL-PL | 434 | 977.8 (6.3e-44) | 9439.9 (0) | 2566 (1.8e-298) | 581.3 (2.7e-06) |
| L-HDL-C | 434 | 580 (3.2e-06) | 1257.1 (4.4e-81) | 4502.7 (0) | 604.3 (1.1e-07) |
| L-HDL-CE | 434 | 549 (0.00014) | 930.2 (3e-38) | 5517.2 (0) | 557.2 (5.6e-05) |
| L-HDL-FC | 441 | 627.6 (1.2e-08) | 8415.3 (0) | 3594 (0) | 658.4 (7.9e-11) |
| L-HDL-L | 434 | 603.6 (1.2e-07) | 6743.8 (0) | 5314.7 (0) | 623.7 (5.7e-09) |
| L-HDL-P | 432 | 601.1 (1.2e-07) | 7769.3 (0) | 6024.6 (0) | 633.2 (8.6e-10) |
| L-HDL-PL | 434 | 584.5 (1.8e-06) | 9935.5 (0) | 3544.3 (0) | 611.3 (3.8e-08) |
| XL-HDL-C | 430 | 732.9 (3.9e-18) | 10426.6 (0) | 2077.7 (1.4e-213) | 686.9 (4e-14) |
| XL-HDL-CE | 430 | 771.4 (9.3e-22) | 8564.4 (0) | 2457 (2.2e-280) | 711.4 (3.3e-16) |
| XL-HDL-FC | 432 | 761.8 (1.4e-20) | 11265.2 (0) | 2549.4 (3.1e-296) | 770.9 (1.9e-21) |
| XL-HDL-L | 429 | 767.6 (1.6e-21) | 11490.7 (0) | 2355.7 (1.2e-262) | 784.6 (3.4e-23) |
| XL-HDL-P | 433 | 724.9 (4.6e-17) | 11372.5 (0) | 2539.9 (3.9e-294) | 798.5 (4.8e-24) |
| XL-HDL-PL | 443 | 809.7 (7.8e-24) | 10093.1 (0) | 5762 (0) | 895.4 (7.5e-33) |
| XL-HDL-TG | 432 | 1849.1 (3.9e-174) | 2635.9 (6.5e-312) | 2240.8 (2.9e-241) | 1267.8 (4.4e-83) |

**Appendix 3—table 8.** Modified Cochran's Q-statistics (p-values) for the multivariable Mendelian randomization analyses (adjusted for ApoA1, ApoB, and TG).
DF is short for degrees of freedom.

| Trait | DF | ApoA1 | ApoB | TG | Subfraction |
|---|---|---|---|---|---|
| VLDL traits | | | | | |
| VLDL-D | 297 | 1194.1 (9.1e-108) | 550 (2.4e-17) | 573.7 (8.2e-20) | 606.7 (2.1e-23) |
| XS-VLDL-L | 295 | 1185.1 (6.7e-107) | 927 (2e-66) | 1151.3 (2.2e-101) | 887.9 (1.1e-60) |
| XS-VLDL-P | 295 | 1194.9 (1.7e-108) | 900 (1.9e-62) | 895.5 (8.7e-62) | 826.7 (6.4e-52) |
| XS-VLDL-PL | 296 | 1148.5 (1.2e-100) | 973.9 (3.2e-73) | 2104.2 (1.4e-269) | 961.4 (2.5e-71) |
| XS-VLDL-TG | 302 | 1263.7 (1.1e-117) | 757.9 (4.7e-41) | 1308.1 (4.4e-125) | 976.5 (4.6e-72) |
| S-VLDL-C | 290 | 988.8 (4.4e-77) | 394 (4.5e-05) | 459.8 (7.8e-10) | 402.6 (1.3e-05) |
| S-VLDL-FC | 296 | 1092 (1.4e-91) | 904 (8.6e-63) | 1238.7 (2.1e-115) | 1010.4 (8.1e-79) |
| S-VLDL-L | 301 | 1107.9 (1.1e-92) | 412.3 (2.1e-05) | 420.8 (5.9e-06) | 384.7 (0.00078) |
| S-VLDL-P | 301 | 1116.6 (4.6e-94) | 424.8 (3.3e-06) | 401.3 (9.4e-05) | 380.6 (0.0013) |

*Continued on next page*

*Appendix 3—table 8 continued*

| Trait | DF | ApoA1 | ApoB | TG | Subfraction |
|---|---|---|---|---|---|
| S-VLDL-PL | 299 | 1096 (2.3e-91) | 428.9 (1.2e-06) | 446 (7.1e-08) | 432.1 (7.1e-07) |
| S-VLDL-TG | 300 | 1152.4 (4.3e-100) | 908.5 (1.8e-62) | 1453.4 (1.8e-150) | 1303.1 (7.1e-125) |
| M-VLDL-C | 298 | 1171.2 (1e-103) | 824 (7.3e-51) | 1480 (8.9e-156) | 1212.5 (1.8e-110) |
| M-VLDL-CE | 298 | 1185.4 (4.9e-106) | 564.4 (1.1e-18) | 468.9 (9.2e-10) | 431.6 (6.3e-07) |
| M-VLDL-FC | 298 | 1190.4 (7.4e-107) | 899.8 (1.1e-61) | 415.2 (8.1e-06) | 398.8 (8.4e-05) |
| M-VLDL-L | 298 | 1144.1 (2.4e-99) | 869.8 (2.4e-57) | 1381 (1e-138) | 1237.4 (1.4e-114) |
| M-VLDL-P | 297 | 1121.3 (5.7e-96) | 821.1 (1.1e-50) | 1250.5 (4.6e-117) | 1206.7 (8.1e-110) |
| M-VLDL-PL | 298 | 1149.9 (2.8e-100) | 843.2 (1.5e-53) | 1391.8 (1.5e-140) | 1226.3 (9.8e-113) |
| M-VLDL-TG | 296 | 1187.4 (5.8e-107) | 717.3 (5.8e-37) | 366.3 (0.0033) | 360.6 (0.006) |
| L-VLDL-C | 295 | 1196.5 (9.1e-109) | 820 (5.6e-51) | 462.5 (1.5e-09) | 376.9 (0.00088) |
| L-VLDL-CE | 302 | 1183.1 (1.8e-104) | 844.6 (7.4e-53) | 541.8 (7.2e-16) | 441.7 (2.6e-07) |
| L-VLDL-FC | 295 | 1172.3 (8.2e-105) | 851.6 (1.9e-55) | 460.8 (2.1e-09) | 406.2 (1.8e-05) |
| L-VLDL-L | 295 | 1163.6 (2.2e-103) | 797 (8.8e-48) | 406.5 (1.7e-05) | 391.5 (0.00014) |
| L-VLDL-P | 293 | 1160.2 (2e-103) | 809.5 (5.9e-50) | 420.2 (1.5e-06) | 407.9 (1e-05) |
| L-VLDL-PL | 296 | 1292 (2.6e-124) | 833.4 (1.3e-52) | 1216.5 (9.7e-112) | 1098.9 (1.1e-92) |
| L-VLDL-TG | 294 | 1150.8 (1.3e-101) | 1213.6 (7e-112) | 1262.6 (5.2e-120) | 1162.8 (1.5e-103) |
| XL-VLDL-L | 294 | 1196 (5.4e-109) | 829.4 (1.6e-52) | 442 (4.9e-08) | 423.6 (1.1e-06) |
| XL-VLDL-P | 294 | 1265.9 (1.4e-120) | 1180.9 (1.6e-106) | 1202.2 (5.2e-110) | 982.1 (5.4e-75) |
| XL-VLDL-PL | 296 | 1199.1 (6.9e-109) | 874.2 (1.9e-58) | 421.2 (2.3e-06) | 405.6 (2.3e-05) |
| XL-VLDL-TG | 296 | 1184.3 (1.8e-106) | 828.6 (5.9e-52) | 430.8 (4.9e-07) | 430.1 (5.5e-07) |
| XXL-VLDL-L | 304 | 1119.2 (1.2e-93) | 1041.9 (1.6e-81) | 900.9 (2e-60) | 699.6 (3.2e-33) |
| XXL-VLDL-P | 303 | 1148 (1.7e-98) | 876.4 (4e-57) | 382.2 (0.0013) | 366 (0.0076) |
| XXL-VLDL-PL | 303 | 1203 (2.1e-107) | 775.1 (4e-43) | 438.1 (5.8e-07) | 376.5 (0.0025) |
| XXL-VLDL-TG | 303 | 1183 (3.7e-104) | 881.8 (6.6e-58) | 393.7 (0.00034) | 372.7 (0.0039) |
| LDL/IDL traits | | | | | |
| LDL-C | 293 | 1198.7 (9.6e-110) | 938.8 (1.1e-68) | 1060.2 (2.1e-87) | 917.6 (1.5e-65) |
| LDL-D | 296 | 1325.2 (6.7e-130) | 747.9 (5.9e-41) | 879.1 (3.7e-59) | 1163.5 (4.6e-103) |
| S-LDL-C | 296 | 1195.3 (2.9e-108) | 706 (1.6e-35) | 1426 (4.1e-147) | 686.4 (4.8e-33) |
| S-LDL-L | 296 | 1054.7 (1.1e-85) | 608 (1e-23) | 1519.6 (2.2e-163) | 586.4 (2.5e-21) |
| S-LDL-P | 297 | 852.9 (3.6e-55) | 438.7 (1.6e-07) | 954.7 (4.5e-70) | 440.1 (1.3e-07) |
| M-LDL-C | 296 | 1210.9 (8e-111) | 396.2 (8.6e-05) | 409 (1.4e-05) | 398.9 (6e-05) |
| M-LDL-CE | 295 | 1204.3 (4.8e-110) | 350.8 (0.014) | 361.7 (0.0048) | 351.3 (0.013) |
| M-LDL-L | 296 | 1212 (5.3e-111) | 370 (0.0022) | 392.3 (0.00015) | 371.6 (0.0019) |
| M-LDL-P | 297 | 1125.4 (1.2e-96) | 623.9 (2.3e-25) | 911.4 (1.3e-63) | 582.4 (9.6e-21) |
| M-LDL-PL | 299 | 1172.5 (1.2e-103) | 399.3 (9.1e-05) | 434.9 (4.5e-07) | 396.2 (0.00014) |
| L-LDL-C | 300 | 1174.6 (1.1e-103) | 325.5 (0.15) | 325.5 (0.15) | 325.5 (0.15) |
| L-LDL-CE | 299 | 1179.5 (9e-105) | 769.8 (3e-43) | 902.5 (7.7e-62) | 743.8 (8.4e-40) |
| L-LDL-FC | 295 | 1161 (5.8e-103) | 322.4 (0.13) | 323.2 (0.12) | 322.3 (0.13) |
| L-LDL-L | 300 | 1172.3 (2.6e-103) | 336.9 (0.07) | 349.6 (0.026) | 340.3 (0.055) |
| L-LDL-P | 300 | 1185.4 (2e-105) | 352.1 (0.021) | 378.4 (0.0014) | 355.4 (0.015) |
| L-LDL-PL | 296 | 1155.2 (9.8e-102) | 343.2 (0.031) | 360.1 (0.0063) | 344.5 (0.027) |
| IDL-C | 296 | 1181.7 (4.9e-106) | 426.5 (9.8e-07) | 427.6 (8.3e-07) | 427.7 (8.1e-07) |
| IDL-FC | 298 | 1096.5 (9.9e-92) | 986.9 (1.1e-74) | 1075.8 (1.9e-88) | 975.4 (6.1e-73) |

*Continued on next page*

*Appendix 3—table 8 continued*

| Trait | DF | ApoA1 | ApoB | TG | Subfraction |
|---|---|---|---|---|---|
| IDL-L | 296 | 1176.1 (4e-105) | 516.7 (3.3e-14) | 531 (1.4e-15) | 521.4 (1.2e-14) |
| IDL-P | 297 | 1094.8 (9.5e-92) | 910.9 (1.5e-63) | 1103.9 (3.5e-93) | 890.2 (1.6e-60) |
| IDL-PL | 297 | 1107.8 (8.3e-94) | 798.9 (1.3e-47) | 931.6 (1.3e-66) | 785.6 (8.6e-46) |
| IDL-TG | 302 | 1060.8 (5.4e-85) | 1052.1 (1.2e-83) | 1092.6 (5.6e-90) | 1118.3 (4.7e-94) |
| HDL traits | | | | | |
| HDL-C | 298 | 318.7 (0.2) | 336.3 (0.063) | 329.1 (0.1) | 318.6 (0.2) |
| HDL-D | 300 | 637.4 (1.9e-26) | 1156.6 (9.1e-101) | 2305.2 (1.3e-305) | 1183.8 (3.5e-105) |
| S-HDL-L | 299 | 1597.7 (4.8e-176) | 1222.5 (8.2e-112) | 1916.4 (1.5e-233) | 1057 (3.1e-85) |
| S-HDL-P | 299 | 1666.8 (2.5e-188) | 1249.4 (2.9e-116) | 2146.5 (3.4e-276) | 1103.3 (1.6e-92) |
| S-HDL-TG | 299 | 899 (2.5e-61) | 464.9 (2.4e-09) | 464.5 (2.6e-09) | 457.6 (9.2e-09) |
| M-HDL-C | 299 | 1145.2 (3.2e-99) | 768.2 (4.9e-43) | 951.8 (4e-69) | 786.8 (1.5e-45) |
| M-HDL-CE | 299 | 1201.9 (2e-108) | 1183.9 (1.7e-105) | 2139.7 (6.4e-275) | 843.9 (1.9e-53) |
| M-HDL-FC | 298 | 881.1 (5.6e-59) | 1252 (5.5e-117) | 1989.1 (2.4e-247) | 660.1 (1.8e-29) |
| M-HDL-L | 299 | 1059 (1.5e-85) | 766.4 (8.7e-43) | 920.6 (1.7e-64) | 672.5 (8.6e-31) |
| M-HDL-P | 298 | 990.2 (3.5e-75) | 760.4 (3.4e-42) | 1027.6 (6.2e-81) | 613.7 (4.7e-24) |
| M-HDL-PL | 295 | 929.5 (8.3e-67) | 763.9 (2.7e-43) | 1057.2 (2.3e-86) | 588.3 (1.1e-21) |
| L-HDL-C | 299 | 579.3 (4.1e-20) | 623.2 (5.7e-25) | 639.6 (7.3e-27) | 617.8 (2.3e-24) |
| L-HDL-CE | 299 | 612.2 (1e-23) | 650.7 (3.6e-28) | 690.4 (5.5e-33) | 644 (2.2e-27) |
| L-HDL-FC | 308 | 581.7 (4.4e-19) | 857.5 (2.6e-53) | 1213.3 (1.4e-107) | 915.8 (1.3e-61) |
| L-HDL-L | 299 | 655.9 (8.7e-29) | 747.7 (2.6e-40) | 670.7 (1.4e-30) | 713.2 (7.5e-36) |
| L-HDL-P | 298 | 591.3 (1.5e-21) | 934 (9.9e-67) | 1269.7 (6.2e-120) | 956.8 (3.9e-70) |
| L-HDL-PL | 299 | 580 (3.4e-20) | 863.5 (3.3e-56) | 1262.4 (2.1e-118) | 891.8 (2.8e-60) |
| XL-HDL-C | 298 | 475.3 (2.7e-10) | 734 (1e-38) | 976.1 (4.9e-73) | 554 (1.3e-17) |
| XL-HDL-CE | 299 | 472.9 (5.4e-10) | 736.9 (6.7e-39) | 1117.4 (9e-95) | 517.5 (6.5e-14) |
| XL-HDL-FC | 295 | 527.8 (2.1e-15) | 1182.8 (1.6e-106) | 2169.4 (3.1e-282) | 677.3 (4.3e-32) |
| XL-HDL-L | 298 | 555.2 (9.6e-18) | 701.2 (1.6e-34) | 1014 (7.9e-79) | 775.3 (3.4e-44) |
| XL-HDL-P | 300 | 578.9 (6.3e-20) | 744.5 (1.1e-39) | 1015.5 (1.6e-78) | 751.3 (1.4e-40) |
| XL-HDL-PL | 306 | 604.9 (7.8e-22) | 1153.9 (1.4e-98) | 1899 (1.5e-227) | 909.3 (3.7e-61) |
| XL-HDL-TG | 300 | 702.2 (2.8e-34) | 779.8 (2.2e-44) | 1140.8 (3.2e-98) | 1399.2 (3.7e-141) |

## Appendix 4

### Diagnostic plots and the genetic markers

As mentioned above, RAPS is more robust against invalid instruments than other statistical methods for univariable MR, but it still needs the InSIDE assumption to be approximately satisfied. *Zhao et al., 2019b* described two diagnostic plots RAPS that checks whether there is clear evidence that the InSIDE assumption is violated. Here, we report these plots for HDL-C and M-HDL-P in different studies (*Appendix 4—figures 1* and *2*). Notice that a lack of evidence to falsify the InSIDE assumption does not mean that it is true.

### S-HDL-P

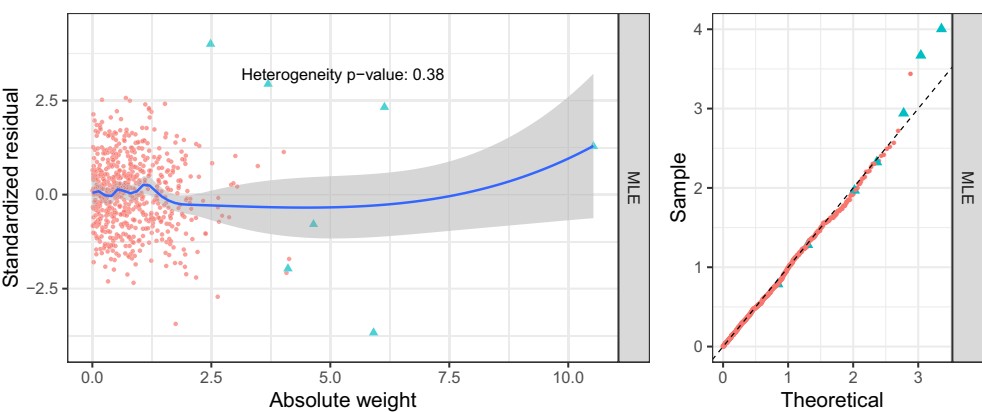

**Appendix 4—figure 1.** Diagnostic plots for S-HDL-P (selection: Davis; exposure: Kettunen; outcome: UK Biobank).

### M-HDL-P

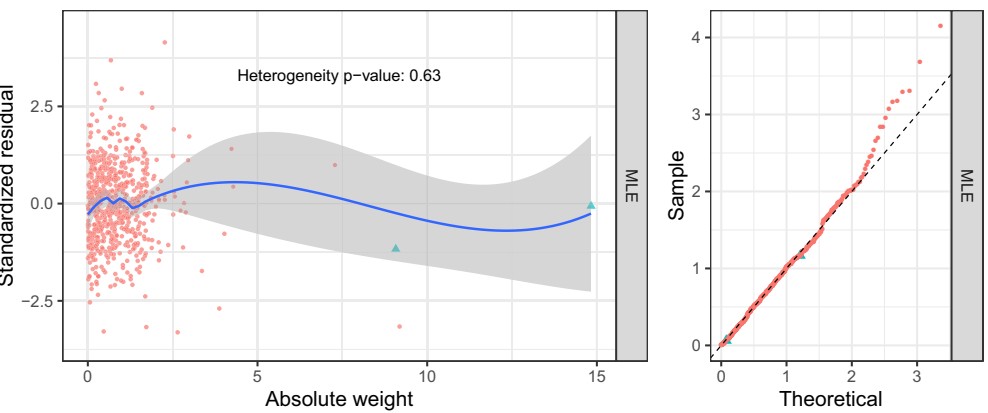

**Appendix 4—figure 2.** Diagnostic plots for M-HDL-P (selection: Davis; exposure: Kettunen; outcome: UK Biobank).

### Genetic markers for M-HDL-P and S-HDL-P

We can further assess the validity of the InSIDE assumption for M-HDL-P and S-HDL-P but examining the associations of their genetic instruments with the traditional lipid risk factors and other subfraction traits. We meta-analyzed the summary results in the two lipidome GWAS (Davis and Kettunen)

and obtained SNPs that are associated with S-HDL-P and M-HDL-P (p-value $\leq 5 \times 10^{-8}$; the results are LD-clumped). The next two Tables show some information about these genetic markers and their associations with other traits (*Appendix 4—table 1* and *2*).

*Appendix 4—figures 3* and *4* shows how adjusting for LDL-C and TG changes the effects of the selected SNPs for S-HDL-P and M-HDL-P on CAD. The adjusted effect on CAD is obtained by original effect on CAD – 0.45 * effect on LDL-C – 0.25 * effect on TG. After the adjustment, the associations of the genetic variants with CAD generally became closer to the fitted lines that correspond to the estimated effects of S-HDL-P and M-HDL-P.

**Appendix 4—table 1.** List of SNPs associated with M-HDL-P.

| SNP | Chr | Gene | S-HDL-P | M-HDL-P | L-HDL-P | XL-HDL-P | HDL-C | LDL-C | TG | CAD |
|---|---|---|---|---|---|---|---|---|---|---|
| rs11208004 | 1 | DOCK7 | -0.039 ** | -0.075 *** | -0.015 | -0.002 | -0.015 ** | -0.050 *** | -0.069 *** | -0.012 |
| rs4846913 | 1 | GALNT2 | -0.000 | -0.061 *** | -0.062 *** | -0.023 . | -0.055 *** | -0.006 | -0.044 *** | -0.025 . |
| rs2126259 | 8 | LOC157273 | -0.066 *** | -0.082 *** | -0.063 ** | -0.025 . | -0.075 *** | -0.063 *** | -0.016 . | -0.004 |
| rs2083637 | 8 | LPL | -0.001 | -0.058 *** | -0.092 *** | -0.053 ** | -0.105 *** | -0.008 | -0.108 *** | -0.047 ** |
| rs10468017 | 15 | ALDH1A2/ LIPC | -0.096 *** | -0.060 *** | -0.209 *** | -0.202 *** | -0.118 *** | -0.002 | -0.038 *** | -0.013 |
| rs247616 | 16 | CETP | -0.058 *** | -0.121 *** | -0.198 *** | -0.129 *** | -0.243 *** | -0.055 *** | -0.039 *** | -0.044 ** |
| rs1943973 | 18 | LIPG | -0.022 | -0.108 *** | -0.104 *** | -0.078 *** | -0.077 *** | -0.024 ** | -0.009 | -0.016 |
| rs737337 | 19 | DOCK6 | -0.047 . | -0.087 *** | -0.081 ** | -0.058 * | -0.056 *** | -0.007 | -0.011 | -0.038 . |
| rs769449 | 19 | APOE | -0.016 | -0.078 *** | -0.071 *** | -0.015 | -0.064 *** | -0.214 *** | -0.042 *** | -0.085 *** |
| rs7679 | 20 | PCIF1/PLTP | -0.188 *** | -0.071 *** | -0.129 *** | -0.152 *** | -0.059 *** | -0.009 | -0.051 *** | -0.025 . |

**Appendix 4—table 2.** List of SNPs associated with S-HDL-P.

| SNP | Chr | Gene | S-HDL-P | M-HDL-P | L-HDL-P | XL-HDL-P | HDL-C | LDL-C | TG | CAD |
|---|---|---|---|---|---|---|---|---|---|---|
| rs780094 | 2 | GCKR | -0.074 *** | -0.034 * | -0.04 ** | -0.034 * | -0.011 . | -0.021 ** | -0.110 *** | -0.005 |
| rs10935473 | 3 | ST3GAL6-AS1 | -0.052 *** | -0.014 | -0.029 . | -0.031 * | -0.009 . | -0.003 | -0.005 | -0.007 |
| rs4936363 | 11 | SIK3 | -0.064 *** | -0.046 ** | -0.019 | -0.006 | -0.034 ** | -0.018 . | -0.043 *** | -0.022 |
| rs2043085 | 15 | ALDH1A2/ LIPC | -0.092 *** | -0.056 *** | -0.202 *** | -0.197 *** | -0.106 *** | -0.003 | -0.033 *** | -0.008 |
| rs1800588 | 15 | ALDH1A2/ LIPC | -0.106 *** | -0.050 ** | -0.215 *** | -0.212 *** | -0.114 *** | -0.002 | -0.044 *** | -0.015 |
| rs289714 | 16 | CETP | -0.077 *** | -0.122 *** | -0.162 *** | -0.102 *** | -0.214 *** | -0.036 *** | -0.035 *** | -0.012 |
| rs6065904 | 20 | PLTP | -0.171 *** | -0.060 *** | -0.127 *** | -0.149 *** | -0.052 *** | -0.008 | -0.040 *** | -0.022 . |

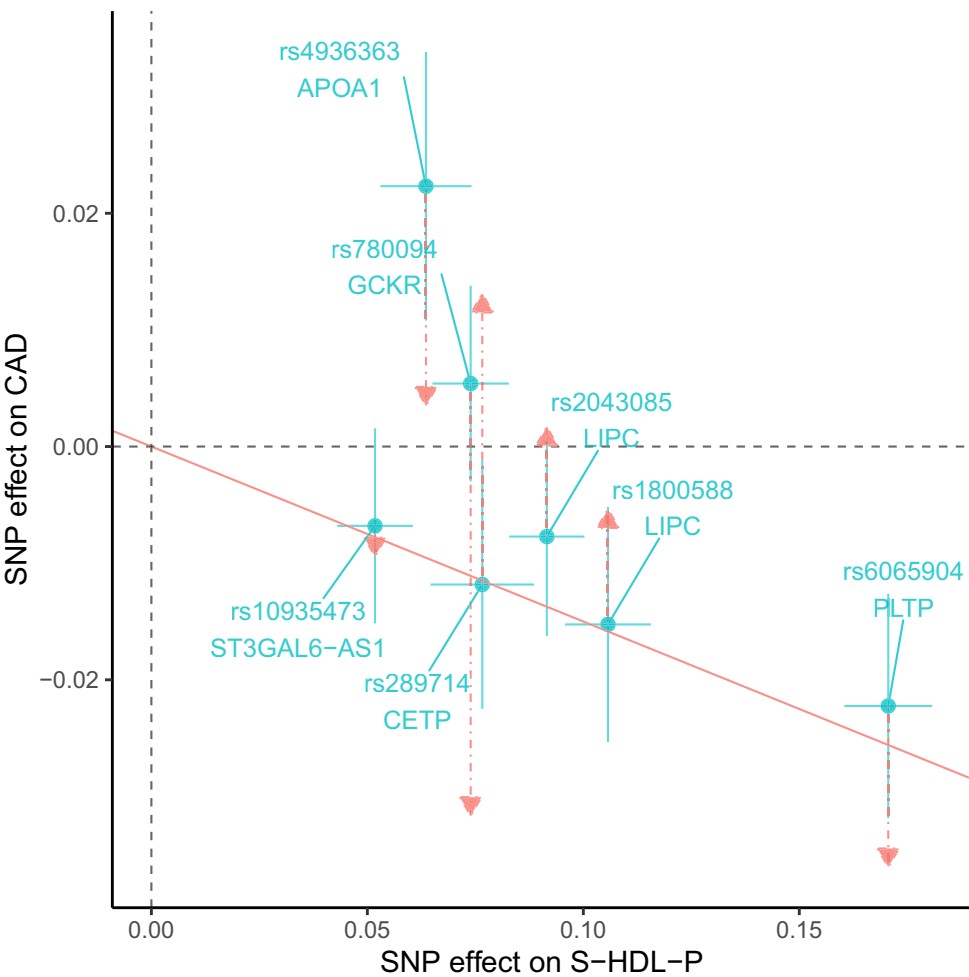

**Appendix 4—figure 3.** Scatter-plots for S-HDL-P with the effects on CAD adjusted for LDL-C and TG. Red lines correspond the fitted effects of S-HDL-P in multivariable MR.

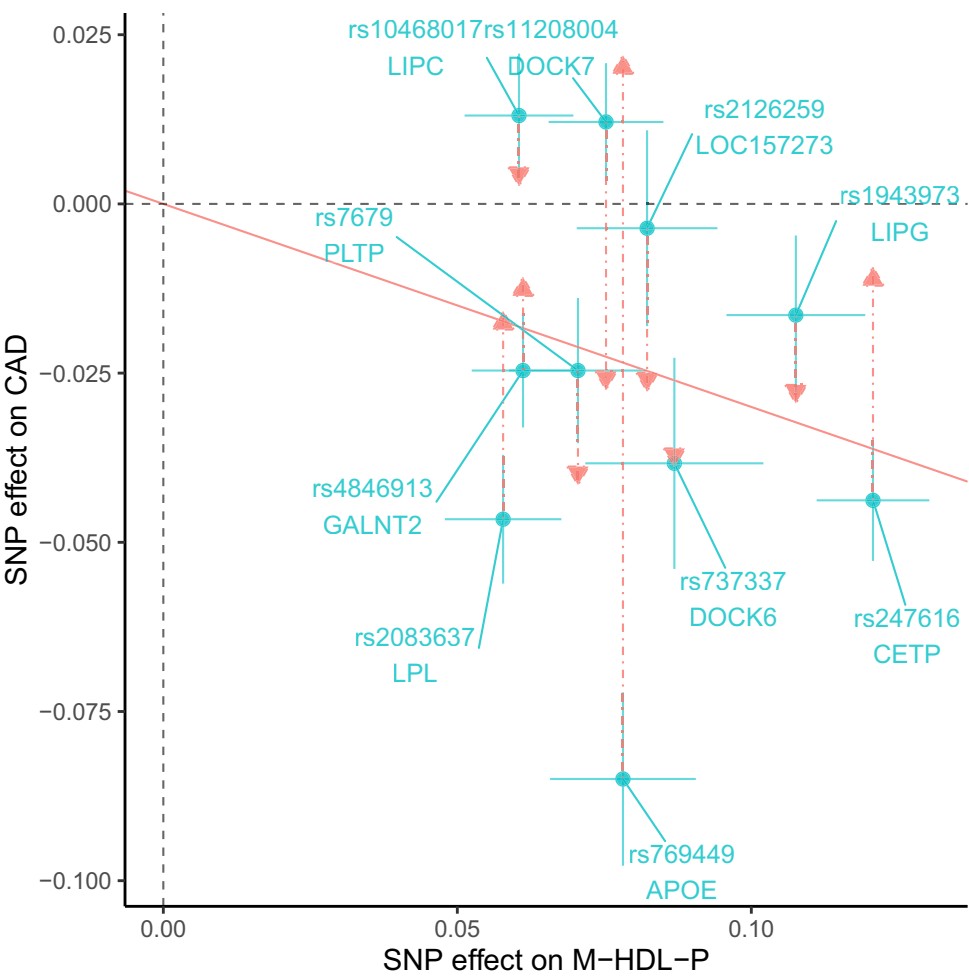

**Appendix 4—figure 4.** Scatter-plots for M-HDL-P with the effects on CAD adjusted for LDL-C and TG. Red lines correspond the fitted effects of M-HDL-P in multivariable MR.

## Gene expression

Here we provide evidence of variant-gene associations from Quantatitive Trait Locus (QTL) analyses in the GTEx project (*Appendix 4—table 3*).

**Appendix 4—table 3.** Tissue-specific gene expressions associated with the 4 discovered genetic markers in the GTEx project.

| SNP.Id | Type | Gene. Symbol | Variant.Id | p value | Effect | Tissue |
|---|---|---|---|---|---|---|
| rs838880 | eQTL | SCARB1 | chr12_124777047_C_T_b38 | 1.5E-08 | -0.20 | Cells - Cultured fibroblasts |
| rs838880 | sQTL | SCARB1 | chr12_124777047_C_T_b38 | 4.1E-06 | -0.34 | Testis |
| rs737337 | sQTL | DOCK6 | chr19_11236817_T_C_b38 | 3.8E-43 | 0.99 | Artery - Tibial |
| rs737337 | sQTL | DOCK6 | chr19_11236817_T_C_b38 | 6.4E-35 | 0.93 | Adipose - Subcutaneous |
| rs737337 | sQTL | DOCK6 | chr19_11236817_T_C_b38 | 6.4E-35 | 0.93 | Adipose - Subcutaneous |
| rs737337 | sQTL | DOCK6 | chr19_11236817_T_C_b38 | 1.6E-27 | 0.95 | Esophagus - Muscularis |
| rs737337 | sQTL | DOCK6 | chr19_11236817_T_C_b38 | 3.2E-20 | 1.10 | Colon - Sigmoid |

*Continued on next page*

*Appendix 4—table 3 continued*

| SNP.Id | Type | Gene.Symbol | Variant.Id | p value | Effect | Tissue |
|---|---|---|---|---|---|---|
| rs737337 | sQTL | DOCK6 | chr19_11236817_T_C_b38 | 1.1E-17 | 0.93 | Esophagus - Gastroesophageal Junction |
| rs737337 | sQTL | DOCK6 | chr19_11236817_T_C_b38 | 1.8E-09 | 0.81 | Artery - Coronary |
| rs737337 | sQTL | DOCK6 | chr19_11236817_T_C_b38 | 1.2E-07 | -0.49 | Thyroid |
| rs737337 | sQTL | KANK2 | chr19_11236817_T_C_b38 | 4.4E-07 | 0.43 | Artery - Tibial |
| rs737337 | sQTL | KANK2 | chr19_11236817_T_C_b38 | 3.5E-06 | 0.55 | Heart - Left Ventricle |
| rs2943641 | eQTL | IRS1 | chr2_226229029_T_C_b38 | 1.4E-16 | -0.30 | Adipose - Subcutaneous |
| rs2943641 | eQTL | IRS1 | chr2_226229029_T_C_b38 | 6.1E-12 | -0.23 | Adipose - Visceral (Omentum) |
| rs2943641 | eQTL | RP11-395N3.2 | chr2_226229029_T_C_b38 | 3.5E-09 | -0.23 | Adipose - Subcutaneous |
| rs2943641 | eQTL | RP11-395N3.1 | chr2_226229029_T_C_b38 | 2.1E-07 | -0.23 | Adipose - Subcutaneous |
| rs2943641 | eQTL | RP11-395N3.2 | chr2_226229029_T_C_b38 | 2.3E-06 | -0.19 | Adipose - Visceral (Omentum) |
| rs6065904 | eQTL | PLTP | chr20_45906012_G_A_b38 | 4.4E-22 | -0.27 | Muscle - Skeletal |
| rs6065904 | eQTL | PLTP | chr20_45906012_G_A_b38 | 1.6E-16 | -0.27 | Adipose - Subcutaneous |
| rs6065904 | eQTL | PLTP | chr20_45906012_G_A_b38 | 1.2E-15 | -0.28 | Adipose - Visceral (Omentum) |
| rs6065904 | eQTL | PLTP | chr20_45906012_G_A_b38 | 3.2E-15 | -0.42 | Heart - Atrial Appendage |
| rs6065904 | eQTL | PLTP | chr20_45906012_G_A_b38 | 7.2E-14 | -0.25 | Artery - Tibial |
| rs6065904 | eQTL | PLTP | chr20_45906012_G_A_b38 | 1.8E-12 | -0.27 | Nerve - Tibial |
| rs6065904 | eQTL | PLTP | chr20_45906012_G_A_b38 | 7.3E-12 | -0.26 | Esophagus - Muscularis |
| rs6065904 | eQTL | PLTP | chr20_45906012_G_A_b38 | 2.0E-11 | -0.29 | Colon - Transverse |
| rs6065904 | eQTL | PLTP | chr20_45906012_G_A_b38 | 4.1E-11 | -0.32 | Colon - Sigmoid |
| rs6065904 | eQTL | PLTP | chr20_45906012_G_A_b38 | 1.2E-09 | -0.26 | Artery - Aorta |
| rs6065904 | eQTL | PLTP | chr20_45906012_G_A_b38 | 4.2E-09 | -0.29 | Heart - Left Ventricle |
| rs6065904 | eQTL | PLTP | chr20_45906012_G_A_b38 | 5.0E-09 | -0.22 | Thyroid |
| rs6065904 | eQTL | PLTP | chr20_45906012_G_A_b38 | 1.7E-08 | -0.29 | Stomach |
| rs6065904 | eQTL | PLTP | chr20_45906012_G_A_b38 | 4.3E-08 | -0.24 | Lung |
| rs6065904 | eQTL | NEURL2 | chr20_45906012_G_A_b38 | 6.6E-08 | -0.26 | Adipose - Subcutaneous |
| rs6065904 | eQTL | PLTP | chr20_45906012_G_A_b38 | 6.8E-08 | -0.33 | Liver |
| rs6065904 | eQTL | CTSA | chr20_45906012_G_A_b38 | 4.0E-07 | -0.14 | Nerve - Tibial |
| rs6065904 | eQTL | PLTP | chr20_45906012_G_A_b38 | 5.3E-07 | -0.37 | Spleen |
| rs6065904 | sQTL | NEURL2 | chr20_45906012_G_A_b38 | 5.6E-07 | -0.26 | Adipose - Visceral (Omentum) |
| rs6065904 | eQTL | PLTP | chr20_45906012_G_A_b38 | 8.9E-07 | -0.46 | Small Intestine - Terminal Ileum |
| rs6065904 | eQTL | RP3-337O18.9 | chr20_45906012_G_A_b38 | 1.8E-06 | -0.22 | Adipose - Subcutaneous |
| rs6065904 | eQTL | WFDC3 | chr20_45906012_G_A_b38 | 2.9E-06 | -0.31 | Nerve - Tibial |
| rs6065904 | eQTL | DNTTIP1 | chr20_45906012_G_A_b38 | 3.1E-06 | -0.17 | Artery - Tibial |
| rs6065904 | eQTL | WFDC3 | chr20_45906012_G_A_b38 | 4.5E-06 | -0.27 | Skin - Sun Exposed (Lower leg) |
| rs6065904 | eQTL | SNX21 | chr20_45906012_G_A_b38 | 4.8E-06 | -0.15 | Esophagus - Muscularis |
| rs6065904 | eQTL | WFDC3 | chr20_45906012_G_A_b38 | 8.9E-06 | -0.27 | Skin - Not Sun Exposed (Suprapubic) |
| rs6065904 | eQTL | DNTTIP1 | chr20_45906012_G_A_b38 | 1.0E-05 | -0.14 | Nerve - Tibial |
| rs6065904 | eQTL | PLTP | chr20_45906012_G_A_b38 | 1.1E-05 | -0.27 | Prostate |
| rs6065904 | eQTL | PLTP | chr20_45906012_G_A_b38 | 1.3E-05 | -0.26 | Pituitary |

*Continued on next page*

*Appendix 4—table 3 continued*

| SNP.Id | Type | Gene.Symbol | Variant.Id | p value | Effect | Tissue |
|---|---|---|---|---|---|---|
| rs6065904 | eQTL | PLTP | chr20_45906012_G_A_b38 | 1.4E-05 | -0.21 | Esophagus - Gastroesophageal Junction |
| rs6065904 | eQTL | SNX21 | chr20_45906012_G_A_b38 | 1.5E-05 | -0.16 | Esophagus - Mucosa |
| rs6065904 | eQTL | SNX21 | chr20_45906012_G_A_b38 | 1.7E-05 | -0.23 | Colon - Sigmoid |
| rs6065904 | eQTL | SNX21 | chr20_45906012_G_A_b38 | 1.7E-05 | -0.17 | Thyroid |
| rs6065904 | eQTL | PLTP | chr20_45906012_G_A_b38 | 2.6E-05 | -0.21 | Breast - Mammary Tissue |
| rs6065904 | eQTL | WFDC3 | chr20_45906012_G_A_b38 | 2.9E-05 | -0.23 | Artery - Tibial |
| rs6065904 | eQTL | NEURL2 | chr20_45906012_G_A_b38 | 3.2E-05 | -0.21 | Thyroid |
| rs6065904 | eQTL | PLTP | chr20_45906012_G_A_b38 | 3.7E-05 | -0.17 | Testis |
| rs6065904 | eQTL | CTSA | chr20_45906012_G_A_b38 | 4.4E-05 | -0.11 | Skin - Not Sun Exposed (Suprapubic) |
| rs6065904 | eQTL | WFDC3 | chr20_45906012_G_A_b38 | 5.8E-05 | -0.23 | Muscle - Skeletal |
| rs6065904 | eQTL | NEURL2 | chr20_45906012_G_A_b38 | 8.2E-05 | -0.27 | Heart - Atrial Appendage |
| rs6065904 | eQTL | SNX21 | chr20_45906012_G_A_b38 | 8.4E-05 | -0.17 | Artery - Aorta |
| rs6065904 | eQTL | NEURL2 | chr20_45906012_G_A_b38 | 9.5E-05 | -0.24 | Artery - Aorta |
| rs6065904 | eQTL | WFDC3 | chr20_45906012_G_A_b38 | 9.5E-05 | -0.31 | Artery - Aorta |
| rs6065904 | eQTL | RP3-337O18.9 | chr20_45906012_G_A_b38 | 9.5E-05 | -0.29 | Heart - Atrial Appendage |
| rs6065904 | eQTL | PLTP | chr20_45906012_G_A_b38 | 1.2E-04 | -0.15 | Skin - Sun Exposed (Lower leg) |
| rs6065904 | eQTL | WFDC13 | chr20_45906012_G_A_b38 | 1.5E-04 | 0.28 | Esophagus - Muscularis |
| rs6065904 | eQTL | DNTTIP1 | chr20_45906012_G_A_b38 | 2.1E-04 | -0.12 | Cells - Cultured fibroblasts |
| rs6065904 | sQTL | ZNF335 | chr20_45906012_G_A_b38 | 3.3E-11 | -0.65 | Testis |
| rs6065904 | sQTL | ACOT8 | chr20_45906012_G_A_b38 | 1.3E-09 | 0.58 | Heart - Left Ventricle |
| rs6065904 | sQTL | PLTP | chr20_45906012_G_A_b38 | 4.5E-08 | -0.32 | Whole Blood |
| rs6065904 | sQTL | PLTP | chr20_45906012_G_A_b38 | 4.8E-08 | 0.53 | Spleen |
| rs6065904 | sQTL | ACOT8 | chr20_45906012_G_A_b38 | 1.3E-07 | 0.42 | Esophagus - Mucosa |
| rs6065904 | sQTL | ACOT8 | chr20_45906012_G_A_b38 | 2.6E-07 | 0.49 | Heart - Atrial Appendage |
| rs6065904 | sQTL | CTSA | chr20_45906012_G_A_b38 | 1.0E-06 | -0.41 | Artery - Aorta |
| rs6065904 | sQTL | ACOT8 | chr20_45906012_G_A_b38 | 1.2E-06 | 0.33 | Nerve - Tibial |
| rs6065904 | sQTL | ACOT8 | chr20_45906012_G_A_b38 | 1.2E-06 | 0.67 | Brain - Spinal cord (cervical c-1) |
| rs6065904 | sQTL | TNNC2 | chr20_45906012_G_A_b38 | 2.1E-06 | 0.54 | Brain - Cerebellum |
| rs6065904 | sQTL | ACOT8 | chr20_45906012_G_A_b38 | 2.1E-06 | 0.54 | Brain - Cerebellum |
| rs6065904 | sQTL | WFDC3 | chr20_45906012_G_A_b38 | 5.5E-06 | 0.23 | Skin - Sun Exposed (Lower leg) |
| rs6065904 | sQTL | WFDC3 | chr20_45906012_G_A_b38 | 9.4E-06 | -0.28 | Skin - Not Sun Exposed (Suprapubic) |

