## [Decision Letter]

**Acceptance summary:**

This paper is a welcome extension of the use of Mendelian Randomization in the evaluation of the role of lipoprotein sub-fractions. It is necessarily complex. The confirmation of the atherogenicity of summary estimates of LDL and VLDL is useful as are new insights into the protective role of HDL sub-fractions.

**Decision letter after peer review:**

Thank you for submitting your article "A Mendelian randomization study of the role of lipoprotein subfractions in coronary artery disease" for consideration by *eLife*. Your article has been reviewed by 4 peer reviewers, one of whom is a member of our Board of Reviewing Editors, and the evaluation has been overseen by Matthias Barton as the Senior Editor. The following individuals involved in review of your submission have agreed to reveal their identity: David Sullivan (Reviewer #1); Jie Zheng (Reviewer #4).

The reviewers have discussed the reviews with one another and the Reviewing Editor has drafted this decision to help you prepare a revised submission.

As the editors have judged that your manuscript is of interest, but if as described below additional experiments are required before it is published, we would like to draw your attention to changes in our revision policy that we have made in response to COVID-19 (https://elifesciences.org/articles/57162). First, because many researchers have temporarily lost access to the labs, we will give authors as much time as they need to submit revised manuscripts. We are also offering, if you choose, to post the manuscript to bioRxiv (if it is not already there) along with this decision letter and a formal designation that the manuscript is "in revision at *eLife*". Please let us know if you would like to pursue this option. (If your work is more suitable for medRxiv, you will need to post the preprint yourself, as the mechanisms for us to do so are still in development.)

Summary:

Zhao et al. apply an array of MR approaches to attempt to disentangle the contributions of different lipoprotein subclasses to coronary artery disease risk. They also try to answer the debate about "what matters the most", the lipid content of the lipoproteins or the number of particles. They conclude that LDL and VLDL sub-fractions have a universally adverse effect on coronary artery disease and myocardial infarction and that small dense LDL is not more atherogenic. HDL sub-fractions are heterogeneous in their effects with medium sized particle being protective and this supports the HDL function hypothesis.

Essential revisions:

There are aspects of the study design that could be improved and discussed more critically because some of the interpretations are not straightforward. Overall comprehension of the paper would be greatly aided by a more detailed consideration of the relationship between the different lipoprotein classes and subclasses, and a clear explanation of parameters of interest.

Essential are a substantially clearer description of the results (requiring major revision of the manuscript), stronger justification of the statistical approach taken and explanation as to why results for some fractions are not presented. A preferred alternative is to present results for all fractions even if only in supplementary tables.

1. In the abstract and text please modify the use of the terms positive and negative effects. Clinicians interpret positive as meaning beneficial when referring to outcomes eg in RCTs and negative as ineffective or harmful. In this paper positive and negative appropriately refer to the directions of the associations found. To remove any potential ambiguity for readers please consider using positive (harmful) and negative (protective) or a similar rewording of your choice when these terms first appear in the abstract and main text.

2. The analyses presented are laudably thorough in detail, using different combinations of datasets in which variants to be used for analysis are identified in different summary statistics from those used for the MR itself. There does not appear to have been a prior hypothesis to this work. Rather, it seems that all possible analyses have been conducted and the results then mined for potentially noteworthy findings. While such analyses are useful in themselves the conclusions drawn are suspect and, in places, contradicted by other parts of the authors' own results. The main results also rest on a new method which has yet to be established as valid or workable in this context – GRAPPLE has only recently been described in a bioRXiV preprint, applied to traits with well over 100 independent genome-wide associations, and separate publication resting almost entirely on results using it, applied to traits with much less well-powered GWAS, is premature at best.

3. Genetic correlation analysis – The genetic correlation analysis seems to be stand alone to other MR analyses. The listed motivation of this analysis is to check whether the MR findings are independent to each other. However, it is clear that all these lipid sub-fractions are highly correlated to each other (genetically and observationally), especially for VLDL and LDL sub-fractions. So the rationale of linking TG with VLDL was not tested (although biologically VLDL is the main particle carrying TG in fasting). Even in a clustering point of view, it is hard to split the lipid sub-fraction into three groups: VLDL-TG, IDL/LDL-LDL-C and HDL-HDL-C.

Also some of the rg estimates are missing in Figure 1, e.g. TG is related to the key findings but not included in this genetic correlation analysis.

Also, in a method point of view there are some new genetic correlation methods coming out recently https://www.nature.com/articles/s41588-020-0653-y, which could be considered.

4. Instrument selection – From a practical point of view, It is tricky to jointly model TG with VLDL subtypes characteristics in MVMR because it is unlikely that one would have genetic instruments that strongly predict each exposure once adjusting for the others. Therefore, weak instrument bias is probably an issue here. This is worrying as weak instrument bias in MVMR is not necessarily conservative. At a minimum please provide the conditional F statistics for the MVMR model for each exposure.

5. The authors present a large amount of results for (unadjusted) univariate MR. Given the known highly pleiotropic nature of variants affecting lipoproteins and their sub-fractions, presentation of these in the main figure adds little and obscures the main results of potential note. The full results could be presented as supplementary data, but the main figures should be restricted to multivariable MR.

6. Independent effects of lipid sub-fractions? multivariable MR (MVMR)

One of the key issue for lipids sub-fractions (and other metabolites such as fatty acids) is all the sub-fractions are highly correlated to each other. It is almost impossible to include all of the sub-fractions in the same MVMR model. Some methods considering reduce the dimensionality to solve this issue. In this study, the author used a different approach, which split the lipid sub-fractions into three classes, HDL, LDL and TG related. In this way, the MVMR could be conducted with limited number of variables. However, a lot of SNPs are associated with two or more lipid phenotypes, e.g. the CETP example the author provided in Table 3, the SNP is associated with HDL-C, LDL-C and TG. Even in a MVMR model controlling for LDL-C and TG, it is still hard to prove that the highlighted M-HDL-P finding showed an independent effect on CAD.

Also, more and more studies considered APOB and APOA1 in the MVMR model (e.g. https://journals.plos.org/plosmedicine/article/comments?id=10.1371/journal.pmed.1003062), so at least worth including APOB/APOA1 in the MVMR model.

7. VLDL MR results – For VLDL results, some of them showed negative effects (CIs did not cross OR=1) in the MVMR model but positive effect in UVMR model, which seems complex and did not fit with the simple interpretation that "VLDL subfraction traits had uniformly positive effect on coronary artery disease" mentioned in the abstract. Can the authors discuss this in a bit more details and try to find the reason?

8. There are multiple lipidomics measures which are available from the NMR platform, but which are not analyzed, including critical ones such as M-HDL-TG. This needs explanation and/or rectification, since complete data would address multiple points. For instance, the authors dismiss the positive association of S-HDL-TG as being confounded by correlation with VLDL fractions, despite VLDL fractions not having any independent positive effect on CAD in the multivariable analysis. Results for M-HDL-TG would be very informative, and would avoid the possibility that erroneous conclusions are drawn simply because contrary data is missing.

9. It is unclear that it is appropriate to adjust for overall LDL-C when analyzing HDL sub-fractions. Attempting to analyze sub-fractions of one class alongside an aggregate measure of another (which is correlated with aggregate measures of the first) seems suspect and at the very least requires detailed and careful justification.

10. The introduction anticipates the resolution of conflicting findings in relation to LDL size. This is dealt with in discussion, but it needs to be be highlighted beforehand in results. Since levels of different sized LDL and VLDL particles are strongly correlated, a statement that different sizes are not differently atherogenic is not supported by the data. To draw this conclusions would require multivariable MR in which the different size classes were adjusted for each other. In fact, the statement is directly contradicted by the clear associations of VLDL and (particularly) LDL diameter with CAD.

11. Similarly the statement that medium and small HDL particles are protective is contradicted by the fact that HDL particle diameter shows no association at all with CAD risk.

12. Selection of lipid sub-fractions – 82 lipid sub-fractions were selected but it is not directly clear why these fractions (but not others) were selected. It will be helpful to have a DAG to explain the inclusion and exclusion criteria. There is an even more fundamental question: Does this data justify a re-classification of NMR lipoprotein subclasses? Most people would agree that 82 is a few too many. Can the authors nominate a rational condensation of NMR lipoprotein sub-fraction data into a handful of independently predictive parameters with putative mechanisms? Ideally, these would be targets for therapy and indicators of response.

13. HDL MR results – For S-HDL-TG, this is a very complex case and it is doubtful if the current setting can distinguish its effect driven by HDL from its effect driven by TG. I personally think its strong genetic correlation with VLDL (Figure 1) implies that the effect of S-HDL-TG on CAD is driven by TG. However, the MVMR adjusted TG showed similar effect estimate, which suggest the effect is independent to TG (as mentioned about, the reviewer is not sure the MVMR model can prove independent effect). The authors may need to consider integrating some biological information of each instruments been used here to dig into the button of this case study. For example, whether each of instrument colocalized with the expression level of the cis gene in multiple tissues. For the colocalized genes, are they related to HDL pathway or TG pathway etc. This could be very complex so the reviewer suggests to explain the results with caution.

For M-HDL-P, it showed some correlations with HDL-C and APOA1, so hard to say it is not correlated with HDL-C at all.

14. The identification of a leading role for mid-sized HDL particles presents a tantalizing opportunity to link the latter finding with recent NMR studies of cholesterol efflux, but this is not pursued. It would be worth flagging this opportunity at least in the discussion if you cannot easily address it. How do the results (M-HDL-P etc) relate to NMR assessments of cholesterol efflux capacity (eg Direct estimation of HDL-mediated cholesterol efflux capacity from serum. Sanna Kuusisto, Michael V. Holmes, Pauli Ohukainen, Antti J. Kangas, Mari Karsikas, Mika Tiainen, Markus Perola, Veikko Salomaa, Johannes Kettunen, Mika Ala-Korpela doi: https://doi.org/10.1101/396929, now published in Clinical Chemistry doi: 10.1373/clinchem.2018.299222).

15. The multiple versions of the analysis using different combinations of datasets do provide worthwhile technical replication. However, these do *not* provide wider replication of the findings since any shortcomings in one analysis (e.g. in failing to fully address pleiotropy) will inevitably be present in another.

16. A failure to find evidence against the INSIDE assumption, on the basis of only a small number of SNPs, is a very weak basis for making a claim that there is not horizontal pleiotropy affecting LDL-C or TG. The arguments made by the authors on this point are very weak.

17. The issue of horizontal pleiotropy, especially as it applies to m-HDL-P in relation to the inSIDE assumptions for TG and LDL-C, is justified in discussion and acknowledged in limitations, but it is very difficult follow. For example, from table 3, TG is strongly associated with all traits apart from LOC157273, LIPG and DOCK6. Accordingly, the simple clinician is tempted to regard m-HDL-P as a surrogate for TG. Perhaps there needs to be greater clarity concerning the concepts of genetic correlation and "weak instrument bias".

18. Discussion – "Our results for the HDL sub-fractions support the conclusion that HDL-C is not the causally relevant biomarker." The MR estimate of S-HDL-TG, M-HLD-C and M-HDL-L showed the complexity of HDL-C on CAD. So this claim seems against the main finding of the manuscript.

[Editors' note: further revisions were suggested prior to acceptance, as described below.]

Thank you for resubmitting your work entitled "A Mendelian randomization study of the role of lipoprotein sub-fractions in coronary artery disease" for further consideration by *eLife*. Your revised article has been evaluated by a Reviewing Editor and a Senior Editor.

The manuscript has been improved but there are some remaining issues noted by the reviewers that need to be addressed, as outlined below:

You have made substantial efforts to restructure the manuscript (e.g. using genetic correlation as a filtering step in the new version), highlighted the key findings of HDL particle size on CAD as well as identified four potential causal genes linking HDL particle size with CAD. We are in general more convinced that the new findings from this study will bring good value for the existing argument for effect of HDL on CAD.

1. The statistical superiority of HDL over TG is largely a reflection of the greater intra-individual biological variability of the LATTER. This is really the crux of our concern. It seems likely that TG and HDL size represent a gene/environment interaction with a very large environmental component. We are concerned that environmental factors could distort a Mendelian Randomization perspective of this analysis in a way in which the statistics used are valid, but conclusions are not generalizable. This can be addressed in the discussion maybe as a limitation.

2. A sentence in the abstract which summarizes the key findings and potential value of the study would be helpful.

3. Methods, genetic correlation. The authors claim that genetic correlation "is generally different from epidemiological correlation that is estimated from cross-sectional data." By concept, the genetic correlation and phenotypic correlation are different. But in the metabolites case, they have very similar estimates! So please refine this statement to make sure you are just talking about the concept.

4. A very brief introduction about genome-wide MR will be helpful. e.g. did you do LD pruning or any other selection.

5. The genetic marker section. This is value added but the subtitle "genetic markers" is a slightly underselling the value here. You are trying to map variants to genes to inform causal genes that are linking HDL-C size with CAD. It may worth stating this in the subtitle. Also, to make this section more informative, you can try a formal Wald ratio + colocalization analysis to estimate the putative causal effects of these HDL size related genes on CAD (rather than just did a SNP lookup in different GWASs).

6. Table 2, better to show 95%CI rather than just SE.

7. Discussion "small and medium HDL particles appear to be positively correlated with HDL cholesterol and ApoA1, their genetic correlations are much smaller than 1, indicating the existence of independent biological pathways." This statement is too strong by just using a statistical approach. Better to say "indicating possible independent biological pathway(s)."

---

## [Author Response]

Essential revisions:There are aspects of the study design that could be improved and discussed more critically because some of the interpretations are not straightforward. Overall comprehension of the paper would be greatly aided by a more detailed consideration of the relationship between the different lipoprotein classes and subclasses, and a clear explanation of parameters of interest.Essential are a substantially clearer description of the results (requiring major revision of the manuscript), stronger justification of the statistical approach taken and explanation as to why results for some fractions are not presented. A preferred alternative is to present results for all fractions even if only in supplementary tables.

We would like to thank the referees and editors for carefully reviewing our manuscript and giving the constructive comments. We have implemented new analyses and substantially revised our manuscript. The main changes include:

1. We have made it clear in the Introduction that the goal of this study is to use genetic data to "discover lipoprotein subfractions that may be causal risk factors for CAD and MI in addition to the traditional lipid profile". To this end, we used the estimated genetic correlation with the traditional lipid traits to screen the subfraction measurements. This allows us to identify lipoprotein subfractions that may involve independent biological mechanisms before we look at the data about coronary artery disease.

2. We have redesigned our multivariable MR analyses, in which all the tradition lipid risk factors were included as exposures. Following the suggestion of a referee, we now considered two multivariable MR designs:

– In the first design, the exposures are TG, LDL-C, HDL-C, and the subfraction measurement under investigation;

– In the second design, the exposures are TG, ApoB, ApoA1, and the subfraction measurement under investigation.

The results of these two multivariable MR designs were largely comparable.

1. To make the results easier to interpret, we have moved the results of most of the univariable MR analyses to the Online Supplement. This also makes the Materials and methods in the main manuscript more straightforward.

2. We have revised the procedure of identifying genetic markers of interest. Because the main conclusion of our study is that HDL particle size may play a role in coronary artery disease, we selected SNPs that are associated with small or medium HDL subfractions and with CAD, but are not associated with LDL cholesterol or ApoB. Using this procedure, we identified four genetic markers for HDL size.

3. We have rewritten the Results section and the interpretation of the results in Discussion.

1. In the abstract and text please modify the use of the terms positive and negative effects. Clinicians interpret positive as meaning beneficial when referring to outcomes eg in RCTs and negative as ineffective or harmful. In this paper positive and negative appropriately refer to the directions of the associations found. To remove any potential ambiguity for readers please consider using positive (harmful) and negative (protective) or a similar rewording of your choice when these terms first appear in the abstract and main text.

Thank you. We have revised the usage of positive and negative effects in the manuscript.

2. The analyses presented are laudably thorough in detail, using different combinations of datasets in which variants to be used for analysis are identified in different summary statistics from those used for the MR itself. There does not appear to have been a prior hypothesis to this work. Rather, it seems that all possible analyses have been conducted and the results then mined for potentially noteworthy findings. While such analyses are useful in themselves the conclusions drawn are suspect and, in places, contradicted by other parts of the authors' own results. The main results also rest on a new method which has yet to be established as valid or workable in this context – GRAPPLE has only recently been described in a bioRXiV preprint, applied to traits with well over 100 independent genome-wide associations, and separate publication resting almost entirely on results using it, applied to traits with much less well-powered GWAS, is premature at best.

Thank you for the feedback. We had suspected that HDL particles may have a complicated and heterogeneous role prior to this work. For example, in a prior study (Zhao et al., Int J Epidemiol. 2019 Oct;48:1478-92), we concluded that "Further investigations are needed to demystify the observational and genetic associations between HDL-C and CAD." However, we did not want to restrict to this single hypothesis, because more information can be obtained and no real extra effort is needed by considering all lipid subfraction traits simultaneously. By being more agnostic, we also avoid the perils of selection/publication bias. This makes the current study slightly less powerful as we need to adjust for multiple testing, but we thought it is a price worth paying.

It is true that the GRAPPLE paper has not been published in a journal. On the other hand, we truly believe GRAPPLE has solved several issues in existing methods for multivariable MR and is more powerful. You are correct that the GWAS in this study are less well-powered, which is exactly the situation GRAPPLE is superior over other methods because it utilizes weak instruments efficiently and avoids weak instrument bias. So it would be a pity if we don’t use GRAPPLE here for this study. But we understand the concerns of relying the results on an unpublished methodological article and would understand if the editors want to put the publication of this manuscript on hold before the paper describing GRAPPLE finishes the peer-review process.

3. Genetic correlation analysis – The genetic correlation analysis seems to be stand alone to other MR analyses. The listed motivation of this analysis is to check whether the MR findings are independent to each other. However, it is clear that all these lipid sub-fractions are highly correlated to each other (genetically and observationally), especially for VLDL and LDL sub-fractions. So the rationale of linking TG with VLDL was not tested (although biologically VLDL is the main particle carrying TG in fasting). Even in a clustering point of view, it is hard to split the lipid sub-fraction into three groups: VLDL-TG, IDL/LDL-LDL-C and HDL-HDL-C.Also some of the rg estimates are missing in Figure 1, e.g. TG is related to the key findings but not included in this genetic correlation analysis.Also, in a method point of view there are some new genetic correlation methods coming out recently https://www.nature.com/articles/s41588-020-0653-y, which could be considered.

We have thought about the role of the genetic correlation analysis and redesigned our study. The estimated genetic correlations are now used to screen the lipid subfractions and remove those who are highly genetically correlated with the traditional lipid profile. Those subfractions are uninteresting because they do not seem to involve independent biological mechanisms and make the results of the multivariable MR analyses unstable.

We now included the genetic correlations with TG in the results. We found that all VLDL traits (besides a few related to very small VLDL) have extremely high (close to 1) genetic correlations with TG. In consequence, they were excluded from the MR analysis. We no longer group the lipid subfractions in the main manuscript, although the clustering was kept in some tables in the Online Supplement to make the results more organized.

Thanks for suggesting the new method for estimating genetic correlation via the high-definition likelihood. We tried it with a few traits and found the software is much slower than that of LD score regression. Because we needed to compute a very large number of genetic correlations and the precision of LD score regression seemed to be enough for our purposes, we did not switch to the high-definition likelihood approach in this study.

4. Instrument selection – From a practical point of view, It is tricky to jointly model TG with VLDL subtypes characteristics in MVMR because it is unlikely that one would have genetic instruments that strongly predict each exposure once adjusting for the others. Therefore, weak instrument bias is probably an issue here. This is worrying as weak instrument bias in MVMR is not necessarily conservative. At a minimum please provide the conditional F statistics for the MVMR model for each exposure.

Thank you for bringing up this valuable point. This is actually why we did not adjust for TG in multivariable MR analyses of VLDL subfractions in the original submission, but that seemed to have made the results more difficult to interpret. In the revision, most of the VLDL subfractions were excluded from the MR analysis due to their high genetic correlation with TG (see reply to point 3 above). We also decided to use the same set of traditional lipid risk factors in multivariable MR for different subfractions. So for VLDL subfractions, we adjusted for TG, LDL-C, and HDL-C (or TG, ApoB, and ApoA1). In Online Supplement C.4, we reported the conditional Cochran’s Q statistics for the multivariable MR models as described in the following article: Eleanor Sanderson, George Davey Smith, Frank Windmeijer, Jack Bowden, An examination of multivariable Mendelian randomization in the single-sample and two-sample summary data settings, International Journal of Epidemiology, Volume 48, Issue 3, June 2019, Pages 713727, https://doi.org/10.1093/ije/dyy262.

5. The authors present a large amount of results for (unadjusted) univariate MR. Given the known highly pleiotropic nature of variants affecting lipoproteins and their sub-fractions, presentation of these in the main figure adds little and obscures the main results of potential note. The full results could be presented as supplementary data, but the main figures should be restricted to multivariable MR.

We have moved most of the univariable MR results to the Online Supplement. In Figure 2 of the main manuscript, we now report the results of one univariable MR analysis and two multivariable MR analyses.

6. Independent effects of lipid sub-fractions? multivariable MR (MVMR)One of the key issue for lipids sub-fractions (and other metabolites such as fatty acids) is all the sub-fractions are highly correlated to each other. It is almost impossible to include all of the sub-fractions in the same MVMR model. Some methods considering reduce the dimensionality to solve this issue. In this study, the author used a different approach, which split the lipid sub-fractions into three classes, HDL, LDL and TG related. In this way, the MVMR could be conducted with limited number of variables. However, a lot of SNPs are associated with two or more lipid phenotypes, e.g. the CETP example the author provided in Table 3, the SNP is associated with HDL-C, LDL-C and TG. Even in a MVMR model controlling for LDL-C and TG, it is still hard to prove that the highlighted M-HDL-P finding showed an independent effect on CAD.Also, more and more studies considered APOB and APOA1 in the MVMR model (e.g. https://journals.plos.org/plosmedicine/article/comments?id=10.1371/journal.pmed.1003062), so at least worth including APOB/APOA1 in the MVMR model.

Thank you for the suggestions. We have used genetic correlation analysis to pre-screen the lipid subfractions and reduce the dimensionality. Our purpose was not to identify the single "most causal" subfraction. Rather, we were generically interested in whether the lipid subfractions provide any additional value on top of the traditional risk factors. We hope this becomes clearer in the revision.

It is true that many SNPs are still associated with the lipid subfractions under investigation and some traditional lipid risk factors. We have implemented a new procedure to identify genetic markers for medium HDL and HDL size that are not associated with LDL-C or ApoB. In the Discussion, we made it clear that "the role of HDL particles in preventing CAD may be more complicated than, for example, that of LDL cholesterol or ApoB."

Thank you for the pointer to using ApoB and ApoA1 in MVMR. This is now included in our study as well. The results are generally not too different from using LDL-C and HDL-C, although ApoB does seem to have a slightly larger estimated effect than LDL-C and seems to "take away" some of the estimated effect of TG (see Table 2).

7. VLDL MR results – For VLDL results, some of them showed negative effects (CIs did not cross OR=1) in the MVMR model but positive effect in UVMR model, which seems complex and did not fit with the simple interpretation that "VLDL subfraction traits had uniformly positive effect on coronary artery disease" mentioned in the abstract. Can the authors discuss this in a bit more details and try to find the reason?

Thank you for pointing this out. The VLDL subfractions had very high genetic correlations with TG (and to some extent, with ApoB, see Supplement Table B2). So it is very unlikely that a genetic analysis can differentiate the VLDL subfractions from the traditional risk factors, if there is any. Due to this reason, we now exclude most of them from the MR analysis (see also the reply to point 3 above).

8. There are multiple lipidomics measures which are available from the NMR platform, but which are not analyzed, including critical ones such as M-HDL-TG. This needs explanation and/or rectification, since complete data would address multiple points. For instance, the authors dismiss the positive association of S-HDL-TG as being confounded by correlation with VLDL fractions, despite VLDL fractions not having any independent positive effect on CAD in the multivariable analysis. Results for M-HDL-TG would be very informative, and would avoid the possibility that erroneous conclusions are drawn simply because contrary data is missing.

In the genetic correlation analysis, it seems that S-HDL-TG is highly correlated with TG and behave more like a VLDL subfraction than an HDL subfraction (and thus is removed from the main MR analysis, see reply to point 3 above). The other triglyceride measurement for a HDL subfraction, XL-HDL-TG, had a much weaker genetic correlation with TG (see Table B2). We agree with you that an analysis for M-HDL-TG could be quite informative, but unfortunately data about M-HDL-TG were not reported in the two lipidomic GWAS available to us.

9. It is unclear that it is appropriate to adjust for overall LDL-C when analyzing HDL sub-fractions. Attempting to analyze sub-fractions of one class alongside an aggregate measure of another (which is correlated with aggregate measures of the first) seems suspect and at the very least requires detailed and careful justification.

Thank you for this interesting point. As explained in our reply to point 6, our purpose was not to identify the single "most causal" subfraction. Rather, we were generically interested in whether the lipid subfractions provide any additional value on top of the traditional risk factors. Relatedly, any subfraction discovered in the MR analysis is not necessarily the only causal agent. In the Discussion, we made it clear that "it is possible that HDL cholesterol, HDL subfractions, and HDL particle size are all phenotypic markers for some underlying causal mechanism." We then discussed possible connections with the HDL function hypothesis and the cholesterol efflux capacity.

10. The introduction anticipates the resolution of conflicting findings in relation to LDL size. This is dealt with in discussion, but it needs to be be highlighted beforehand in results. Since levels of different sized LDL and VLDL particles are strongly correlated, a statement that different sizes are not differently atherogenic is not supported by the data. To draw this conclusions would require multivariable MR in which the different size classes were adjusted for each other. In fact, the statement is directly contradicted by the clear associations of VLDL and (particularly) LDL diameter with CAD.

Thank you. We now made it clear in the Results section that "The mean diameter of LDL particles (LDL-D) showed a harmful effect on MI in univariable MR, though the effect was smaller than those of the LDL subfractions in univariable MR. The estimated effect of LDL-D was attenuated in the multivariable MR analyses." In the Discussion, we concluded that "we find some weak evidence that larger LDL particle size may have a small harmful effect on myocardial infarction and coronary artery disease." We were a little conservative in this conclusion because there was not a very convincing evidence from the multivariable MR analyses. The individual LDL subfractions had the right trend in multivariable MR (see Figure C4), but the confidence intervals were quite wide due to strong genetic correlation with LDL-C/ApoB. In one multivariable MR, LDL-D showed a protective effect, but it was barely statistically significant.

11. Similarly the statement that medium and small HDL particles are protective is contradicted by the fact that HDL particle diameter shows no association at all with CAD risk.

Thank you for this observation. HDL-D was not associated with CAD risk in univariable MR, but did show a positive effect in multivariable MR analyses now (notice that HDL-C/ApoA1 is now included in the multivariable MR for HDL-D, but was not included in the original submission). It turns out that adjusting for HDL-C/ApoA1 is quite important here, see Table 2. We do not interpret this as a necessary contradiction to the findings about small and medium HDL particles, but do acknowledge that the mechanism involving HDL particle size may be quite complicated. In the Discussion, we stated the following: "Notice that the harmful effect of larger HDL particle diameter found in this study relies on including HDL-C or ApoA1 in the multivariable MR analysis. Thus, the role of HDL particles in preventing CAD may be more complicated than, for example, that of LDL cholesterol or ApoB. It is possible that HDL cholesterol, HDL subfractions, and HDL particle size are all phenotypic markers for some underlying causal mechanism."

12. Selection of lipid sub-fractions – 82 lipid sub-fractions were selected but it is not directly clear why these fractions (but not others) were selected. It will be helpful to have a DAG to explain the inclusion and exclusion criteria. There is an even more fundamental question: Does this data justify a re-classification of NMR lipoprotein subclasses? Most people would agree that 82 is a few too many. Can the authors nominate a rational condensation of NMR lipoprotein sub-fraction data into a handful of independently predictive parameters with putative mechanisms? Ideally, these would be targets for therapy and indicators of response.

We did not select the lipid subfractions. The 82 subfractions were those reported in the lipidome GWAS that are related to VLDL, LDL, IDL, and HDL. This is now made clear in the beginning of Materials and methods.

Thank you for the suggestion about condensing the subfraction data. As described in the reply to point 3, we now use genetic correlation to screen the subfractions and remove the ones with highly genetic correlations with the traditional lipid risk factors (TG, LDL-C, HDL-C, ApoB, or ApoA1). This left us with 27 traits. Although one may argue that is still a few too many, we wanted to be more agnostic about generating hypotheses and let the data to speak for themselves. Relatedly, the purpose of this study was not to identify the single "most causal" subfraction. Rather, we were generically interested in whether the lipid subfractions provide any additional value on top of the traditional risk factors. In the Discussion, we pointed that the role of HDL particle size may be complicated and avoided making simple-minded conclusions like "increasing M-HDL-P by any means is causally protective, period".

13. HDL MR results – For S-HDL-TG, this is a very complex case and it is doubtful if the current setting can distinguish its effect driven by HDL from its effect driven by TG. I personally think its strong genetic correlation with VLDL (Figure 1) implies that the effect of S-HDL-TG on CAD is driven by TG. However, the MVMR adjusted TG showed similar effect estimate, which suggest the effect is independent to TG (as mentioned about, the reviewer is not sure the MVMR model can prove independent effect). The authors may need to consider integrating some biological information of each instruments been used here to dig into the button of this case study. For example, whether each of instrument colocalized with the expression level of the cis gene in multiple tissues. For the colocalized genes, are they related to HDL pathway or TG pathway etc. This could be very complex so the reviewer suggests to explain the results with caution.For M-HDL-P, it showed some correlations with HDL-C and APOA1, so hard to say it is not correlated with HDL-C at all.

We agree that it is impossible to make any conclusions about S-HDL-TG because of its high genetic correlation with TG. In fact, S-HDL-TG has been excluded from the main MR analysis in our new design with phenotypic screening. It is true that M-HDL-P is genetically correlated with HDL-C and ApoA1, but adjusting for HDL-C and ApoA1 did not alter the estimated effect of M-HDL-P (the story is different for HDL-D). In summary, we think there is enough evidence to suggest that S-HDL-P and MHDL-P are risk factors that are reasonably independent of the traditional HDL-C and ApoA1. However, we also made it clear in the Discussion that the role of HDL particle size may be quite complicated.

Thank you for the suggestion about integrating biological information. We have used a new procedure to identify instruments associated with HDL particle sizes but not with LDL-C or ApoB. We have also included the cis genes that show up in eQTL data.

14. The identification of a leading role for mid-sized HDL particles presents a tantalizing opportunity to link the latter finding with recent NMR studies of cholesterol efflux, but this is not pursued. It would be worth flagging this opportunity at least in the discussion if you cannot easily address it. How do the results (M-HDL-P etc) relate to NMR assessments of cholesterol efflux capacity (eg Direct estimation of HDL-mediated cholesterol efflux capacity from serum). Sanna Kuusisto, Michael V. Holmes, Pauli Ohukainen, Antti J. Kangas, Mari Karsikas, Mika Tiainen, Markus Perola, Veikko Salomaa, Johannes Kettunen, Mika Ala-Korpeladoi: https://doi.org/10.1101/396929, now published in Clinical Chemistry doi: 10.1373/clinchem.2018.299222.

Thank you. We agree this is a tantalizing opportunity and a MR study for cholesterol efflux capacity can complement the results here. This is mentioned in the last paragraph of Discussion as a future work.

15. The multiple versions of the analysis using different combinations of datasets do provide worthwhile technical replication. However, these do not provide wider replication of the findings since any shortcomings in one analysis (e.g. in failing to fully address pleiotropy) will inevitably be present in another.

We agree with your assessment and have moved the results of the replication analyses to the Online Supplement.

16. A failure to find evidence against the INSIDE assumption, on the basis of only a small number of SNPs, is a very weak basis for making a claim that there is not horizontal pleiotropy affecting LDL-C or TG. The arguments made by the authors on this point are very weak.

We agree, although this is the best we could do to assess the InSIDE assumption. We have deemphasized this in the paper.

17. The issue of horizontal pleiotropy, especially as it applies to m-HDL-P in relation to the inSIDE assumptions for TG and LDL-C, is justified in discussion and acknowledged in limitations, but it is very difficult follow. For example, from table 3, TG is strongly associated with all traits apart from LOC157273, LIPG and DOCK6. Accordingly, the simple clinician is tempted to regard m-HDL-P as a surrogate for TG. Perhaps there needs to be greater clarity concerning the concepts of genetic correlation and "weak instrument bias".

Thank you for the suggestion. Although TG is strongly associated with several instruments for M-HDL-P, their associations were actually in different directions (see Table D9 in the Supplement) and are consistent with the InSIDE assumption. So M-HDL-P should not be regarded as a surrogate for TG. We have moved the diagnostics plots for M-HDL-P to the Online Supplement and only presented the genetic markers that are not associated with LDL-C or ApoB in the main paper. The four genetic markers we identified had no or weak associations with TG.

18. Discussion – "Our results for the HDL sub-fractions support the conclusion that HDL-C is not the causally relevant biomarker." The MR estimate of S-HDL-TG, M-HLD-C and M-HDL-L showed the complexity of HDL-C on CAD. So this claim seems against the main finding of the manuscript.

We meant HDL-C is not causal in the narrow sense that any intervention that increases HDLC will protect against CAD. Essentially we were trying to convey that the role of HDL is complicated, but we agree that this statement is not accurate. In the revised Discussion, we removed this sentence and made it clear that "Thus, the role of HDL particles in preventing CAD may be more complicated than, for example, that of LDL cholesterol or ApoB. It is possible that HDL cholesterol, HDL subfractions, and HDL particle size are all phenotypic markers for some underlying causal mechanism."

[Editors' note: further revisions were suggested prior to acceptance, as described below.]

The manuscript has been improved but there are some remaining issues noted by the reviewers that need to be addressed, as outlined below:You have made substantial efforts to restructure the manuscript (e.g. using genetic correlation as a filtering step in the new version), highlighted the key findings of HDL particle size on CAD as well as identified four potential causal genes linking HDL particle size with CAD. We are in general more convinced that the new findings from this study will bring good value for the existing argument for effect of HDL on CAD.

We appreciate the additional comments and suggestions by the reviewers. We have revised the manuscript accordingly.

The only suggested change we did not implement is the second part of issue 5:

Also, to make this section more informative, you can try a formal Wald ratio + colocalization analysis to estimate the putative causal effects of these HDL size related genes on CAD (rather than just did a SNP lookup in different GWASs).

This is an interesting suggestion, but we did not make the change because we think the Wald ratio estimator might not provide a full picture of the potential mechanisms involved. In particular, the genetic markers in Figure 3 are not only related to HDL particle size but also the overall HDL cholesterol. The last two markers also have a small association with triglycerides. We are worried that reporting the Wald ratio estimates may provide an oversimplified summary of the information in Figure 3 to the reader. That being said, we are also submitting the raw data for Figure 3. So if some reader is interested in knowing the putative causal effect, they can also take a ratio easily themselves.